# Evaluation of ocean dimethylsulfide concentration and emission in CMIP6 models

Josué Bock[1], Martine Michou[1], Pierre Nabat[1], Manabu Abe[2], Jane P. Mulcahy[3], Dirk J.L. Olivié[4], Jörg Schwinger[5], Parvadha Suntharalingam[6], Jerry Tjiputra[5], Marco van Hulten[7], Michio Watanabe[2], Andrew Yool[8], and Roland Séférian[1]

[1]CNRM, Université de Toulouse, Météo-France, CNRS, Toulouse, France
[2]Research Institute for Global Change, Japan Agency for Marine-Earth Science and Technology (JAMSTEC)
[3]Met Office Hadley Center, Exeter, UK
[4]Norwegian Meteorological Institute, Oslo, Norway
[5]NORCE Climate and Bjerknes Centre for Climate Research, Bergen, Norway
[6]School of Environmental Sciences, University of East Anglia, Norwich Research Park, Norwich NR4 7TJ, UK
[7]Geophysical Institute, University of Bergen and Bjerknes Centre for Climate Research, Bergen, Norway
[8]National Oceanography Centre, European Way, Southampton, SO14 3ZH, United Kingdom

**Correspondence:** Josué Bock (josue.bock@laposte.net) and Martine Michou (martine.michou@meteo.fr)

**Abstract.** Characteristics and trends of surface ocean dimethylsulfide (DMS) concentrations and fluxes into the atmosphere of four Earth System Models (ESMs: CNRM-ESM2-1, MIROC-ES2L, NorESM2-LM and UKESM1-0-LL) are analysed over the recent past (1980–2009) and into the future, using Coupled Model Intercomparison Project 6 (CMIP6) simulations. The DMS concentrations in historical simulations systematically underestimate the most widely-used observed climatology, but compare
more favourably against two recent observation based datasets. The models better reproduce observations in mid to high latitudes, as well as in polar and westerlies marine biomes. The resulting multi-model estimate of contemporary global ocean DMS emissions is of $16$–$24\ \mathrm{Tg\,S\,year^{-1}}$, which is narrower than the observational-derived range of 16 to $28\ \mathrm{Tg\,S\,year^{-1}}$. The four models disagree on the sign of the trend of the global DMS flux from 1980 onwards, with two models showing an increase and two models a decrease. At the global scale, these trends are dominated by changes in surface DMS concentrations
in all models, irrespective of the air-sea flux parameterisation used. In turn, three models consistently show that changes in DMS concentrations are correlated with changes in marine productivity, however the latter is poorly constrained in the current generation of ESMs, thus limiting the predictive ability of this relationship. In contrast, a consensus is found among all models over polar latitudes where an increasing trend is predominantly driven by the retreating sea-ice extent. However, the magnitude of this trend between models differs by a factor of three, from 2.9 to $9.2\ \mathrm{Gg\,S\,decade^{-1}}$ over the period 1980–2014, which
is at the low end of a recent satellite-derived analysis. Similar increasing trends are found in climate projections over the 21st century.

# 1   Introduction

Despite several decades of investigations, the quantification of interactions between aerosols and climate remains poorly constrained and understood (Bender, 2020). Recent work using the latest generation of ESMs suggests that aerosol-climate interactions constitute a key driver of the inter-model spread of the simulated response to rising $CO_2$ (Meehl et al., 2020). One of the sources of uncertainties is related to natural aerosols whose relative abundance compared to that of anthropogenic aerosols directly influences the level of the anthropogenic aerosol forcing (Schmidt et al., 2012; Carslaw et al., 2013). Dimethylsulfide (DMS) is a by-product of microbial food webs and is considered the largest natural source of sulfur to the atmosphere (e.g., Liss et al., 1994; Simó, 2001; Stefels, 2000). Once in the atmosphere, DMS is mainly oxidised into $SO_2$ and then gas-phase sulfuric acid, which rapidly condenses onto pre-existing aerosol particles or nucleates to form new sulfate aerosol particles (Carslaw et al., 2010; Liss et al., 2014). Among natural aerosols, sulfate aerosols formed from DMS represent a major part of the aerosol-climate interactions in large pristine regions such as the Southern Ocean (Mulcahy et al., 2018, and references therein) or the Arctic (Abbatt et al., 2019, and references therein).

Changes in climate variables, for instance surface wind, sea surface temperature (SST) or downwelling irradiance, can affect both the production of DMS and its surface concentration, as well as its transfer rate from the ocean to the atmosphere, potentially driving a DMS-climate feedback (Vallina and Simo, 2007; Carslaw et al., 2010; Quinn and Bates, 2011). The importance of such a feedback is debated due to a lack of comprehensive observations operating across a wide range of Earth system realms, from marine biogeochemistry to cloud microphysics (Boucher et al., 2014, and references therein). Therefore, modelling estimates of the strength of this feedback are poorly constrained. The latest assessed value of this feedback is $-0.02 \, \mathrm{W \, m^{-2} \, K^{-1}}$ (Ciais et al., 2013) which has been estimated from a single model (HadGEM2-ES). A recent estimate deduced from three CMIP6 models (GISS-E2-1-G-CC, NorESM2-LM, UKESM1-0-LL) suggests a slight amplification of global warming due to a positive feedback of $0.005 \pm 0.006 \, \mathrm{W \, m^{-2} \, K^{-1}}$ (Thornhill et al., 2021). These global estimates hide large regional differences both in terms of radiative forcing and in terms of changes in DMS emissions under global warming (Thornhill et al., 2021). So far, studies have focused more closely on high latitudes regions eventhough a few recent ones demonstrate the dominant role of DMS on marine low cloud aldebo over most oceans (e.g., Quinn et al., 2017) or illustrate regional impacts on low latitudes (Zavarsky et al., 2018). In polar regions, studies suggested an overall negative DMS feedback (e.g., Kim et al., 2018; Mahmood et al., 2019). However, because they are particularly sensitive to global warming, the quantification of feedback in these regions is complex (Goosse et al., 2018). Therefore, an improved understanding of how the pattern of marine DMS emissions may change with climate and environmental changes is required to constrain the magnitude of the DMS climate feedback.

Recent observations and mesocosm experiments have improved our understanding of how changes in microalgae dominance and DMS production in response to climate warming (e.g., Blanchard et al., 2012), eutrophication (e.g., Mackenzie et al., 2011; Gypens and Borges, 2014) or ocean acidification (Six et al., 2013; Schwinger et al., 2017; Hopkins et al., 2020) may change DMS emissions in the future. Furthermore, satellite observations have been used recently to derive global estimates of DMS seawater concentration for over a decade (Galí et al., 2018). The resulting algorithm was developed at the global scale but

further tuned over the northern latitudes, allowing an assessment of the recent evolution of DMS in this region (Galí et al., 2019). These advances coincide also with those of global models, from ocean biogeochemistry ones (Le Clainche et al., 2010; Séférian et al., 2020) to full ESM ones enabling investigations on either (i) the physical factors that impact DMS behaviour, for instance Xu et al. (2016) demonstrate that there seems to be a two-way interaction between DMS and ENSO in the tropical region, or (ii) the ecological factors, for instance representing in the model more explicitly diverse phytoplankton groups (e.g., *Phaeocystis*: Wang et al., 2015).

In this paper, we use the most up-to-date generation of observational data products and long-term measurements to assess estimates of the surface ocean DMS concentrations and emissions to the atmosphere from the latest generation of ESMs, using their contributions to the 6[th] phase of the Coupled Model Intercomparison Project (Eyring et al., 2016). The goal of this work is two-fold. First, we aim to pull together various lines of evidence (observational data products, long-term measurements, model simulations) that will be used in further multi-model analysis of DMS-climate interactions. And, second, by combining these lines of evidence, we provide an assessment of both the direction and the magnitude of the change in marine DMS emissions.

In Section 2, we provide details of the key characteristics of the ESMs and the observational datasets used in this work. In Section 3, we focus on the analysis of the modern mean state. Building upon this analysis, we investigate the recent and future evolution of marine DMS concentration and emission in Section 4. We assess the reliability of the model predictions in the light of key biogeochemical and physical drivers in Section 4.2. We further focus on the Arctic domain where model long-term behaviours are scrutinized against long-term measurements of DMS concentrations and emissions in Section 4.3.

## 2 Models and observational datasets

### 2.1 Models

The present work draws on the results of four state-of-the-art ESMs that have contributed to CMIP6 (CNRM-ESM2-1, MIROC-ES2L, NorESM2-LM and UKESM1-0-LL), whose key characteristics are provided in Table 1.

**Table 1.** Key characteristics of the ocean and marine biogeochemical components of the ESMs. Column 2: horizontal grid points (tripolar grids) and number of vertical levels. Column 4 in brackets: prognostic variables involved in the DMS parameterisations.

| ESM name | Ocean model (grid) | Marine biogeochem. model | Key Characteristics of the DMS scheme | Reference |
|---|---|---|---|---|
| CNRM-ESM2-1 | NEMO (294×362, 75 z-levels) | PISCESv2-gas | Prognostic (variable phyto. S/C ratios and DMSP-DMS yield) | Séférian et al. (2019) |
| MIROC-ES2L | COCO (256×360, 62 sigma-levels) | OECO2 | Diagnostic (chlorophyll, mixed-layer depth) | Hajima et al. (2020) |
| NorESM2-LM | BLOM (360×385, 53 sigma-levels) | iHAMOCC | Prognostic (export production, temperature) | Seland et al. (2020) |
| UKESM-1-0-LL | NEMO (330×360, 75 z-levels) | MEDUSA-v2 | Diagnostic (chlorophyll, surface irradiance, nitrate) | Sellar et al. (2019) |

Table 1 shows that the ocean component of the four ESMs studied here offer a relatively coarse resolution: they all use tripolar grids (such as eORCA1 in CNRM-ESM2-1), with a nominal grid size of 1° and grid refinements in the tropics (circa 0.3°). As a consequence, we can anticipate that these models will suffer from deficiencies in replicating observations in regions where small-scale features are important, such as in coastal areas.

Nonetheless, as documented in the reference papers of CNRM-ESM2-1, MIROC-ES2L, NorESM2-LM and UKESM1-0-LL, these ESMs are able to simulate the main large-scale features of the ocean circulation. Recent work has also suggested that these models have improved their performance in simulating the ocean mixed-layer depth (MLD), an important driver for marine biogeochemistry and marine DMS emissions (Séférian et al., 2020).

### 2.1.1 DMS concentration and flux parameterisations

As shown in Table 1, the four studied ESMs use DMS parameterisations of various complexity. Two models use empirical parameterisations to compute DMS concentrations from chlorophyll and other variables (MIROC-ES2L and UKESM1-0-LL), while the other two use prognostic models including marine biota (CNRM-ESM2-1, NorESM2-LM). Characteristics of these parameterisations are essential to understand model deficiencies at simulating observed features of DMS concentration. Besides, a close look at these parameterisations is also necessary to infer the ability of the biogeochemistry models to simulate the evolution of the DMS concentration and emission in the future, and the climate feedback that it may trigger. Here, we detail DMS parameterisations used in the four ESMs.

**Prognostic DMS parameterisations** In CNRM-ESM2-1, DMS concentration is computed by the biogeochemical model PISCES (Aumont and Bopp, 2006), embedded within the global general ocean circulation model NEMO. A prognostic DMS scheme was introduced in PISCES by Bopp et al. (2008), and updated by Belviso et al. (2012) based on the PlankTOM5 model of Vogt et al. (2010). The version of PISCES used in CNRM-ESM2-1 for CMIP6 is PISCESv2-gas, based on PISCES-v2 (Aumont et al., 2015) with the addition of a specific module to compute the cycle of gases relevant to climate. In brief, the model simulates three processes releasing the precursor of DMS, dimethylsulfoniopropionate (DMSP) to seawater: grazing by zooplankton, exudation by phytoplankton, and cell lysis. Each of these processes is parameterised specifically for the two phytoplankton functional groups represented in PISCES: nanophytoplankton and diatoms. DMSP is then converted to DMS with yields that increase with bacterial nutrient stress. Three more processes describe the sinks for DMS in seawater: bacterial and photochemical degradation, and ventilation to the atmosphere. A more thorough description of PISCES can be found in Belviso et al. (2012), with some adjustments further listed in Masotti et al. (2016). An additional stress factor accounting for the change in pH is also included in this version following the study of Six et al. (2013). The fluxes to the atmosphere are then computed using the parameterisation of gas exchange coefficients of Wanninkhof (2014). The air resistance is neglected. There is currently no online coupling of the fluxes of DMS towards the atmospheric model in CNRM-ESM2-1. Instead, a prescribed DMS flux is applied as an input to the aerosol scheme (see Michou et al., 2020, for details).

NorESM2-LM includes a fully interactive description of the DMS cycle across the ocean-atmosphere interface following Kloster et al. (2006). As opposed to PISCES, the conversion of DMSP to DMS is not explicitly described in the model. Instead, DMS is directly released in the water, and is computed as a function of temperature and simulated detritus export production (Tjiputra et al., 2020). DMS production is further modified by the export rate of opal and $CaCO_3$ shell material – that is, calcite producing organisms are assumed to have a higher sulfur-to-carbon ratio than opal producing organisms. Although a tunable pH dependency, that was not present in the original parameterisation of Kloster et al. (2006), has been implemented in NorESM2, it has not been activated in CMIP6 runs (Tjiputra et al., 2020). As for CNRM-ESM2-1, three sink processes for

ocean DMS are accounted for: bacterial consumption, photolysis and ventilation to the atmosphere. As in CNRM-ESM2-1, the latter is parameterised according to Wanninkhof (2014). DMS concentration in the air is modified via chemical reactions as described in Kirkevåg et al. (2018) and Seland et al. (2020).

**Diagnostic DMS parameterisations** Compared to CNRM-ESM2-1 and NorESM2-LM, MIROC-ES2L and UKESM1-0-LL use a much simpler approach to simulate the marine DMS cycle. Indeed, DMS concentration is diagnosed using empirical

parameterisations, that relate the DMS concentration to other marine biogeochemical or ocean hydrodynamical variables such as chlorophyll (hereafter Chl) and MLD. Despite their relative simplicity, the two parameterisations remain quite different.

In MIROC-ES2L, the seawater concentration of DMS is computed according to the parameterisation of Aranami and Tsunogai (2004), which relates the sea surface DMS concentration to the MLD, and to surface water Chl concentration. This DMS parameterisation is a modified version of that of Simó and Dachs (2002), calibrated with further measurements carried out in the

northern North Pacific (Aranami and Tsunogai, 2004). These parameterisations distinguish between two regimes, depending on the Chl/MLD ratio:

$$
\text{DMS} = \begin{cases} 60.0/\text{MLD} & \text{if Chl/MLD} < 0.02 \text{ mg/m}^4 \\ 55.8 \times (\text{Chl/MLD}) + 0.6 & \text{if Chl/MLD} \geq 0.02 \text{ mg/m}^4. \end{cases} \tag{1}
$$

Low Chl/MLD ratio is found in open ocean, and holds for over 80 % of the ocean global surface (Simó and Dachs, 2002). In this regime, DMS depends solely on MLD with an inverse relationship (dilution model: DMS×MLD=constant). Conversely,

high Chl/MLD ratio occurs either with very high Chl concentration, or with moderate-low Chl concentration in shallow mixed waters. Only in these situations is the DMS concentration positively correlated with Chl. Both MLD and Chl are simulated by the ocean biogeochemical model OECO-v2 embedded in MIROC-ES2L (Hajima et al., 2020). Here, the MLD is defined as the depth where the potential density becomes larger than that at the sea surface by $0.125 \text{ kg m}^{-3}$ (Simó and Dachs, 2002). The flux of DMS to the atmosphere is also computed according to Aranami and Tsunogai (2004). The gas transfer velocity is

calculated following Nightingale et al. (2000), but the Schmidt number used for DMS adjustment is calculated according to Wanninkhof (2014), as advised in the OMIP-BGC protocol (Orr et al., 2017). The DMS emission is then considered by the aerosol module (Hajima et al., 2020).

In UKESM1-0-LL, the seawater concentration of DMS is computed within the ocean biogeochemistry model MEDUSA (Yool et al., 2013) and is interactively coupled with the atmosphere. The parameterisation of DMS concentration is based

on the work of Anderson et al. (2001), and linearly relates the DMS concentration to a composite variable formed by the logarithm of the product of Chl concentration ($C$, $\text{mg m}^{-3}$), light ($J$, mean daily shortwave, $\text{W m}^{-2}$) and a nutrient term ($Q$, dimensionless) that depends on nitrate concentration. The parameterisation includes a minimum DMS concentration if the composite variable is lower than a threshold, $s$, as follows:

$$
\text{DMS} = \begin{cases} a & \text{if } \log_{10}(CJQ) \leq s \\ a + b \times (\log_{10}(CJQ) - s) & \text{if } \log_{10}(CJQ) > s. \end{cases} \tag{2}
$$

In the original parameterisation, the fitted parameter values were: $a = 2.29$, $b = 8.24$ and $s = 1.72$ (Anderson et al., 2001). During calibration of UKESM1-0-LL, the minimum DMS concentration ($a$) was changed to 1 nM and the threshold ($s$) was adjusted to 1.56 (Sellar et al., 2019). This tuning was required to reduce the excessively strong negative forcing induced by the higher DMS minimum concentration. Finally, the flux of DMS from the surface ocean to the atmosphere is parameterised according to the air-sea gas transfer scheme of Liss and Merlivat (1986). DMS concentration in the atmosphere is subsequently modified through a number of gas phase aerosol precursor reactions of the UKESM1-0-LL stratospheric/tropospheric chemistry scheme (see Mulcahy et al., 2020, Table 2 for the list of reactions).

### 2.1.2 Simulations

In this paper, we use monthly outputs of the CMIP6 historical (1850–2014) and ssp585 scenario (2015–2100) experiments. All datasets were downloaded from ESGF nodes. The number of realisations of each model for both experiments is reported in Table 2.

NorESM2-LM has been run with two different grid resolutions of the atmospheric model: the version labeled LM has a low atmosphere (250 km) and a medium ocean (100 km) resolution, while MM has a medium atmosphere (100 km) and a medium ocean resolution. Both versions have the same number of realisations (3 for historical, and 1 for ssp585). We evaluated and compared both versions, and they appeared to be very similar regarding the various metrics used hereafter. Thus, only NorESM2-LM is presented in the study.

UKESM1-0-LL realisations of the historical experiment include two different forcings variants, f3 (runs 5, 6 and 7) and f2 (all other available runs). The difference between both forcings is related to the stratospheric sulfate AODs that influence stratospheric sulfur chemistry. In f3 forcing, the AODs were accidentally kept at 1850 values (Sellar et al., 2020, Table A8). However, this difference is believed to have a "close to non-existent" impact (Colin Jones, personal communication, Apr. 2020). All runs are thus analysed together, whatever the forcing variant. Regarding the historical experiment, the outputs of realisations 13–15 of UKESM1-0-LL are only partly available on ESGF, depending on the variable. To keep consistency in the analysis, these three realisations were always discarded when available, only the sixteen other realisations (1–12 and 16–19) were used.

**Table 2.** Number of available members used in the ensemble means, and in brackets reference of the dataset, for the historical and ssp585 CMIP6 simulations.

| Model | historical | ssp585 |
| --- | --- | --- |
| CNRM-ESM2-1 | 11 (Seferian, 2018) | 5 (Voldoire, 2019) |
| MIROC-ES2L | 10 (Hajima et al., 2019) | 10 (Tachiiri et al., 2019) |
| NorESM2-LM | 3 (Seland et al., 2019a) | 1 (Seland et al., 2019b) |
| UKESM1-0-LL | 16 (Tang et al., 2019) | 5 (Good et al., 2019) |

Our analysis focusses on surface ocean DMS concentration and marine emission of DMS to the atmosphere (variables dmsos and fgdms in CMIP6). Where relevant, we use other variables such as the ocean surface Chl concentration (chlos), the vertically integrated marine primary production (intpp), the 10-meter wind speed (sfcWind), the sea surface temperature (tos), or the sea-ice cover (siconc) for additional analysis. When compared to observations or between each other, model outputs are interpolated on a regular Mercator grid of 1° using CDO remapping functions (Schulzweida, 2019, see Appendix A1.1 for more information). In the following, the multi-model ensemble mean (hereafter MMM) is calculated by averaging the ensemble means of the four models using an equal weight for each model, regardless of the number of realisations.

## 2.2 Observational and reference datasets

### 2.2.1 Surface ocean DMS concentration

In this study, we compare model outputs to three climatologies of surface ocean DMS concentration.

First, the widely used climatology of Lana et al. (2011) (hereafter referred to as L11), which is based on a large database of in-situ measurements (over 47,000 data) from the Global Surface Seawater (GSS) DMS database (https://saga.pmel.noaa.gov/dms/). Most measurements were performed between 1980 and 2009. The interannual variations are not accounted for, thus the resulting monthly climatology is considered to be representative as an average over this period. The dataset is processed as follows: no quality control is applied on the data, but the largest values above the 99.9 percentile are removed (i.e., values above 148 nM). Monthly data is first binned into a $1° \times 1°$ grid, and a monthly mean is then calculated in each of the 54 static biogeographical provinces defined by Longhurst (2007, see Appendix A2 for details). In each of these provinces, a minimum of three data points is required to get a valid (monthly) value. For regions with at least four valid values over the year, a temporal interpolation is applied to fill the gaps. Conversely, in provinces with three or less months of data, neighboring or biogeographically equivalent provinces are used to construct an annual cycle, which is scaled with the few available data, if any. This results in a "first-guess field" global dataset (Lana et al., 2011, section 2.2). Last, a distance-weighted interpolation process (radius of influence of 555 km) is applied twice to smooth the resulting global monthly climatology: a first time to smooth the transitions across the province borders, and a second time after re-introducing individual in-situ data (Lana et al., 2011, section 2.3 and supplementary Fig. S2). On top of the monthly climatology, L11 provides an assessment of the uncertainty range, through two other monthly datasets representing the estimated minimum and maximum DMS concentrations. The raw data, binned into the $1° \times 1°$ grid, is also provided along with the climatology.

Despite the large number of measurements included in this DMS climatology, the spatial and temporal data coverage is limited (Lana et al., 2011, Figs. 1 and S1), and many regions remain poorly documented. Thus, as pointed out by Tesdal et al. (2016), small scale features are transformed into large scale ones by the interpolation procedure, and anomalous values observed at local scale could induce bias when extrapolated across data-sparse regions. This is illustrated by Hayashida et al. (2020), who show that the entire Arctic region in L11 is based on extremely limited data (0–4 % areal coverage north of 60° N). The resulting extrapolation of open water DMS concentration to sea-ice covered areas, where primary production is presumably lower, may lead to a positive bias in L11. Another potential positive bias in L11 stems from the overrepresentation

of biologically productive conditions in the in-situ DMS database from which L11 is built upon. This is supported by the study of Galí et al. (2018, Fig. 7 and Sect. 4.1) who pointed out that the distribution of DMS concentration in L11 is right-skewed as compared to DMS concentration derived from satellite chlorophyll measurements. Conversely, recent studies report on high

DMS concentrations measured in the North Atlantic (Bell et al., 2021) and in a coastal station of the West Antarctic Peninsula or in the Ross Sea (Webb et al., 2019; del Valle et al., 2009, respectively) which are not represented in L11. To conclude, although L11 presents some weaknesses that are inherent to the original data and the interpolation methodology, it has been considered so far as a reference (Tesdal et al., 2016) and is the only DMS climatology solely based on in-situ measurements. It has also been widely used to calibrate or validate other DMS estimation techniques (see for instance Galí et al., 2019). We

thus used L11 as the leading reference climatology in this study, along with the more recent products presented hereafter. The annual mean of this climatology is shown in Fig. 1. Its global area-weighted mean is 2.35 nM (area-weighted median 2.25 nM, see Table 3).

Another methodology using an artificial neural network (ANN) has very recently been developed by Wang et al. (2020) (here referred to as W20) in order to improve the interpolation method used by Lana et al. (2011). This study relies on the

210 same database of in-situ DMS measurements, which now contains twice as many measurements (over 93k after removing concentrations below 0.1 nM and above 100 nM) as in the study of Lana et al. (2011). Wang et al. (2020) used a machine learning algorithm, trained with existing DMS data along with a set of eight ancillary variables (MLD, SST, sea surface salinity (SSS), photosynthetically active radiation (PAR), Chl, nitrate, phosphate and silicate), plus the time and location. While DMS data measurements are scarce in vast areas of the oceans, and especially at high latitudes during the winter,

these other variables are better constrained through observations or climatologies. After training the ANN on a subset of the available data and validating the technique on another subset, it is thus possible to use the ancillary variables as predictors of DMS concentration in undersampled areas. Along with this ANN method, the authors also performed a conventional linear and multi-linear regression analyses, to compare the skills of the three methods. While the latter reproduces only 39 % of the observed DMS variance, the ANN approach captures 66 % of it. The yearly mean of the climatology derived from the ANN

method is shown in Fig. 1. Its global area-weighted mean is 1.75 nM (area-weighted median 1.65 nM, see Table 3).

Another strategy to produce a reliable climatology of sea surface DMS concentration has been proposed recently by Galí et al. (2018) (here referred to as G18). These authors developed a remote sensing algorithm to derive total DMSP (DMSPt) from the Chl concentration and the ratio between euphotic layer depth and MLD, which are both satellite-retrieved (Galí et al., 2015). In a subsequent work, Galí et al. (2018) proposed a further parameterisation to derive DMS concentration from

225 DMSPt and PAR. The resulting climatology of Galí et al. (2018) can be regarded as an intermediate between L11 and W20 on the one side, which both use DMS in-situ measurements to infer DMS concentration over the globe, and the models using empirical parameterisations for DMS on the other side, where the input variables (Chl, MLD, PAR) are calculated instead of being satellite-retrieved observation. This method also has its own limitations, and the developed algorithm used to relate the proxies together has to be tuned. While the resulting parameterisation can be better constrained in specific basins, it remains

approximate when applied to the global oceans. Another limitation of this approach is the lack of satellite observations over sea-ice and at low solar elevations, resulting in observational gaps in high latitudes ($> 48°$) in winter. In coastal regions,

the remotely sensed Chl signal may be biased by the presence of riverine coloured dissolved organic matter (CDOM), and ultimately lead to an overestimation of the computed DMS concentration (Galí et al., 2019; Hayashida et al., 2020). The annual mean of this climatology is shown in Fig. 1. Compared to L11, the global mean G18 DMS concentration is 33 % smaller: 1.69 nM (weighted median 1.53 nM, see Table 3).

### 2.2.2 Marine DMS emissions

The flux of DMS to the atmosphere has been assessed by several authors, but the resulting datasets are not readily available. However, a recent work from Granier et al. (2019), as part of the European Copernicus Atmosphere Service (CAMS) project, provides a climatology of DMS flux (hereafter referred as CAMS19). We used the latest version (3.1) of this dataset. This climatology is computed using the DMS sea surface concentration from L11, the gas exchange coefficient of Nightingale et al. (2000) (with the quadratic formulation of the Schmidt number as a function of the temperature from Wanninkhof (2014); M. Gauss, personal communication), and wind fields from ERA5 reanalysis over the period 2000-2019 (at 0.5° and 3 h resolution). The calculated flux also accounts for the sea-ice cover, which is usually assumed to linearly reduce the DMS emission. Indeed, gas transfer across the sea-ice and in partly ice-covered areas involves a series of complex processes and is subject to debate (Lovely et al., 2015; Rutgers van der Loeff et al., 2014), however the linear scaling of the flux to the open-water fraction is generally accepted as a good first-order approximation (Prytherch et al., 2017). The resulting DMS flux is provided as daily means on a 0.5° grid, and can be accessed from https://eccad3.sedoo.fr/.

## 3 Modern mean state

### 3.1 DMS surface ocean concentration

#### 3.1.1 Spatial pattern analysis

The annual mean DMS concentration over the period 1980–2009 is plotted in Fig. 1 for the 4 studied models, the MMM, L11, G18 and W20. The global area-weighted mean and median are also provided in Table 3. In Fig. 1, an additional hatching is added on the models to show the location where the modelled DMS concentration is not in the range of the three climatologies (the range being defined by the minimun and the maximum of the three climatologies, including the minimum and maximum range given by L11). Note that the Arctic region which is undocumented in G18 throughout the year is excluded from this treatment so that the range is always built on three DMS concentration values.

Overall, the striking observation that can be made is the lack of agreement between models, and between models and observational products. Table 3 shows that L11 stands out with the highest median (and mean) DMS concentration. With respect to the lowest global median DMS concentration (UKESM1-0-LL), that of L11 is over 60 % higher. The models generally agree on enhanced DMS concentration in the eastern equatorial Pacific. Most models also predict higher DMS concentration in coastal regions, with some exceptions: for instance, MIROC-ES2L shows little or no enhancement along the southern coasts of Australia and South America (south of the 30° S latitude). Conversely, this model predicts significant DMS

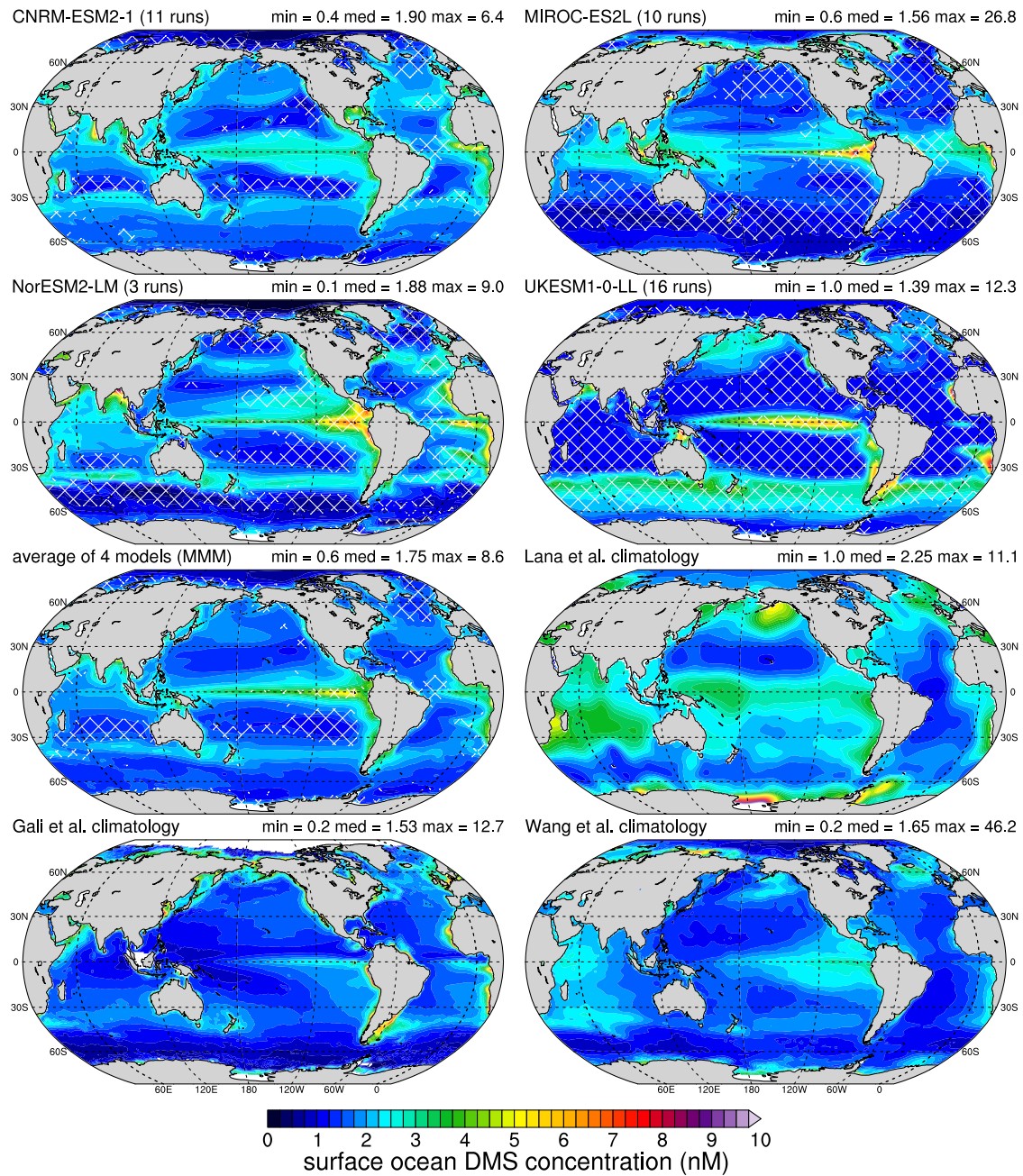

**Figure 1.** Mean (1980–2009) surface ocean DMS concentration (nM) for the CMIP6 historical experiments of the four models, CNRM-ESM2-1, MIROC-ES2L, NorESM2-LM, UKESM1-0-LL, and the MMM. Annual means of L11, G18 and W20 are also plotted. Minimum, area-weighted median and maximum from each model or data product are also displayed. Hatching on the model plots shows locations where the DMS concentration is outside the range covered by L11, G18 and W20 (see text for details).

**Table 3.** Area-weighted statistics, median, mean, and spatial standard deviation of annual DMS surface ocean concentration (nM) shown in Fig. 1. For the L11 climatology, means of the minimum and maximum values estimated by the authors are shown in brackets.

|  | Median | Mean $\pm\sigma$ |
|---|---|---|
| L11 | 2.25 | 2.35 $\pm$0.89 (2.02–3.38) |
| G18 | 1.53 | 1.69 $\pm$0.89 |
| W20 | 1.65 | 1.75 $\pm$0.72 |
| CNRM-ESM2-1 | 1.90 | 1.98 $\pm$0.66 |
| MIROC-ES2L | 1.56 | 1.77 $\pm$0.85 |
| NorESM2-LM | 1.88 | 1.98 $\pm$0.95 |
| UKESM1-0-LL | 1.39 | 1.78 $\pm$0.90 |
| MMM | 1.75 | 1.88 $\pm$0.59 |

enhancement along the coasts of the Arctic Ocean, as opposed to the other three models. MIROC-ES2L and NorESM2-LM, and to a smaller extent CNRM-ESM2-1, show a band of low DMS concentration between 40° S and 60° S, while UKESM1-0-LL predicts significantly enhanced DMS concentration in this latitude band. In MIROC-ES2L, the low concentration in that latitude band can likely be explained by the inverse relationship defined in the parameterisation of Aranami and Tsunogai (2004) between the DMS concentration and the MLD, which is deep in the Southern Ocean. This major discrepancy between models had already been highlighted by Tesdal et al. (2016), and seems even more pronounced in these CMIP6 results.

Individual model patterns shown in Fig. 1 have not changed significantly compared to previous publications presenting these models, or the parameterisations they are based on. For instance, the DMS concentration modelled by previous versions of PISCES has been shown in Belviso et al. (2012, Fig. 9f) and Masotti et al. (2016, Fig. 1f), and is also included in the review by Tesdal et al. (2016, Fig. 2). The current version of PISCES in CNRM-ESM2-1 shows very similar patterns, apart from the widespread elevated concentration around Antarctica that was present in the version used by Tesdal et al. (2016) but is significantly lower in the current version.

As expected, the output of MIROC-ES2L resembles those of the parameterisations of Simó and Dachs (2002) and Aranami and Tsunogai (2004), the latter being the closest (Tesdal et al., 2016, Fig. 2). As compared to the latter, the enhancement in the eastern equatorial Pacific is stronger in MIROC-ES2L. Interestingly, such strong enhancement was also observed when the parameterisation of Simó and Dachs (2002) was embedded in HadOCC (Tesdal et al., 2016, Fig. 2).

NorESM2-LM results can be compared to those obtained with the rather similar ocean biogeochemistry model HAMOCC, within the MPI-ESM-LR model, as presented by Tesdal et al. (2016). Both models present similar concentration enhancement off the coasts of western Africa, and in the equatorial Pacific, plus in a thin latitude band between 30° S and 40° S. However, there is little agreement at higher latitudes, especially in the northern Pacific and the Arctic, plus around Antarctica, where NorESM2-LM simulates smaller DMS concentration than HAMOCC.

UKESM1-0-LL features specific patterns that are distinct from the other three models, especially regarding the uniform low concentration over vast areas. This reflects the threshold set to 1 nM in the parameterisation based on that of Anderson et al. (2001) (see Sect. 2.1.1). As compared to the review of Tesdal et al. (2016), apart from the revised threshold value leading to lower DMS values, UKESM1-0-LL displays a stronger enhancement in the equatorial Pacific, that extends further west as compared to the other models. In that respect, the output of UKESM1-0-LL shares several common features with the HadOCC results shown in Tesdal et al. (2016). Lastly, as mentioned above, UKESM1-0-LL also stands out by the high DMS concentration in the Southern Ocean (40° S–60° S), which can be explained by the high bias in Chl in this region (Séférian et al., 2020) and the positive relationship between DMS and Chl in the parameterisation of Anderson et al. (2001). The patterns of DMS concentration are also very similar to the patterns of net primary production (NPP), which feature summer maximums in both hemispheres (Sellar et al., 2019, Fig. 19). This suggest a strong relationship between both variables in UKESM1-0-LL.

**Table 4.** Spatial correlation coefficients between L11, G18 and W20 and the models (first three rows), and between the individual models (rows 4–7), derived from the data displayed in Fig. 1. Last row: area fraction (%) within the range of L11, G18 and W20, i.e. fraction of oceans that is not hatched in Fig. 1. See text for details about the observational range definition.

| | CNRM-ESM2-1 | MIROC-ES2L | NorESM2-LM | UKESM1-0-LL | MMM |
|---|---|---|---|---|---|
| L11 | 0.26 | 0.17 | 0.24 | 0.08 | 0.25 |
| G18 | 0.38 | 0.30 | 0.49 | 0.29 | 0.53 |
| W20 | 0.39 | 0.40 | 0.46 | 0.13 | 0.49 |
| CNRM-ESM2-1 | 1 | 0.44 | 0.62 | 0.25 | – |
| MIROC-ES2L | – | 1 | 0.53 | −0.09 | – |
| NorESM2-LM | – | – | 1 | 0.20 | – |
| UKESM1-0-LL | – | – | – | 1 | – |
| Surf. area within Obs. range | 84 % | 59 % | 67 % | 37 % | 84 % |

The general findings outlined above are strengthened by the analysis of the Pearson spatial pattern correlation, which is presented in Table 4. When compared to L11, the annual pattern correlation ranges from 0.08 to 0.26 for individual models. When compared to G18 and W20, this pattern correlation is improved for all models, ranging from 0.13 to 0.46 for the latter. Note that because of the year-round undocumented area in the Arctic in G18 due to the lack of satellite observations, the G18 pattern correlation is computed on a slightly smaller area than the ones of the other two climatologies. However, computing the pattern correlation with L11 and W20 after masking them by the same extent as in G18 has a minor effect, with correlation coefficients reduced by only 0.01 or 0.02, which does not change the conclusions of this comparison. Regardless the observational product, UKESM1-0-LL has the lowest pattern correlation amongst the four models, which can likely be attributed to the constant minimum DMS concentration prescribed in the parameterisation of Anderson et al. (2001). It is also worth noting that whatever the climatology used to compare with, the MMM has the highest pattern correlation (or second highest when comparing with L11), even if the improvement as compared to individual models is small.

The cross correlation between models, presented in the second part of Table 4, is rather small as well, and even slightly negative between MIROC-ES2L and UKESM1-0-LL. Interestingly, the highest correlation (0.62) is between both prognostic models, CNRM-ESM2-1 and NorESM2-LM. Conversely, as in the comparison with the climatologies, UKESM1-0-LL has the lowest correlations with each of the other three models.

The last row in Table 4 shows the area fraction, for each model, that lies in the range of values of the three climatologies. Up to 84 % of the total area falls in this range for CNRM-ESM2-1, which can be related to the smoother patterns displayed by this model (Fig. 1). Conversely, the DMS concentration modelled in UKESM1-0-LL lies in the range of climatologies over only 37 % of the surface. Again, the constant minimum value seems to be responsible for this poorer agreement: as can be seen in Fig .1, most of the hatched regions of UKESM1-0-LL are those where the DMS concentration is assigned to the fixed value of 1 nM.

The difference in annual mean concentration between each models and the three climatologies is shown in Fig. 2. An alternative presentation using the logarithm of the ratio (model/clim) is shown in the Supplementary Information (Fig. SI-1) to highlight the places where the relative difference is important. Cross comparison between all models, with a single climatology taken as a reference, or conversely with a single model comparing the three climatologies, can emphasise some of the features that clearly stand out either in models or in the climatologies. For instance, the elevated DMS concentration for three models in the eastern equatorial Pacific or in the tropical Atlantic, and in the case of UKESM1-0-LL in the 40° S–60° S, are the modelled features that stand out, against all climatologies, thus suggesting that these are model biases. Conversely, the very high concentration displayed in L11 around Antarctica, and to a lesser extent in the south of Alaska and in the Indian Ocean, are neither predicted by any model nor by G18 or W20. For the former two regions, high concentrations have been reported in long-time measurements, at a site of the West Antarctic Peninsula, 2012–2017 period (Webb et al., 2019), and at the Ocean Station P in the North East Pacific, 1996-2010 period (Steiner et al., 2012) and 2005-2017 period (Galí et al., 2018). Further investigations would be required to explain these discrepancies between measurements and models or climatologies. Some specific processes, such as the DMS concentration enhancement following sea-ice break-up (Webb et al., 2019) are not accounted for in the models, but are not sufficient to explain all discrepancies. Overall, assessing the relevance of high DMS events at the global scale and the spatial resolution of climate models is mandatory to improve them.

The monthly mean (1980–2009) zonal DMS concentration is shown in Fig. 3 and monthly maps are shown for each model in the Supplementary Information (Figs. SI-2 and SI-3). Models and climatologies all show a clear seasonal cycle in the mid to high latitudes, with a summer maximum in both hemispheres. However, they disagree on the amplitude and timing of this cycle. These results compare well with the analog plot in Tesdal et al. (2016, Fig. 3) with some differences. CNRM-ESM2-1 shows a summer maximum in the northern hemisphere which is shifted southwards and is less intense as compared to the version of PISCES shown in Tesdal et al. (2016). In this version of PISCES, a pronounced year-round low DMS concentration centered on the 30° latitude bands of both hemispheres was also present, while this feature is much less clear in CNRM-ESM2-1.

The zonal mean seasonal cycle of MIROC-ES2L is very similar to that of both Simó and Dachs (2002) and Aranami and Tsunogai (2004). The summer maximum in the northern hemisphere occurs only in June and July in MIROC-ES2L, while it

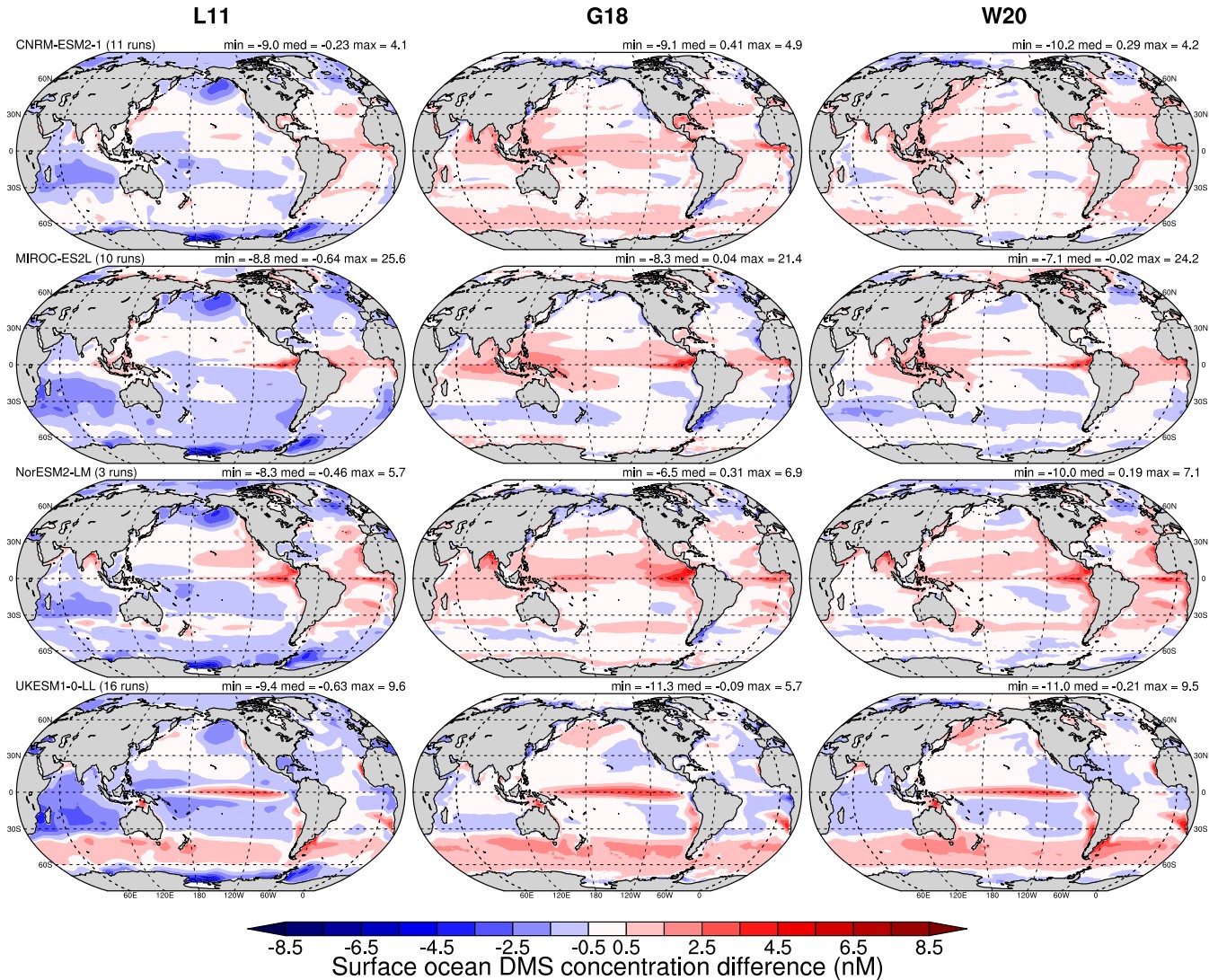

**Figure 2.** Mean annual differences of surface ocean DMS concentration (nM) between models and the climatologies of L11, G18 and W20 (model minus climatology). Model data as in Fig. 1. Minimum, weighted median and maximum of each panel are displayed.

lasts from May to September in the latter. A clear imprint of the equatorial Pacific concentration enhancement is also visible in
MIOC-ES2L from January to May.

NorESM2-LM shows a noticeable double maximum in both hemispheres, around 40° and at latitudes higher than 60°. In the northern hemisphere, the maximum around 40° N is mostly driven by high concentration in the Atlantic from May to August, and to a lesser extent in the Pacific in June and July (see Fig. SI-3 in the Supplement). In the Southern Ocean, a pronounced DMS maximum around 40° S from November to February results in the zonal maximum observed in Fig. 3.

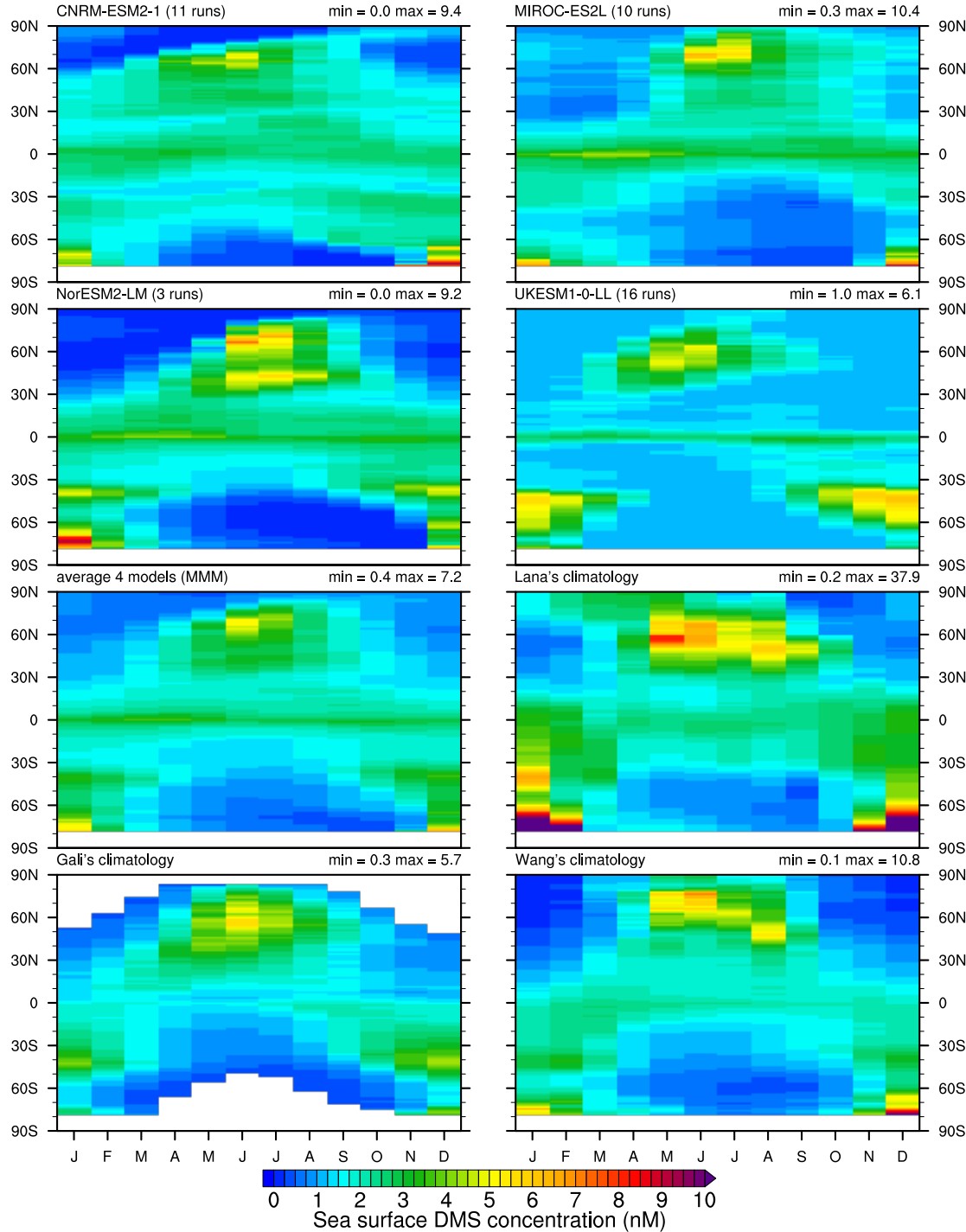

**Figure 3.** Monthly mean (1980–2009) zonal surface ocean DMS concentration (nM) for the same models and observations as in Fig. 1.

UKESM1-0-LL shows a northern maximum starting earlier than the other models, in agreement with L11. However, as pointed out by Hayashida et al. (2020), this feature could be an extrapolation artefact in L11 (see Sect. 2.2.1). Fig. 3 also shows that the already noticed feature of high DMS concentration in a 40° S to 60° S latitude band is mostly an austral summer feature, as for the NPP (Sellar et al., 2019, Fig. 19). The fixed minimum DMS concentration of 1 nM is also striking on this figure, with large zonal bands with uniform DMS concentration.

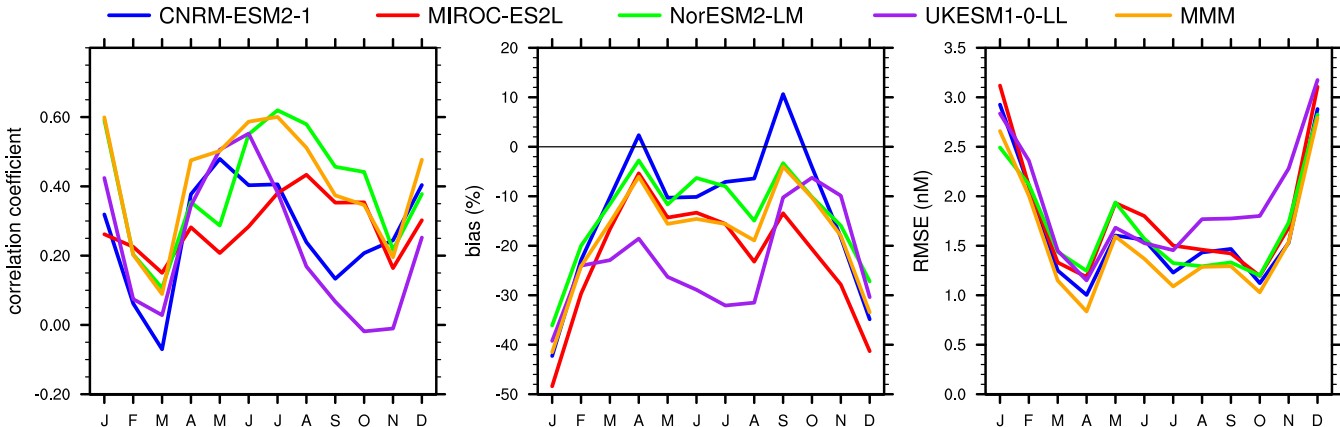

**Figure 4.** Monthly mean statistics of surface ocean DMS concentration. Left: spatial correlations between a model and L11, middle: biases (in percentage of L11), and right: RMSE (nM). A 0 % bias line is drawn for visual aid. Model diagnostics as in Fig. 1. The values are provided in SI Table SI-1. The same metrics computed against G18 and W20 are plotted in Fig. SI-4.

We further extended the pattern correlation analysis presented above for annual mean to monthly data. Figure 4 presents three spatial monthly statistics between the models and L11, namely correlation coefficients, biases (in % of the reference) and root mean square errors (RMSE). The same statistics using G18 and W20 as reference are also provided in SI Figs. SI-4. All models outputs have been gridded to the 1° grid of the reference to compute these statistics. All models show similar seasonal variations across all three statistics. Biases (negative except for two months of CNRM-ESM2-1) and RMSE are significantly 355 higher during the austral summer (November to February), probably due to the high DMS concentration featured in L11 around Antarctica, that no model agrees with. The same explanation holds for northern hemisphere summer months (May to September), where the bias and RMSE are slightly larger than during the other months of the year, because the summer maximum of DMS concentration in all models is less intense and has a smaller spatial extent (Fig.3).

Pattern correlations have low values (lower than 0.2) in February-March and again in September to November, which can 360 be related to the smoother features and inter-hemisphere DMS gradients during these months. Conversely, during the summer months in both hemispheres, even if the models do not reach such elevated DMS values as in L11, the stronger concentration gradients contribute to a higher correlation. When compared to G18 and W20 (Fig. SI-4), the pattern correlations show smoother seasonal variations.

Overall, the MMM has the lowest RMSE in nearly all months, with significantly lower values than each model when 365 compared to G18 and W20 (Fig. SI-4). The MMM and NorESM2-LM also have the best correlation coefficients with L11 in

almost all months, with coefficients higher than 0.4 during seven months of the year. They also have the best pattern correlations when compared to G18 and W20.

### 3.1.2 Seasonal variation analysis

Further to the spatial analysis presented in the previous section, we now focus on the analysis of seasonal variations. For that
purpose, we used the same ocean partitioning into 54 biogeographical provinces (Longhurst, 2007) that Lana et al. (2011) used to built the L11 climatology (see the description in Sect. 2.2.1 and Appendix A2). These 54 provinces are also grouped into six biomes (polar N and S, westerlies N and S, trades and coastal), to bring further information regarding the models' strengths and weaknesses.

In Fig. 5 we present the seasonal variations, in each of the 54 ocean provinces, of the 4 models and MMM, along with L11,
with its min-max range as a shaded envelope. This figure is inspired from that in Lana et al. (2011, Fig. 2). However, it is important to stress that the latter depicts the "first guess" DMS concentration, which is an intermediate stage of the climatology calculation process (see Sect. 2.2.1 for details). Regarding individual model characteristics, one should note that in MIROC-ES2L, the Red Sea and the Persian Gulf are treated as land, thus this model has no data in region 25 (code REDS). Also, in some regions, UKESM1-0-LL computes constant DMS concentration, equal to its lower bound of 1 nM: see for instance regions 9,
14, 16, 17, 25, 38. For each province, the resulting correlation of modelled versus L11 seasonal variations is computed, and displayed in Table 5.

This figure shows the skills of each model in reproducing the seasonal cycle for each region. The seasonal cycle in the northern Atlantic (regions 2–6 and 11) is relatively well reproduced regarding the timing and duration of seasonal maximum, but the amplitude is generally lower in the models. The agreement is also satisfying in the northern Pacific (regions 30, 32, 33,
44) and in the regions located south of $\sim 40°$ S (regions 51–54). Equatorial regions are characterised by a weak seasonality and low DMS concentrations, and models poorly reproduce the climatology in these provinces (regions 9, 17 and 41 for instance).

In Table 5, the regions are sorted according to the six biomes, which helps at identifying general model behaviour. It is noteworthy that the seasonal cycle in most polar and westerly provinces of both hemispheres is well captured by all models. Conversely, all models poorly reproduce the seasonal cycle in most trade wind regions. Regarding coastal regions, some are
rather well reproduced regarding their seasonal cycles, while others not. There is a relatively good consistency between models: in coastal regions where the seasonal cycle is well (respectively poorly) reproduced, this is generally similar for all (or all but one) models. When looking in detail at the coastal regions, it is clear that model skills are better in coastal regions located in high latitudes (for instance region 11, 15, 20, 43, 44, 50) than in those located at low latitudes, which is consistent with the conclusions regarding the other biomes.

Among the four models, none appears to have significantly better skills than the other regarding the seasonal cycle. The MMM presents slightly more uniform skills throughout the 54 provinces, with a correlation generally within the range of individual model correlations (35 out of 54 regions), or better than all models (17 out of 54 regions). There are only two exceptions (regions 8 and 17) where the correlation of MMM is worse than all individual models.

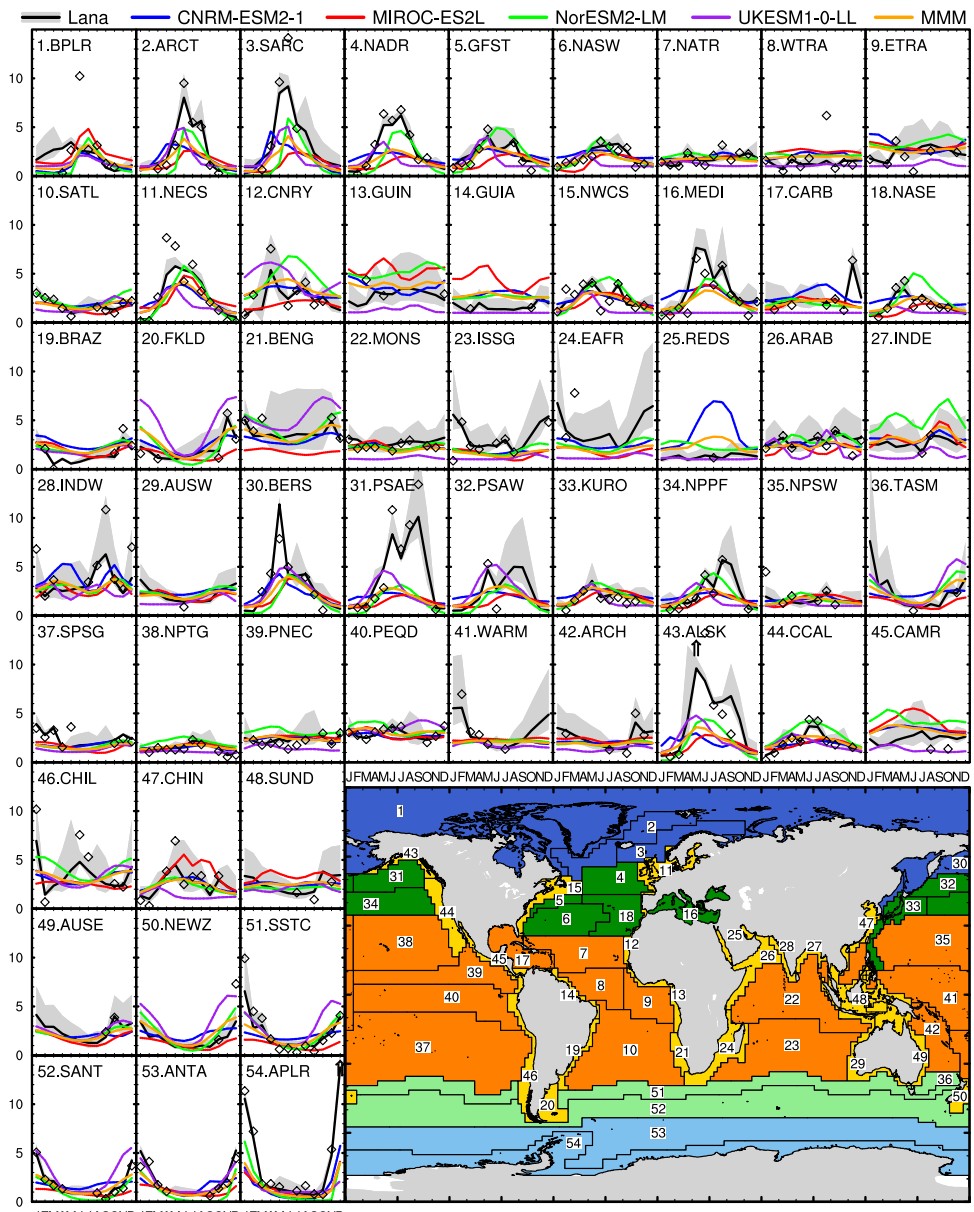

**Figure 5.** Monthly 1980–2009 area-weighted surface ocean DMS concentration (nM) in the 54 oceanic regions defined by Longhurst (2007), for the models as in Fig. 1 (color lines) and L11 (black line). Also plotted are other data of L11: (1) minimum and maximum as light grey envelopes; and (2) "Pixel" data, i.e. raw data binned on the 1°×1° grid, as diamonds (note that they are not identical to the data plotted in Fig. 2 of Lana et al. (2011), see text for details). The oceanic region numbering (54 regions) is repeated on the lower right corner map, together with colours according to Longhurst (2007) for the six biomes (dark and light blue: polar N and S, dark and light green: westerlies N and S, orange: trades, yellow: coastal).

| Region | CNRM-ESM2-1 | MIROC-ES2L | NorESM2-LM | UKESM1-0-LL | MMM |
|---|---|---|---|---|---|
| Polar Arctic 01 | 0.09 | 0.18 | 0.10 | 0.43 | 0.17 |
| Polar Atlantic 02 | 0.67 | 0.77 | 0.88 | 0.75 | 0.94 |
| Polar Atlantic 03 | 0.66 | 0.54 | 0.78 | 0.93 | 0.92 |
| Polar Pacific 30 | 0.85 | 0.50 | 0.56 | 0.81 | 0.75 |
| Westerly Atlantic 04 | 0.75 | 0.65 | 0.88 | 0.55 | 0.96 |
| Westerly Atlantic 05 | 0.77 | 0.38 | 0.78 | 0.63 | 0.86 |
| Westerly Atlantic 06 | 0.75 | 0.79 | 0.82 | -0.01 | 0.88 |
| Westerly Atlantic 16 | 0.69 | 0.91 | 0.84 | -0.40 | 0.88 |
| Westerly Atlantic 18 | 0.74 | 0.34 | 0.74 | 0.46 | 0.80 |
| Westerly Pacific 31 | 0.42 | 0.90 | 0.82 | 0.37 | 0.70 |
| Westerly Pacific 32 | 0.50 | 0.74 | 0.90 | 0.66 | 0.82 |
| Westerly Pacific 33 | 0.79 | 0.62 | 0.82 | 0.69 | 0.85 |
| Westerly Pacific 34 | 0.42 | 0.85 | 0.74 | -0.03 | 0.62 |
| Trade wind Atlantic 07 | 0.02 | 0.58 | -0.19 | -0.44 | 0.05 |
| Trade wind Atlantic 08 | -0.25 | -0.25 | -0.20 | -0.05 | -0.34 |
| Trade wind Atlantic 09 | -0.89 | -0.83 | 0.50 | 0.72 | -0.58 |
| Trade wind Atlantic 10 | 0.54 | 0.67 | 0.52 | -0.19 | 0.45 |
| Trade wind Atlantic 17 | -0.32 | -0.20 | -0.33 | -0.27 | -0.34 |
| Trade wind Pacific 35 | 0.50 | 0.10 | -0.41 | -0.46 | -0.11 |
| Trade wind Pacific 37 | 0.14 | 0.64 | 0.39 | 0.36 | 0.60 |
| Trade wind Pacific 38 | 0.94 | 0.70 | 0.69 | 0.09 | 0.86 |
| Trade wind Pacific 39 | -0.04 | 0.01 | -0.54 | -0.54 | -0.44 |
| Trade wind Pacific 40 | -0.41 | -0.33 | -0.11 | -0.55 | -0.54 |
| Trade wind Pacific 41 | -0.82 | -0.19 | -0.81 | 0.12 | -0.63 |
| Trade wind Pacific 42 | -0.43 | 0.76 | 0.57 | 0.34 | 0.66 |
| Trade wind Indian 22 | -0.41 | -0.31 | 0.17 | 0.20 | -0.18 |
| Trade wind Indian 23 | 0.70 | 0.69 | 0.66 | 0.14 | 0.80 |
| Westerly Antarctic 36 | 0.09 | 0.75 | 0.59 | 0.52 | 0.58 |
| Westerly Antarctic 51 | 0.67 | 0.74 | 0.87 | 0.62 | 0.79 |
| Westerly Antarctic 52 | 0.49 | 0.72 | 0.91 | 0.83 | 0.87 |
| Polar Antarctic 53 | 0.68 | 0.78 | 0.96 | 0.99 | 0.99 |
| Polar Antarctic 54 | 0.97 | 0.95 | 0.87 | 0.85 | 0.97 |
| Coastal Atlantic 11 | 0.94 | 0.82 | 0.94 | 0.77 | 0.97 |
| Coastal Atlantic 12 | 0.57 | 0.48 | 0.63 | 0.13 | 0.63 |
| Coastal Atlantic 13 | -0.89 | -0.05 | 0.36 | 0.80 | -0.17 |
| Coastal Atlantic 14 | -0.70 | 0.17 | -0.33 | -0.34 | -0.18 |
| Coastal Atlantic 15 | 0.85 | 0.65 | 0.77 | 0.64 | 0.87 |
| Coastal Atlantic 19 | 0.60 | 0.41 | 0.75 | 0.70 | 0.73 |
| Coastal Atlantic 20 | 0.75 | 0.30 | 0.65 | 0.66 | 0.66 |
| Coastal Atlantic 21 | 0.45 | 0.05 | 0.42 | 0.33 | 0.44 |
| Coastal Pacific 43 | 0.43 | 0.89 | 0.93 | 0.71 | 0.91 |
| Coastal Pacific 44 | 0.82 | 0.65 | 0.89 | 0.54 | 0.89 |
| Coastal Pacific 45 | -0.01 | 0.34 | -0.72 | -0.75 | -0.22 |
| Coastal Pacific 46 | -0.08 | 0.17 | 0.01 | -0.14 | -0.06 |
| Coastal Pacific 47 | 0.65 | 0.56 | 0.70 | -0.22 | 0.66 |
| Coastal Pacific 48 | 0.48 | -0.28 | 0.20 | -0.46 | -0.33 |
| Coastal Pacific 49 | 0.50 | 0.86 | 0.76 | 0.80 | 0.86 |
| Coastal Pacific 50 | 0.64 | 0.83 | 0.94 | 0.79 | 0.88 |
| Coastal Indian 24 | 0.73 | 0.56 | 0.70 | 0.29 | 0.76 |
| Coastal Indian 25 | 0.13 | 0.56 | -0.50 | -0.25 | 0.03 |
| Coastal Indian 26 | -0.33 | -0.03 | 0.15 | 0.21 | 0.21 |
| Coastal Indian 27 | 0.62 | 0.53 | 0.47 | 0.49 | 0.60 |
| Coastal Indian 28 | -0.18 | -0.11 | -0.08 | 0.21 | -0.10 |
| Coastal Indian 29 | 0.40 | 0.59 | 0.66 | -0.03 | 0.53 |

**Table 5.** Correlation coefficients of the linear regressions between the monthly time series of the models and L11. Time series and regions, with their numbering and color code, are those of Fig. 5. Purple colours for negative correlations, green colours for positive correlations.

Several hypotheses can be proposed to explain the poorer correlation of the modelled seasonal cycles with the observations at low latitudes. First, it was already shown that models present some features at low latitudes that do not agree with climatologies. For instance, the strong enhancement of DMS concentration in the eastern equatorial Pacific found in all models except CNRM-

ESM2-1. While this model feature leaves an imprint in the annual mean (see Fig. 1), the monthly analysis shows that it is mostly a spring (February–May) phenomenon (see Fig. 3). Indeed, when looking at the corresponding regions 39 and 40 (to a lesser extent, the Atlantic region 7), it is clear that this typical model behaviour can explain the poor temporal correlation. The second argument to explain the low correlation in equatorial/subtropical latitudes, is that the seasonality is very weak around the Equator (Fig .3). As noted by previous authors (see for instance Lana et al., 2012, Fig. 4), this weak seasonality can be responsible for the low or even negative correlation, since the model uncertainties may have a larger amplitude than the seasonal variability. The third explanation is that uncertainties also originate from the climatology itself. To investigate this effect, we performed the same province-based seasonality analysis using W20. The corresponding table built with the same presentation as Table 5 is shown in the SI (Table SI-5). While trades regions still show lower correlations than higher latitudes provinces (polar and westerlies), the overall agreement is significantly better than with L11, which suggests that the disagreement visible in Table 5 can be partly attributed to L11 uncertainties.

## 3.2 Marine DMS emissions

### 3.2.1 Detailed insight into the sea-air flux parameterisations

The choice of the gas transfer parameterisation has an important impact on the calculated marine DMS emission, we thus briefly recall here their main characteristics. To compute the transfer of DMS from the ocean into the atmosphere, UKESM1-0-LL uses the parameterisation of Liss and Merlivat (1986) which distinguishes between three wind regimes. For each of them, the parameterisation is a linear function of surface wind. The resulting broken stick parameterisation has an overall dependence on the whole wind speed range which is thus intermediate between a linear and a quadratic relation. MIROC-ES2L uses the widely used parameterisation of Nightingale et al. (2000), that is a second-order polynomial function of the surface wind speed. It has been shown in a number of studies to result in intermediate estimations of the emissions, as compared to other widely used parameterisations (e.g. Lana et al., 2011; Tesdal et al., 2016). The remaining two models (CNRM-ESM2-1 and NorESM2-LM) use the DMS flux parameterisation of Wanninkhof (2014) where the flux is proportional to the squared wind speed, and leads to gas exchange values similar to those of Nightingale et al. (2000) at intermediate wind speed range. This parameterisation is a revision of the work of Wanninkhof (1992) which leads to significantly lower exchange coefficients than in the original publication. Wanninkhof (2014) now also includes a formulation of the Schmidt number for DMS as a function of temperature, with a fourth-order polynomial fit valid from $-2$ °C to 40 °C. A bias in modelled SST can thus contribute to the bias in flux calculation, but is estimated to be smaller than the uncertainty of flux parameterisation. The reader is referred to Figures 12 of Nightingale et al. (2000) and Figure 2 of Wanninkhof (2014) to illustrate the relationships between wind speed and exchange coefficients for the three gas transfer parameterisations of interest here.

### 3.2.2 Annual mean emission

DMS flux into the atmosphere is presented in Fig. 6 as 2D annual mean fields, in Fig. 7 as yearly emissions per 10° latitude bands, and in Fig. 8 as zonal monthly means. Model fields represent values averaged over the 1980–2009 period. The reference dataset added in the analysis here is the CAMS19 climatology described in subsection 2.2.2.

The flux of DMS features spatial patterns and seasonal cycles that stem from both the surface ocean concentration and the wind speed. To help understand how these main drivers impact the resulting flux, the maps of annual wind speed and the zonal monthly wind speed are shown in the Supplementary (Fig. SI-6).

Overall, the maps of annual mean flux show a large spatial variation, which mirror in part the patterns of annual mean concentration (see Fig. 1), with clear imprints from annual wind patterns (see Fig. SI-6). For instance, higher westerly winds 440 and weaker trade winds have a visible impact on the annual flux, both in models and climatology. For UKESM-1-0-LL, the weak trade winds combined with the very low DMS concentration over large parts of the 30° N–30° S band (see Fig. 1) result in a flux lower than those of the other models. CNRM-ESM2-1 and UKESM1-0-LL display higher flux in the Southern Ocean than the other models, but the underlying mechanism is different: while this feature clearly mirrors the pattern of concentration in UKESM1-0-LL (see Fig. 1), it stems from the combination of year-round sustained high wind and moderate 445 DMS concentration in CNRM-ESM2-1. Relative maxima in CAMS19 over the Indian Ocean and the northeastern Pacific reflect the high DMS concentration of the L11 climatology, from which CAMS19 fluxes are derived. These high concentration features in L11 are also reinforced by high wind speed in these regions. The resulting emission patterns in CAMS19 are not present in the models. Despite the variety of simulated patterns, a number of common features can also be seen, such as the low emissions at high latitudes in both hemispheres. This agreement can be explained by the common simulation framework 450 shared by all models and CAMS19 (see Sect. 2.2.2), which linearly scales DMS emission on the non sea-ice covered fraction. Other common features include high emission in several coastal zones, such as the coast of Mauritania, Namibia and south of the Arabian peninsula. All models also simulate low annual emission over Southeast Asia, which is another clear imprint of the low surface wind speed. This feature is especially pronounced in MIROC-ES2L.

The variety of patterns shown in Fig. 6 results in an important spread when looking at the annual emission per latitude band, 455 as shown in Fig. 7. Two models, CNRM-ESM2-1 and UKESM1-0-LL, have their largest fluxes in the 40° S–50° S latitude band, in line with the previous observation. Conversely, the zonal profile of annual emission peaks in the 10° N – 20° N band for NorESM2-LM, which results from the combination of elevated flux in subtropical Pacific and in several coastal regions located in the same latitude band (Fig. 6). In turn, the explanation of these elevated fluxes can be found in both surface concentration of DMS, and rather high surface wind. The zonal profile of annual emission in MIROC-ES2L is flatter than in the other models, 460 with similar emissions across the 30° N – 50° S band. This is in agreement with the map of annual flux which shows a smoother global emission as compared to other models, and will be further explained in the next subsection. Lastly, Fig. 7 also confirms that the DMS emission in UKESM1-0-LL across the 30° N – 30° S band is significantly lower than the other models, but also lower than all the other estimates based on climatologies. This results in global annual DMS emissions lower than those of the other models (see also Table 6).

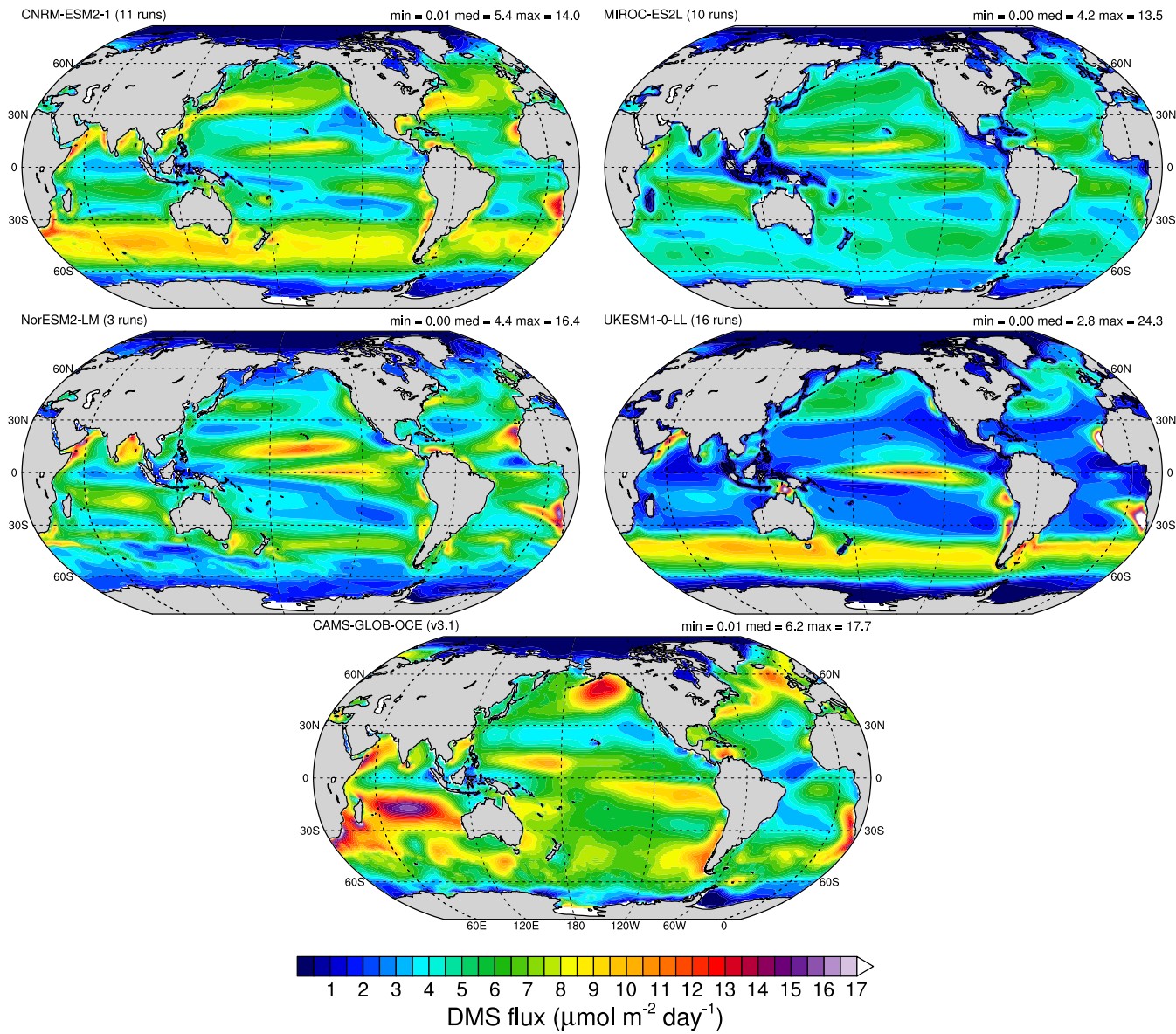

**Figure 6.** Annual mean DMS flux ($\mu$mol m$^{-2}$ day$^{-1}$). First two rows: 1980–2009 means for the four CMIP6 models. Last row: CAMS19. Min, area-weighted median and maximum are provided for each panel.

### 3.2.3 Seasonal cycle analysis

Figure 8 shows the seasonal cycle of the DMS flux. First, a year-round low emission in a thin latitude band around the Equator is featured by all models and CAMS19. This is clearly related to the low wind speed in the ITCZ (Inter Tropical Convergence Zone), which is slightly more pronounced in CNRM-ESM2-1 and MIROC-ES2L (see Fig. SI-6). This figure also shows that

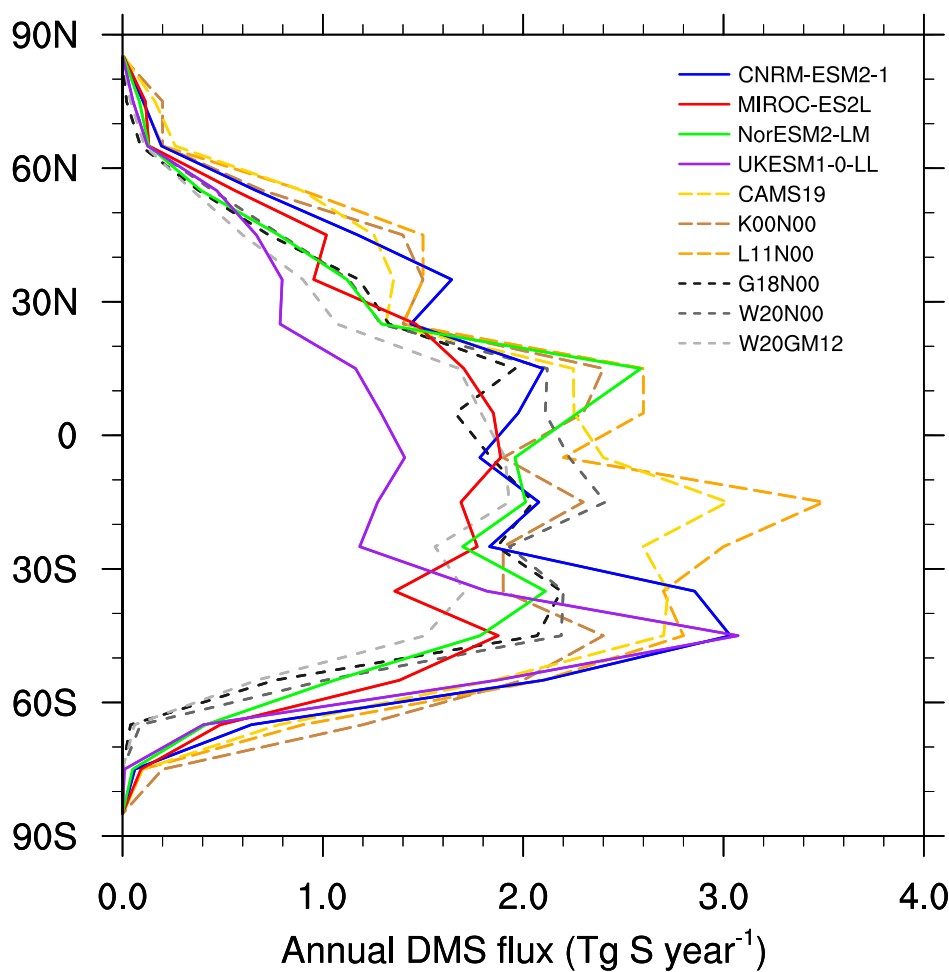

**Figure 7.** Mean annual DMS flux $(\text{Tg S year}^{-1})$ per 10° latitudinal bands for the models and CAMS19 as in Fig. 6. Other references are shown applying the Nightingale et al. (2000) parameterisation of air-sea fluxes and different DMS concentration: K00: Kettle and Andreae (2000) and L11 (data from Table 2 of Lana et al., 2011) - G18N00: G18 - W20N00: W20. W20GM12 is W20 with the parameterisation of air-sea fluxes of Goddijn-Murphy et al. (2012). G18N00, W20N00 and W20GM12: data from Table 2 of Wang et al. (2020).

the summer maximum in both hemispheres that was observed for the DMS concentration (Fig. 3) is preserved in the emission
of NorESM2-LM, UKESM1-0-LL and CAMS19, while it is completely smoothed in MIROC-ES2L, with a weak seasonal
cycle in both hemispheres. In the northern hemisphere for CNRM-ESM2-1, the seasonal cycle of the emission is even reversed
as compared to that of the concentration. A detailed side comparison of the same plots of DMS concentration and surface
wind (Figs. 3 and SI-6) explains the nearly opposite behaviours of models. For instance, focusing on the latitude band between
30° N and 50° N, we see that CNRM-ESM2-1 and MIROC-ES2L compute significantly higher surface wind speed during
winter months than the other two models. Apart from that of Liss and Merlivat (1986), this difference is further amplified
by the quadratic dependence of wind speed in the flux parameterisations. In this latitude band and during winter months,

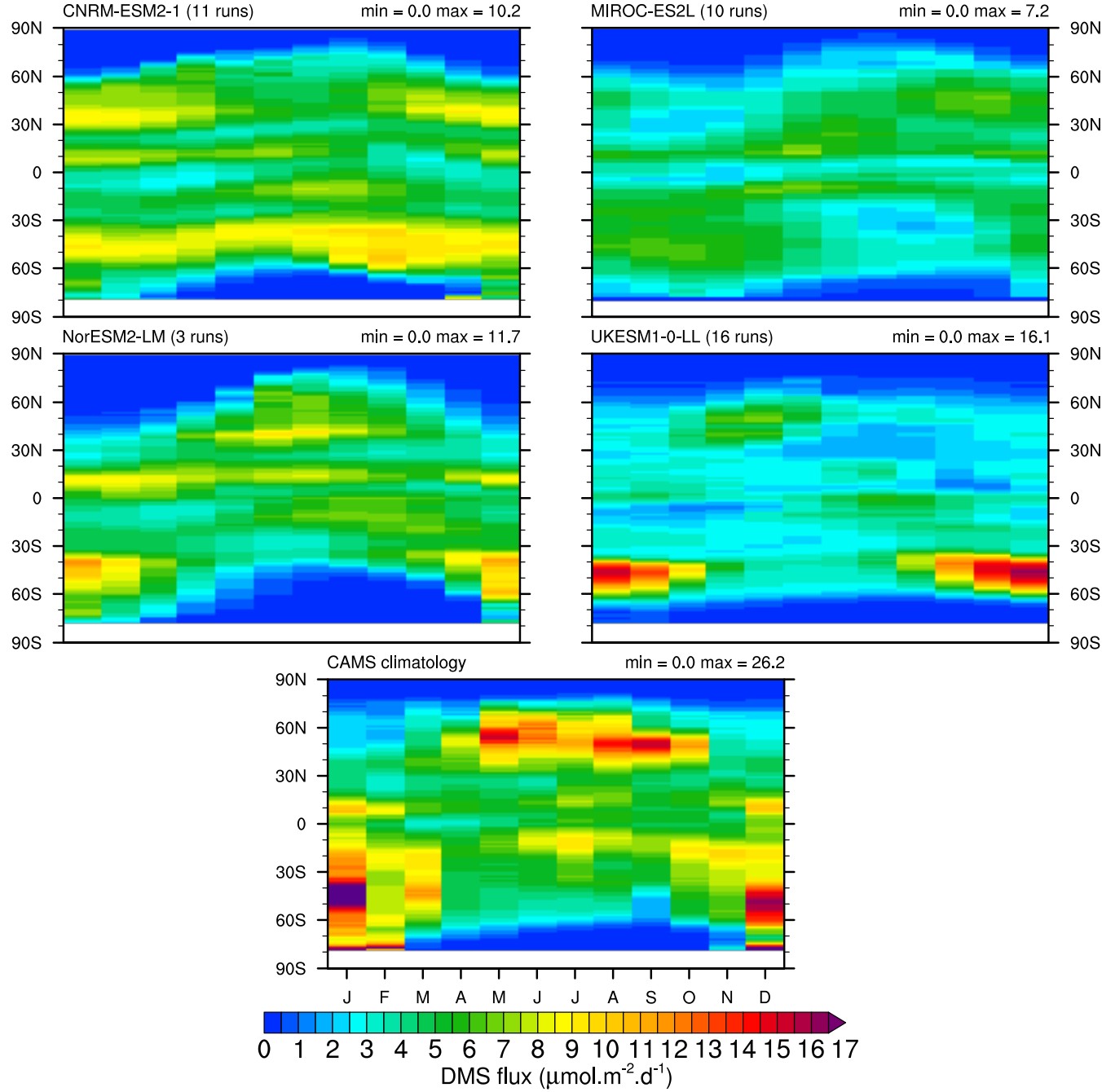

**Figure 8.** Monthly mean zonal DMS flux ($\mu\mathrm{mol\,m^{-2}\,day^{-1}}$) for the same models and datasets as in Fig. 6.

CNRM-ESM2-1 also predicts DMS concentration (around 2 nM) which is nearly twice as much as in the other models. The combination of this relatively high concentration and high wind speed thus results in the highest emission in winter for CNRM-

**Table 6.** Global mean DMS emissions ($\mathrm{Tg\,S\,year^{-1}}$) of the four CMIP6 models over different 30-year periods. Emission totals of current estimations available in literature, displayed in 10° latitudinal bands in Fig. 7, are also given. [a]: the range results from various wind and SST products used in the flux calculation. Data from Table 4 of Kettle and Andreae (2000). [b]: the estimated range depends on the assigned bias in DMS concentration as derived from remote-sensing measurements (Galí et al., 2018); [c]: the flux estimated by Galí et al. (2018) is simply linearly scaled to the concentration, following the multimodel review of Tesdal et al. (2016) which shows a roughly linear relation between global annual DMS flux and DMS concentration; [d]: Tesdal et al. (2016) provided a best estimate of $18 - 24\ \mathrm{Tg\,S\,year^{-1}}$ using available observations and the previous generation of ESMs.

| | 1850 − 1879 | 1980 − 2009 | 2071 − 2100 | Air-Sea flux Ref. |
|---|---|---|---|---|
| CMIP6 model estimates | | | | |
| CNRM-ESM2-1 | 23.4 | 23.7 | 25.4 | Wanninkhof (2014) |
| MIROC-ES2L | 18.1 | 18.4 | 20.4 | Nightingale et al. (2000) |
| NorESM2-LM | 19.7 | 19.8 | 19.0 | Wanninkhof (2014) |
| UKESM1-0-LL | 16.3 | 16.4 | 16.0 | Liss and Merlivat (1986) |
| Observational estimates | | | | |
| Kettle and Andreae (2000)[a] | | 12.9 − 16.0 | | Liss and Merlivat (1986) |
| Lana et al. (2011) | | 17.6 | | Liss and Merlivat (1986) |
| Lana et al. (2011) | | 28.1 | | Nightingale et al. (2000) |
| Galí et al. (2018) | | 16.0 − 20.0[b] | | Tesdal et al. (2016, Fig. 8)[c] |
| CAMS19 | | 25.9 | | Nightingale et al. (2000) |
| Wang et al. (2020) | | 17.9 | | Nightingale et al. (2000) |
| Wang et al. (2020) | | 17.2 | | Goddijn-Murphy et al. (2012) |
| Previous assessed range | | | | |
| Tesdal et al. (2016) | | 8.8 − 27.3[d] | | Liss and Merlivat (1986), Nightingale et al. (2000), |

ESM2-1. Conversely, the very low concentration and moderate wind speed in NorESM2-LM and UKESM1-0-LL lead to low emission in winter. MIROC-ES2L represents the intermediate situation, where the amplitude and phase of the seasonal cycles of DMS concentration and wind counteract in similar proportions, leading to a weak seasonal cycle of the emission.

### 3.2.4 Comparison with other studies

To conclude this section, Table 6 summarises global mean emissions of the four CMIP6 models along with several estimates from other studies for the modern period. Figure 9 also provides a convenient way to compare models in a glance regarding their annual global mean concentration, and emission. This Table does not include previous estimates of the flux calculated with the parameterisation of Wanninkhof (1992) since it has been revisited by the author, and the updated parameterisation of Wanninkhof (2014) leads to significantly lower exchange coefficients (Sect. 3.2.1).

Because of the weakest dependence on wind speed in the parameterisation of Liss and Merlivat (1986), UKESM1-0-LL has the lowest mean emission of $16.4\,\mathrm{Tg\,S\,year^{-1}}$ over the 1980–2009 period. This value agrees with other studies using the same flux parameterisation. MIROC-ES2L has an intermediate emission value of $18.4\,\mathrm{Tg\,S\,year^{-1}}$ which ranges in the low end of other studies also using the flux parameterisation of Nightingale et al. (2000). This is consistent with the rather low global mean DMS concentration in MIROC-ES2L (1.77 nM, Table 3), and the finding of Tesdal et al. (2016, Fig. 8) that to first order the global mean concentration of DMS determines the global mean flux.

The case of CNRM-ESM2-1 and NorESM2-LM is interesting as these models have very close global mean DMS concentration, yet NorESM2-LM computes a total annual DMS flux about 20 % lower than CNRM-ESM2-1 (see also Fig. 9). Both models use the sea-air flux parameterisation of Wanninkhof (2014), however their surface wind speeds differ. As shown in Fig. SI-6, the annual median wind speed is $\sim 6.5$ % higher in CNRM-ESM2-1, leading to a $\sim 13$ % difference when squared, and their seasonal cycles differ as well. Tesdal et al. (2016, Fig. 8) also illustrate that the emission may differ by as much as $1.5$–$3\,\mathrm{Tg\,S\,year^{-1}}$ when two different fields of DMS concentration with identical annual mean concentration are chosen, all other things being equal. Thus, these two arguments together explain the 20 % difference in DMS emission between both models.

To conclude, the global DMS emission computed by the four CMIP6 models are well within the recent literature values, the best estimate of Tesdal et al. (2016) being of $18$–$24\,\mathrm{Tg\,S\,year^{-1}}$. It should be emphasised that the choice of the exchange coefficient parameterisation has a strong influence on the resulting DMS emission, and the uncertainties associated with these parameterisations are elevated (Tesdal et al., 2016; Goddijn-Murphy et al., 2012). Accounting only on the four models, the median flux (1980–2009) is $19\pm3\,\mathrm{Tg\,S\,year^{-1}}$, thus lowering the best estimate of Tesdal et al. (2016) by 10 % but with an identical uncertainty range. Overall, we somewhat contradict with Tesdal et al. (2016) who conclude in a low bias of model DMS fluxes. Indeed, apart from the CAMS19 and the L11/Nightingale et al. (2000) DMS flux estimates, the other current observational estimates coincide with our CMIP6 model estimate. However, the current grid size of ocean models and additional processes that are not accounted for, such as DMS enhancement during sea-ice break-up (see Sect. 3.1.1), prevent models from accounting for high DMS events localised in space and time. Thus, simulated DMS emission might represent the lower bound of actual fluxes.

## 4 Historical and future evolution

### 4.1 Global and regional trends of DMS concentration and flux

Figure 9 shows the evolution of global annual mean DMS concentration, and global total DMS flux over the entire historical and ssp585 period (1850-2100). Both variables are relatively stable over the historical period, with the onset of significant trends between 1970 and 1980. At the end of the century, the spread of these two quantities is larger than at the end of the historical period (more than twice as large for the concentration). The interannual variability of the ensemble mean is larger for NorESM2-LM due to its lower ensemble size (see Table 2). The ensemble spread increases with the number of realisations.

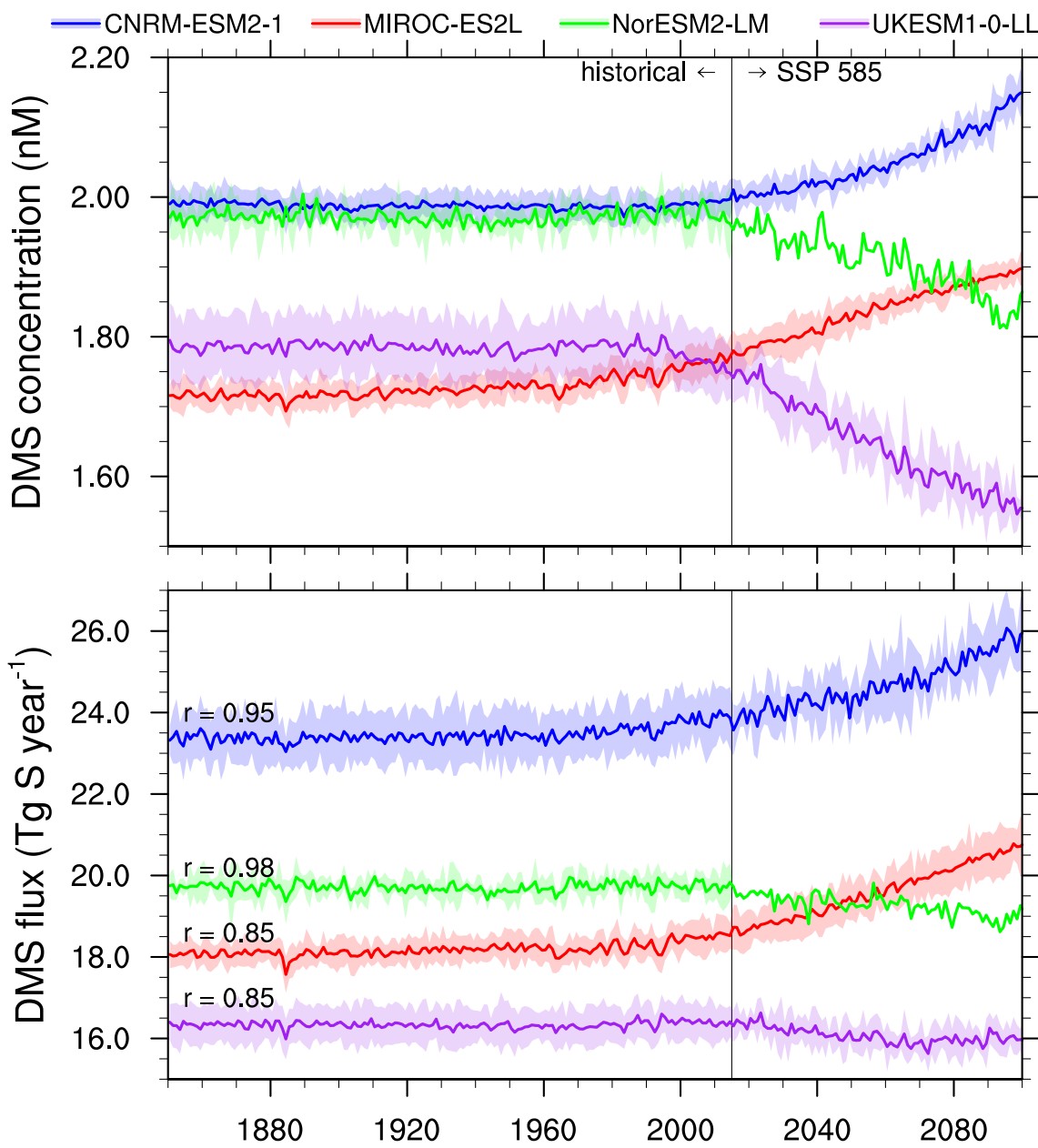

**Figure 9.** Time series of mean annual global area-weighted surface ocean DMS concentration (nM, top panel) and annual global DMS flux ($\mathrm{Tg\,S\,year^{-1}}$, bottom panel) over 1850 – 2100 (CMIP6 historical and ssp585 simulations). The ensemble mean of each model (thick lines), ± 2 standard deviations (shaded envelopes) is plotted. Note that the dimension of each ensemble may be different in the historical and in the ssp585 experiments (see Table 2). The Pearson correlation coefficient between both time series is indicated for each model on the bottom plot.

Each model also has distinct variability, which is especially noticeable in the case of CNRM-ESM2-1, whose spread in the DMS flux is significantly larger than that for UKESM1-0-LL although the latter has more realisations.

**Table 7.** Comparison of the mean and relative difference of DMS concentration (conc, in nM) and flux ($Tg\,S\,year^{-1}$), between the beginning of historical period (1850–1865) and the end of ssp585 period (2085–2100) as shown in Fig. 9.

|  |  | Mean 1850–1865 | Mean 2085–2100 | Relative difference (%) |
|---|---|---|---|---|
| CNRM-ESM2-1 | conc | 1.99 | 2.12 | 6.5 |
|  | flux | 23.4 | 25.7 | 9.8 |
| MIROC-ES2L | conc | 1.72 | 1.89 | 10.0 |
|  | flux | 18.1 | 20.6 | 13.8 |
| NorESM2-LM | conc | 1.97 | 1.84 | −6.6 |
|  | flux | 19.7 | 19.0 | −4.0 |
| UKESM1-0-LL | conc | 1.79 | 1.57 | −12.3 |
|  | flux | 16.3 | 16.0 | −2.1 |

Figure 9 reveals two major points. First, models disagree on the sign of the future trend: two models (CNRM-ESM2-1 and MIROC-ES2L) show positive trends over 2015–2100, while the other two models (NorESM2-LM and UKESM1-0-LL) simulate negative trends. Interestingly, both responses include one prognostic model and one empirical parameterisation of the

525 DMS, thus the adopted parameterisation does not explain this disagreement. However, besides this disagreement on the sign, all models show a strong correlation between the trends of total DMS flux and of concentration. This result suggests that to first order the trend in DMS concentration determines the trend in DMS flux, while the change of other variables involved in the flux calculation, for instance the mean wind speed and the sea surface temperature, is of secondary importance. To gain further insight from these time series, the relative difference of DMS concentration and flux between the beginning of

530 historical (1850–1865) and the end of scenario simulation (2085-2100) is shown in Table 7. It reveals that for each model, whatever the sign of the trend, the relative change in flux (−4 to 13.8 %) is shifted towards positive values compared to the relative change in concentration (−12.3 to 10 %), thus mitigating negative flux trends, but reinforcing positive ones. This is likely explained by the positive dependence of the gas-exchange parameterisations on the SST, which is sharply increasing in ssp585 simulations (see the timeseries of modelled SST in Fig. SI-7). Conversely, the mean annual wind speed over the oceans

shows no or a weakly negative trend, depending on the model (Fig. SI-7), and thus cannot explain an increased DMS flux. Further investigations would be required to evaluate these effects on a finer spatial and temporal scale, however a more detailed analysis of the drivers involved in flux parameterisations is beyond the scope of this study. Notwithstanding the uncertainties with regards to the other drivers, the major role of the DMS concentration to explain the trend in DMS flux is an important result which confirms and goes farther than the conclusion of Tesdal et al. (2016), who showed that in modern climate the

global total flux of DMS depends primarily on the global mean DMS concentration.

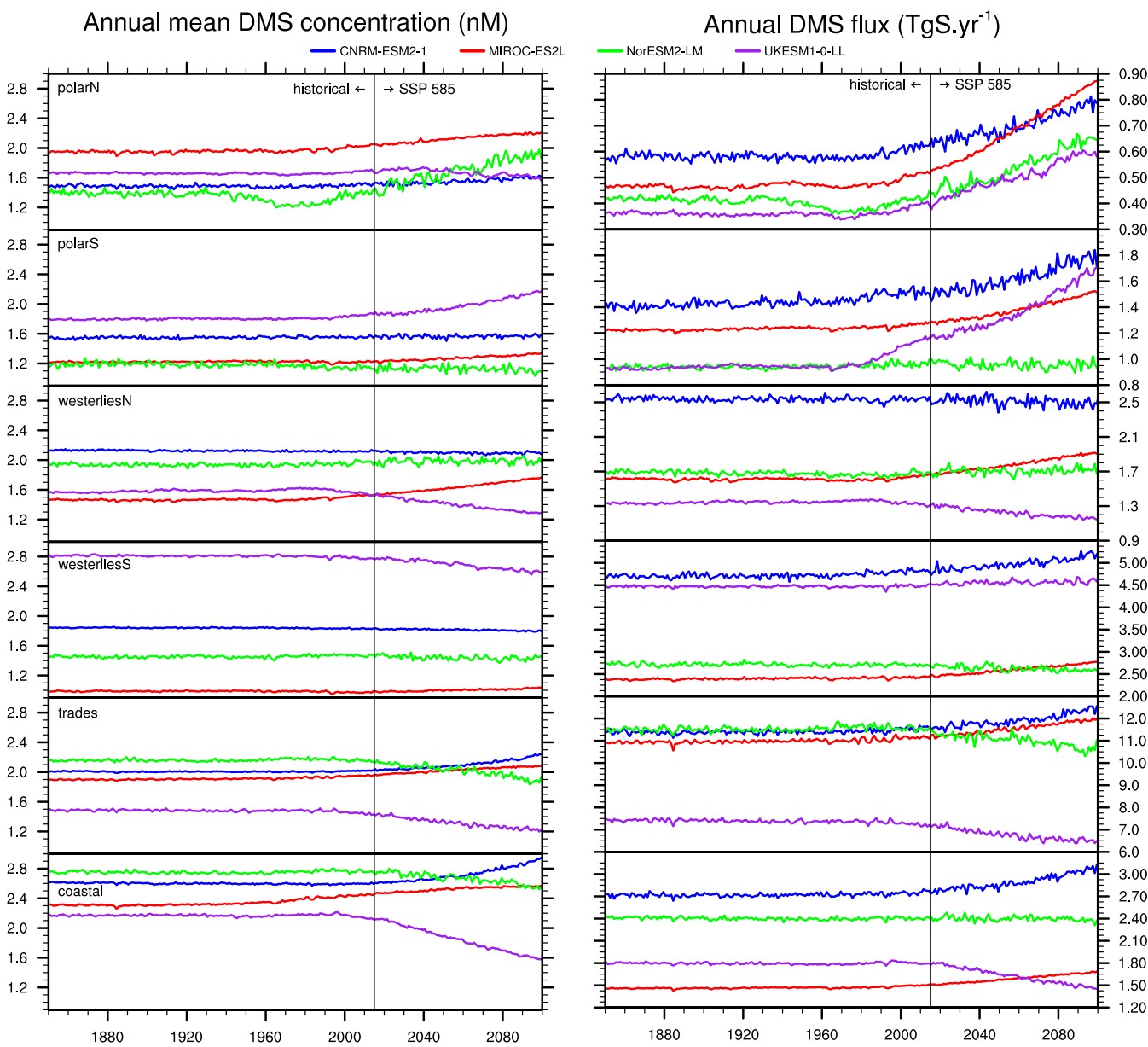

**Figure 10.** Time series of mean annual area-weighted surface ocean DMS concentration (nM, left panels) and annual regional DMS fluxes (Tg S year$^{-1}$, right panels) over $1850 - 2100$ (CMIP6 historical and ssp585 simulations, model ensemble means). Six different Longhurst (2007) biomes are distinguished, from top to botton: polarN, polarS, westerliesN, westerliesS, trades, and coastal (see also Fig. 5 for a mapping of these regions). Please note the common y-axis for all DMS concentration plots.

To gain further insight into the respective trends of DMS concentration and flux, Fig. 10 compares both trends on the six ocean biomes. Overall, the behaviour depicted at the regional scale is similar to what is seen at a global scale: when a trend

of DMS concentration is present, the trend of DMS flux has generally the same sign for most models and biomes. A few exceptions to this occur when other processes are at play, and overcome the DMS concentration as main driver of the flux. This is namely the case in polar biomes, where the fraction of sea-ice cover is expected to be an important driver due to the linear scaling of DMS emission to the ice-free water fraction (see Sect. 3.2). To help understand the underlying mechanisms, Figure SI-8 shows the timeseries of the annual mean sea-ice concentration in both polar biomes. In the Arctic, all models agree on a strong decline in sea-ice, and unanimously predict a sharp rise of DMS flux. The trend in DMS concentration also leaves an imprint in that of flux, but is of secondary importance. Similarly, in the polar S biome, apart from NorESM2-LM which shows no significant trend in any of the three variables, the other models also predict a rise in DMS flux following either the decrease in sea-ice cover or the increase in DMS concentration (or both in the case of UKESM1-0-LL). An in-depth analysis of the Arctic response will be presented in a subsequent section of the paper.

A comparison of Figures 9 and 10 also shows that the global trends of DMS concentration and flux have the same sign as in the trades biome, which represents 50 % of the total ocean surface, and accounts for roughly half of global DMS emissions (45–60 % depending on the model). The coastal biome also depicts a similar response, which is in line with previous finding (Sect. 3.1.2) that this biome could respond similarly to trades or westerlies biomes depending on the latitude. Therefore, improving the models in the low latitudes regions is needed to gain confidence in the predicted global trends of DMS.

Figures 11 and 12 complement the view by global and biome regions with maps of the trends over 1980–2009 and 2071–2100. Only the trends whose statistical significance is $\geq$ 90 % are shown. These maps further demonstrate the good agreement between the trends of concentration and those of flux: pattern correlation range between 0.42 and 0.81 over the 1980–2009 period, and reach 0.89 over the 2071–2100 period. These maps also show that for each model, most features of these trends are similar in both periods. Over 2071-2100, trends of both variables appear larger than trends over 1980–2009, more so for the CNRM-ESM2-1 and the NorESM2-LM models. The extent of statistically significant areas increases between both periods, and is larger for the models using empirical parameterisations (MIROC-ES2L and UKESM1-0-LL), which might be due to a more straightforward response to a limited set of internal drivers, thus reducing the spread and increasing the statistical significance. It is also worth noting that the trend in DMS flux is statistically significant for all models in the Arctic region (at least in the Arctic margin, where the sea-ice retreat is more pronounced).

Ultimately, Figures 11 and 12 also reveal that even if two models agree on the sign of the global trends, the underlying mechanisms are not identical. For instance, comparing CNRM-ESM2-1 and MIROC-ES2L shows that the latter features more uniform trends, while the former has a more patchy response with higher spatial variability. Similarly, NorESM2-LM and UKESM-1-0-LL show very distinct features, for instance with a strong increase in the Southern Ocean band for UKESM-1-0-LL while NorESM2-LM shows a decrease in this region. Opposite behaviours can also be observed in the North Pacific and North Atlantic regions in both periods, with an increase predicted by NorESM2-LM, and a decrease predicted by UKESM1-0-LL.

Several previous studies have investigated the evolution of DMS concentration and flux in the future. Although these studies are not necessarily comparable to ours with respect to scenarios, model type and methods, we summarize the results of these studies regarding global DMS trends. An early study by Gabric et al. (2004) using a $3\times CO_2$ scenario found an annual mean

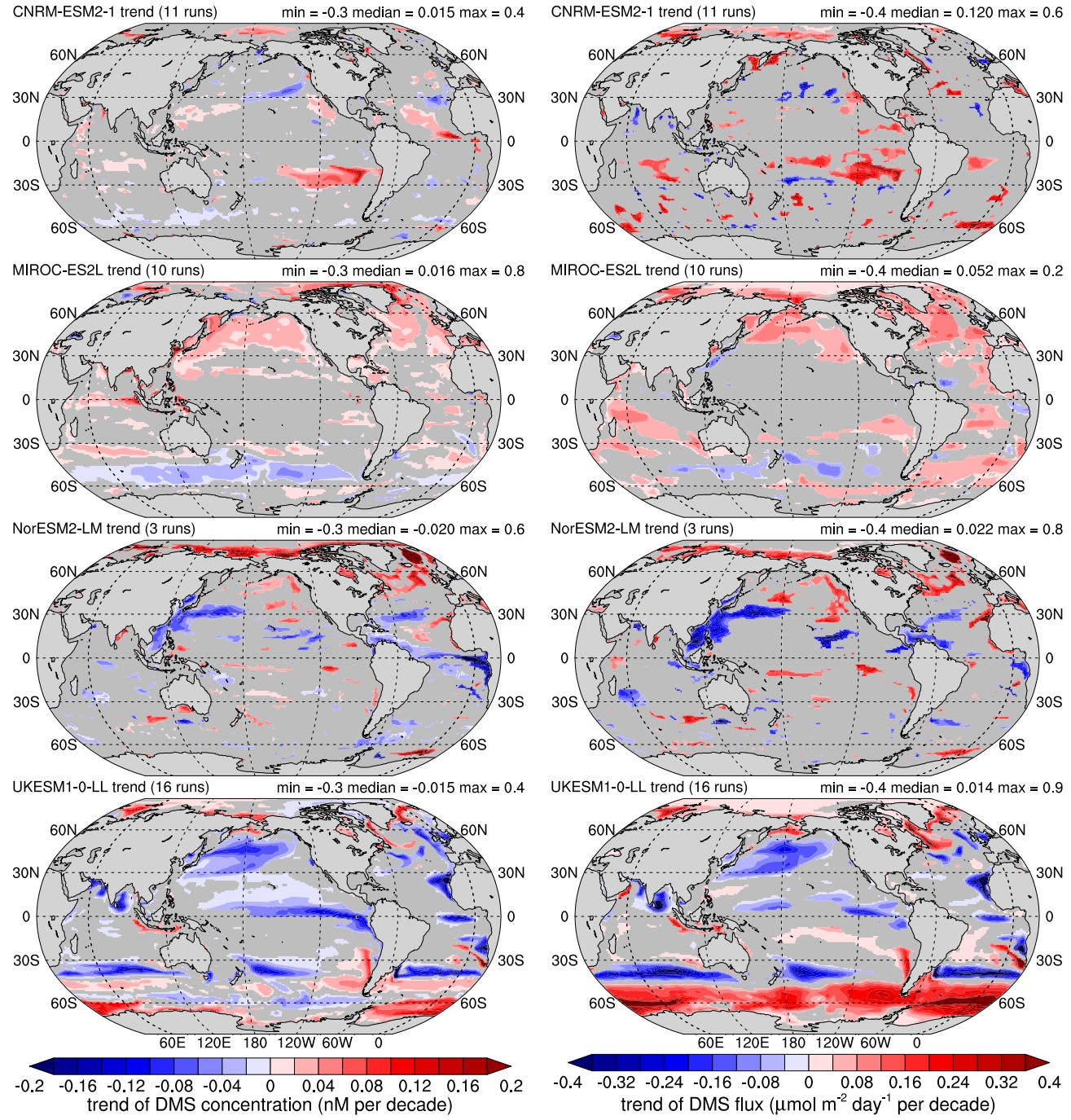

**Figure 11.** Mean trends over 1980–2009, left surface ocean DMS concentration ($\mathrm{nM\,decade^{-1}}$), right DMS flux into the atmosphere ($\mathrm{\mu mol\,m^{-2}\,decade^{-1}}$). Shaded areas denote a statistical significance below the 90 % level.

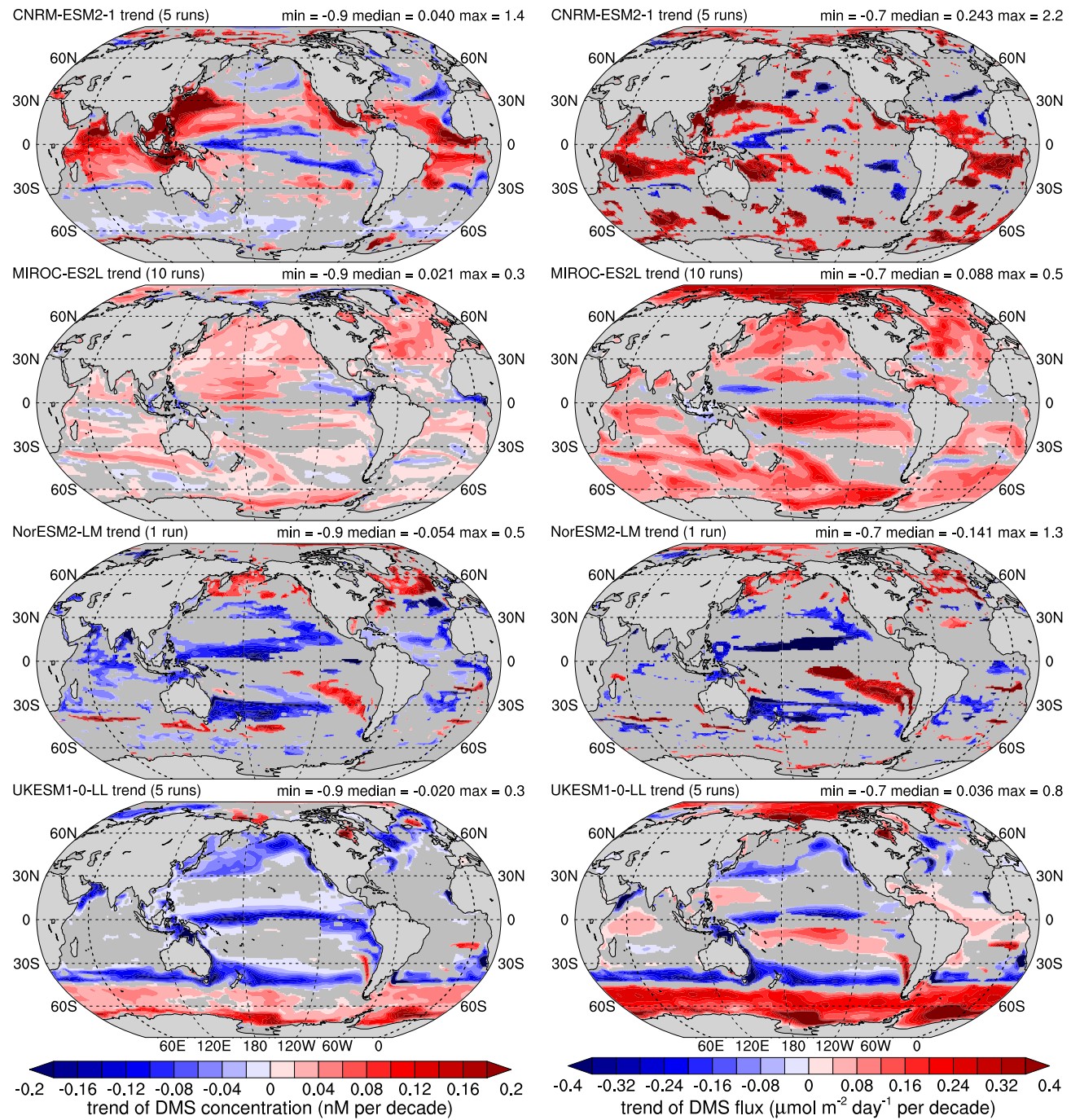

**Figure 12.** Same as Fig. 11 with trends computed here over 2071–2100 from the ssp585 simulations. Note the scales are identical to Fig. 11.

DMS flux increase by 14 %. Bopp et al. (2003) conclude that under $2\times CO_2$ (assuming a 1 % increase per year), the global DMS emission increases by 2 %, and they further observe that there are large spatial heterogeneities (flux change ranging from

580 −15 to 30 % in zonal mean). Vallina et al. (2007) addressed the effect of upper mixing shoaling induced by global warming on the flux of DMS, and found a small increase by 1.2 %. Kloster et al. (2007) performed a global analysis with HAMOCC, under scenario SRES A1B, and found a decrease in the global DMS flux by 10 %. They also concluded that the simulated changes have large spatial variations. Their results (see Kloster et al., 2007, Fig. 4) indeed show negative trends almost all over the oceans, apart from the Arctic and in a thin band around Antarctica. Some features are slightly similar to those displayed

by NorESM2-LM, which is not surprising given the similarity in the ocean carbon cycle models. Wang et al. (2018) performed simulations with the Community Earth System Model (CESM) with RCP8.5 scenario, which computes a global decrase in DMS flux of −8.1 % in 2100, with significant spatial variability. These findings are consistent with those of Kloster et al. (2007), and agree qualitatively with the results obtained with NorESM2-LM despite the projected DMS decrease is only half that found by Wang et al. (2018). Our findings for MIROC-ES2L and UKESM-1-0-LL agree qualitatively with the results

of Halloran et al. (2010), who performed an offline evaluation of the trends predicted for surface ocean DMS concentration, using the output of HadCM3 climate simulation following the IPCC SRES 2a scenario, and the schemes of Anderson et al. (2001) and Simó and Dachs (2002). A moderate global reduction was found with the former, as in UKESM1-0-LL, while a pronounced global increase was found with the latter scheme, as in MIROC-ES2L. However, the patterns of trends in the study of Halloran et al. (2010, Fig. 5) do not closely resembles those in MIROC-ES2L and UKESM1-0-LL. Two modelling

studies by Six et al. (2013) and Schwinger et al. (2017) specifically addressed the issue of decreasing seawater pH along with the rise in $CO_2$. Whilst these studies used fairly different scenarios (SRES A1B and RCP8.5, respectively), they both found a decrease in global DMS flux respectively of 18±3 % and 17–27 % (depending on the chosen pH sensitivity) by the end of the 21[th] century. However, other studies focusing on specific regions or processes have opposite conclusions: for instance, Kim et al. (2010) found an increase in DMS production due to enhanced grazing rates; Gypens and Borges (2014) suggested that

eutrophication can offset the acidification effect and lead to a rise in DMS production in coastal environments. Other studies focused specifically on regional trends, such as in the Southern Ocean (Cameron-Smith et al., 2011; Qu et al., 2020). Both studies reported a very large increase in DMS flux, however this regional behaviour, which strongly depends on the sea-ice cover, cannot be generalised at the global scale. Last, we can also mention the sensitivity analysis performed by Wang et al. (2020, Fig. 9). After they built the W20 climatology using the ANN (see Sect. 2.2.1), they individually changed the most

influential variables (SST, MLD, PAR, SSS and three nutrients) and assessed the resulting change in concentration. The ANN has a non-linear, positive or negative response to each of these variables, and this sensitivity analysis reveals that the overall trend would be a complex combination of multiple factors.

To conclude, there is no agreement in the literature with regards to the sign and amplitude of the trends of DMS concentration and flux in the future (Hopkins et al., 2020). Numerous processes are at play, and there is considerable uncertainty in the overall

result. There is no general consensus either in the CMIP6 models regarding the DMS trends, neither on the regional patterns, nor on the sign of the global annual trend. By the end of the 21[st] century, in the ssp585 simulation, the four CMIP6 models predict a relative change in DMS emission between −4.0 and 13.8 %, which agrees with the large range of estimated values from previous studies. The only exception where models unanimously predict a sharp rise in DMS emission is in the Arctic, and this is analysed further in section 4.3 below.

## 4.2 Role of marine biology

The aim of this section is to answer whether other variables could give further insight on the most likely trend of DMS concentration, especially in low latitudes regions which determine the global trend as found in the previous section.

The biological control of the change in DMS concentration has been highlighted in previous modelling studies (e.g. Bopp et al., 2003; Kloster et al., 2007) and merits to be re-assessed here in a multi-model perspective. In the following, we investigate this relationship for the CMIP6 models using marine net primary productivity (NPP). Figure 13 shows how changes in ocean DMS are associated with changes in NPP across in the four CMIP6 models over 1980–2009 and 2071–2100 (the total annual NPP is also plotted in Fig. SI-9, to compare with the similar plot for DMS concentration, Fig. 9).

**Table 8.** Metrics of the scatter plots of Fig. 13, change in DMS concentration versus change in NPP: slopes of the linear regressions ($10^{-2}$ nM per $\mathrm{g\,C\,year^{-1}}$) and determination coefficients. Metrics are shown for the four models, over two periods (1980-2009 and 2071-2100) in global mean and for the 6 biomes, as in Fig. 13.

| Model | global Slope | $R^2$ | polar N Slope | $R^2$ | polar S Slope | $R^2$ | westerlies N Slope | $R^2$ | westerlies S Slope | $R^2$ | trades Slope | $R^2$ | coastal Slope | $R^2$ |
|---|---|---|---|---|---|---|---|---|---|---|---|---|---|---|
| | | | | | | historical (1980–2009) | | | | | | | | |
| CNRM-ESM2-1 | 0.39 | 0.80 | 0.76 | 0.59 | 0.74 | 0.24 | −0.01 | 0.02 | 0.00 | 0.00 | 0.51 | 0.78 | 0.19 | 0.64 |
| MIROC-ES2L | −0.40 | 0.29 | 2.55 | 0.88 | −0.04 | 0.02 | −1.38 | 0.72 | −0.18 | 0.20 | −0.63 | 0.68 | 1.42 | 0.57 |
| NorESM2-LM | 0.94 | 0.86 | 4.98 | 0.96 | 3.09 | 0.84 | 1.59 | 0.84 | 0.89 | 0.48 | 0.82 | 0.89 | 1.87 | 0.87 |
| UKESM1-0-LL | 1.33 | 0.99 | 0.51 | 0.92 | 1.15 | 0.99 | 1.15 | 0.97 | 2.00 | 0.90 | 1.09 | 0.97 | 1.27 | 0.98 |
| | | | | | | ssp585 (2071–2100) | | | | | | | | |
| CNRM-ESM2-1 | 1.03 | 0.94 | 0.47 | 0.65 | 0.37 | 0.25 | 0.37 | 0.39 | −0.19 | 0.63 | 1.59 | 0.92 | 0.93 | 0.91 |
| MIROC-ES2L | −1.27 | 0.87 | −1.57 | 0.63 | 1.28 | 0.85 | −2.33 | 0.78 | 0.31 | 0.76 | −0.56 | 0.88 | −0.42 | 0.62 |
| NorESM2-LM | 1.36 | 0.87 | 1.84 | 0.70 | 2.07 | 0.68 | 1.64 | 0.78 | 1.91 | 0.78 | 1.41 | 0.91 | 2.01 | 0.91 |
| UKESM1-0-LL | 0.99 | 0.98 | 0.99 | 0.73 | 1.15 | 0.99 | 0.79 | 0.95 | 2.18 | 0.81 | 0.92 | 0.96 | 0.80 | 0.99 |

Figure 13 and Table 8 map two groups of model responses regarding the link between the change in DMS and that of NPP: over the historical period (1980–2009), despite important scatters, three models (CNRM-ESM2-1, NorESM2-LM and UKESM1-0-LL) display a positive correlation between changes in DMS and changes in NPP. This positive correlation is further confirmed for these models in the future (2071-2100), with larger differences and less scatter, which depict a robust relationship between both variables. Beside the qualitative agreement between these three models, the slopes of linear regressions are highly variable between models, between biomes, and between both periods (Table 8), thus considerably limiting the predictive capability of such relationship. In contrast, MIROC-ES2L shows no clear correlation between DMS and NPP, and even a weak negative correlation across the biomes in the future. This specific behaviour, contrasting with that of the other models, stems from the parameterisation of Aranami and Tsunogai (2004), in which there is a loose biological control on the DMS

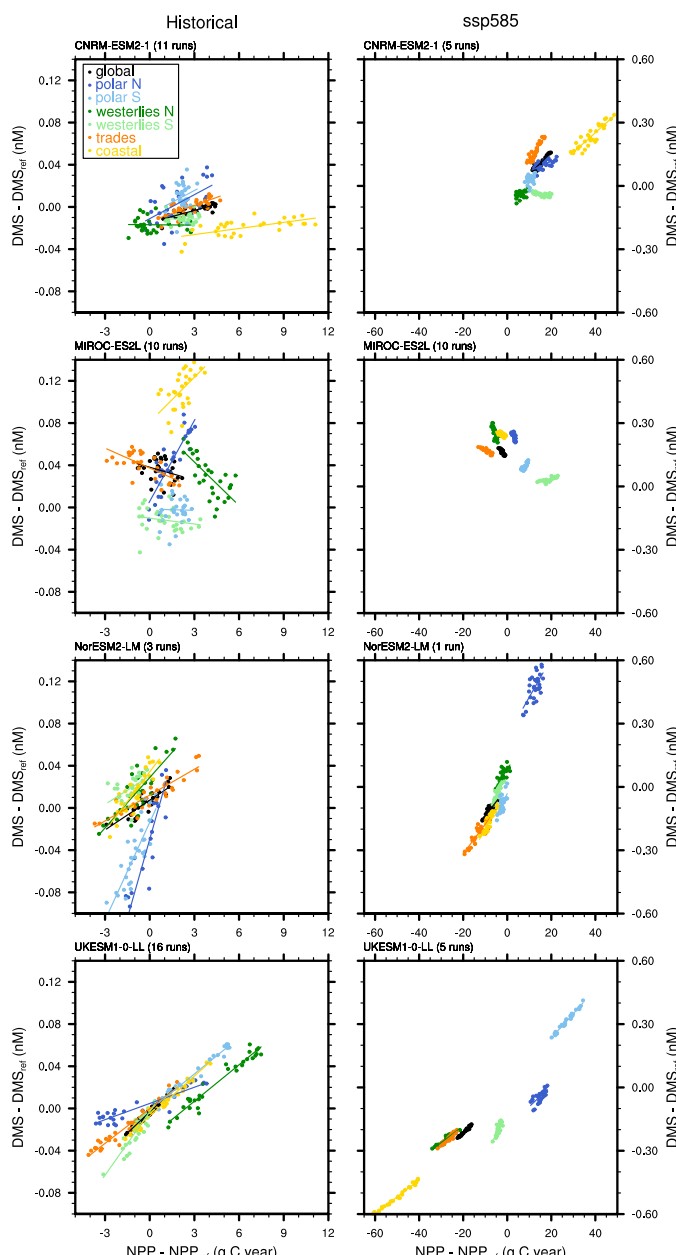

**Figure 13.** Scatter plot of change of DMS concentration as compared to the change in NPP as simulated by CMIP6 models over the historical 1980–2009 period (left column) and ssp585 2071–2100 period (right column). For both variables, the reference (DMS$_{ref}$ and NPP$_{ref}$, respectively) is the mean value over 1850–1879. Each scatter plot displays global values (black dots) and the values in each of the 6 biomes (dark and light blue: polar N and S, dark and light green: westerlies N and S, orange: trades, yellow: coastal); see Fig. 5 for a mapping of these regions. Linear regressions are shown, and related information is given in Table 8. Mind the different scales between both columns.

concentration, through the positive relationship with the Chl which only occurs in high productivity zones (see Sect. 2.1.1 and Fig. SI-9 showing the timeseries of annual mean Chl, to compare with the similar plot for DMS concentration, Fig. 9).

Although the limited current knowledge about the NPP-DMSP-DMS relationships hampers our ability to constrain this emergent property, several lines of evidence tend to suggest that there is a positive correlation between NPP and DMS concentration. Firstly, noting that some studies observed no correlation between DMS and Chl *a* (e.g., Wang et al., 2020, and references therein), a number of other studies showed positive correlations between NPP and DMS production: the link between NPP and DMSP is highlithed at the local scale (e.g., Simó et al., 2002) and at a basin wide scale (e.g., Uhlig et al., 2019), that between NPP and DMS concentration again at a basin wide scale in Osman et al. (2019), and the link between DMSP and DMS concentration has been described in several studies (e.g., Stefels, 2000; Yoch, 2002; Asher et al., 2017; Lizotte et al., 2017). Secondly, factorial experiments conducted by Wang et al. (2020) using an artificial neural network show that a 10 % decrease of Chl a leads to a reduction in DMS concentration in large open-ocean domains. Finally, previous modelling work of Bopp et al. (2003) and of Kloster et al. (2007) show that the response of the marine biology (i.e., declining NPP) is one of the prominent drivers of changes in DMS emissions. The first group of models (CNRM-ESM2-1, NorESM2-LM and UKESM1-0-LL) thus captures a relationship which is consistent with such ocean field experiments, while the response simulated in MIROC-ES2L is not consistent with the current understanding of the DMSP production pathways by marine phytoplankton (Stefels et al., 2007).

Such relationship between changes in NPP and changes in DMS has consequences for future projections because it suggests that the overall model response in DMS concentration and emission will mirror changes in NPP. A recent study by Kwiatkowski et al. (2020, see Figs. 1e and 2o) synthesised the prediction of ten CMIP5 and thirteen CMIP6 models for several diagnostics, and concluded to a relative change in NPP of $-2.99 \pm 9.11$ % in ssp585 scenario (2080–2099) relative to the 1870–1899 mean (Kwiatkowski et al., 2020, Table 4). As compared to the CMIP5 generation of models, which predicted a relative change of $-8.54 \pm 5.88$ % in scenario RCP8.5, the uncertainty has thus increased, and now covers positive trends. This confirms that the response of models in the low-latitude oceans remains highly uncertain, even regarding the sign of the trends, thus limiting the current ability to predict future changes of DMS concentration and emissions in these regions, and thus on a global scale.

## 4.3 Focus on the Arctic region

In polar regions, the role of DMS concentration in governing the DMS emission is superseded by the dynamics of sea-ice cover, in line with the common model assumption of a linear relation between the DMS flux and the ice-free area fraction. To make it clear to the reader, we emphasise here that in all CMIP6 models, the DMS flux variable represents the actual flux over the entire grid cell, and thus already accounts for the reduction due to the sea-ice, if present.

Figure 14 shows time series of various diagnostics over the Arctic region. The motivation for this figure is to present information as shown in Galí et al. (2019) (hereafter G19) and thus evaluate how/if the four CMIP6 models confirm some conclusions derived from the satellite analysis of G19. Three regions are distinguished i.e., the pan-Arctic region (north of 70° N) and its two divisions, an Atlantic sector and a non-Atlantic sector (see Fig. 4H of Galí et al., 2019). The area-weighted mean DMS flux and concentration (third and fourth row in Fig. 14) are computed considering ice-free grid cells (as in G19),

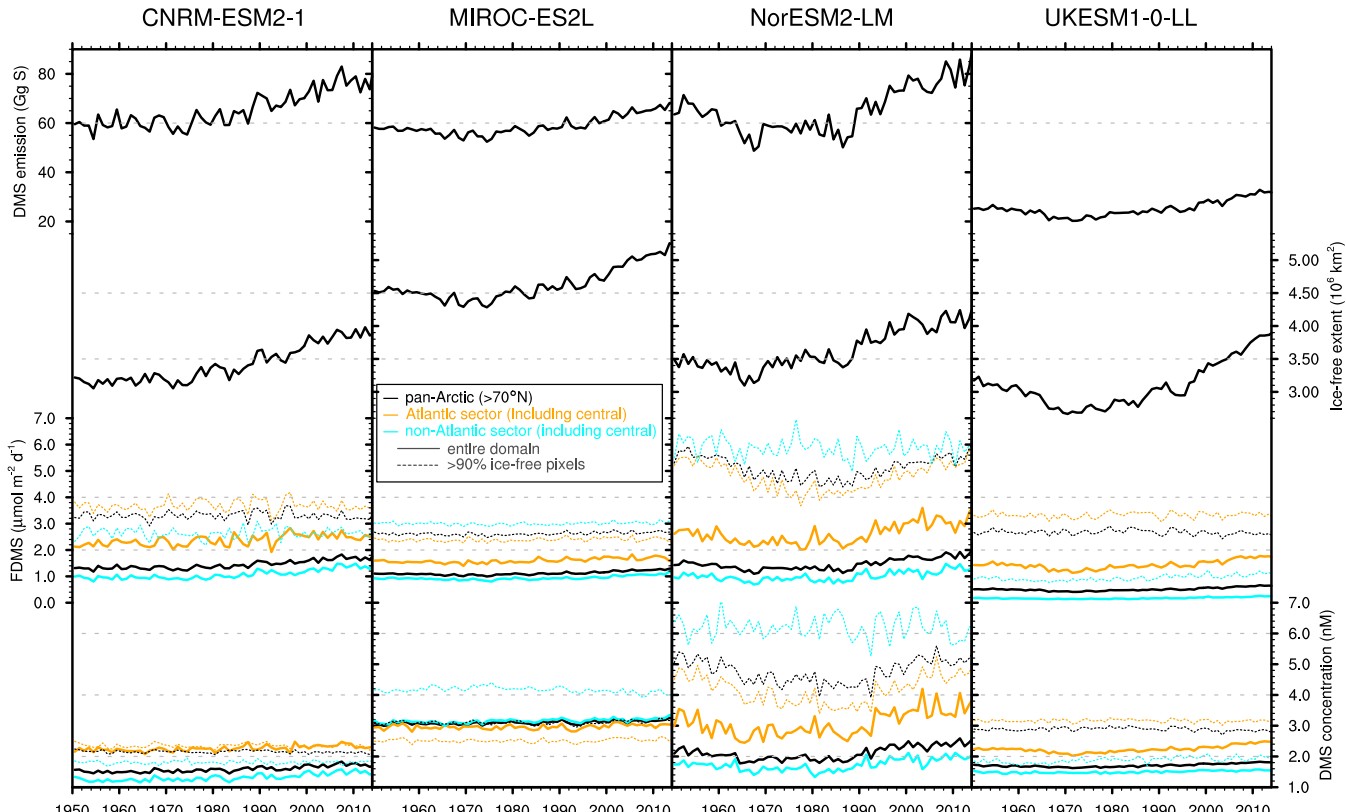

**Figure 14.** From top to bottom in each column are time series (1950–2014) of May–August values of: integrated DMS emission (Gg S), mean ice-free extent ($10^6$ km$^2$), area-weighted mean DMS flux ($\mu$mol m$^{-2}$day$^{-1}$), and area-weighted mean DMS concentrations (nM). The same four months (May to August) and the same regions as those of Galí et al. (2019) are shown for the CMIP6 models in the four columns: black lines pan-Arctic (>70° N), cyan lines non-Atlantic sector (including central), and orange lines Atlantic sector (including central). Solid lines computed from all pixels of a given region. Dashed lines for DMS fluxes and DMS concentrations computed only from >90 % ice-free pixels of a given region, as in Galí et al. (2019). Grey dotted lines are visual aids to ease the comparison between models.

which are cells where 90 % or more of the surface is sea-ice free. We also show results considering all pixels of a given region to evaluate the importance of the sea-ice masking. Time series are shown over part of the CMIP6 historical period, 1950–2014, and this extends the G19 time series (1998–2016).

Several of the time series presented in Figure 14 show interannual variations whose amplitude depends on the model, with larger interannual variations for CNRM-ESM2-1 and NorESM2-LM, and smaller ones for the other two models. Time series of ice-free extent are similar among models except for MIROC-ES2L which predicts significantly less sea-ice. However, the dynamics of sea-ice retreat from 1980 onwards is roughly consistent between models, with a median rate of $+0.23\times10^6$ km$^2$ per decade. This Figure provides new insights regarding the behaviour of models in the Arctic. First, all models show higher DMS concentration over the ice-free pixels (dashed lines) as compared to the entire domains (full lines). This means that

the models consistently predict lower DMS concentration below the sea-ice, in line with reduced photosynthetically active radiation. We want to note here however that a number of recent studies highlighted the large DMS production of ice algae and acknowledged that models likely underestimate the contribution of bottom ice DMS (Hayashida et al., 2020, and references therein). The only exception is seen in the Atlantic sector for MIROC-ES2L, where the mean DMS concentration over ice-free pixels is lower than that in the whole sector. This is likely due to the specific negative correlation between DMS and MLD in

the parameterisation of Aranami and Tsunogai (2004) (see Sect. 2.1.1). The MLD is expected to be thicker in the absence of sea-ice, thus leading to lower DMS concentration. Because the fraction of ice-free water is much higher in the Atlantic sector (∼70 %) than in the non-Atlantic sector (∼30 %), this specific effect of the MLD is expected to leave a stronger imprint on the DMS concentration in the latter, thus explaining this exception.

As compared to the DMS concentration values found by G19 between 1998 and 2016 (in the 2.7–3.0 nM range in the

685 Atlantic sector, in the 3.5–4.5 nM range in the non-Atlantic sector), MIROC-ES2L shows a rather good agreement, while CNRM-ESM2-1 and NorESM2-LM display much lower and higher concentration values, respectively. UKESM1-0-LL has a concentration value in the Atlantic sector which agrees with that in G19, but the concentration in the non-Atlantic sector is roughly half that in G19. G19 further discussed the differences in biogeochemical and meteorological characteristics in both Arctic sub-sectors, to explain why DMS concentration is larger in the non-Atlantic sector. In particular, as the non-Atlantic

sector includes the Siberian shelves, which seem to be quite productive owing to nutrient inputs from large rivers (Terhaar et al., 2021), the G19 data may be biased high in the Siberian shelves due to optical interference of continental materials (Hayashida et al., 2020). While uncertainty in satellite DMS appears higher in the non-Atlantic sector, ESMs possibly struggle to capture the biogeochemical functioning in shallow Arctic seas, due to both too-low resolution and non-represented processes. Notewithstanding the biases in models as compared to G19, only MIROC-ES2L and NorESM2-LM correctly capture this

difference between sectors with higher DMS concentration in the non-Atlantic sector.

Comparing the time series of DMS concentration and flux together demonstrates again the tight relationship between both variables. Especially, when considering only the means over ice-free pixels (dashed lines), the specific role of sea-ice is thus negligible, and the similar behaviours are clearly visible. Comparing the time series of DMS concentration and flux together demonstrates again the tight relationship between both variables, especially when considering only the means over ice-free

pixels (dashed lines). As compared to G19, who found a mean DMS flux in the 3–4 $\mu$mol m$^{-2}$ day$^{-1}$ range whatever the region (with one year up to 5 $\mu$mol m$^{-2}$ day$^{-1}$), three models tend to underpredict the flux, while NorESM2-LM significantly overestimate the flux, with pan-Arctic values in the 4.5–6 $\mu$mol m$^{-2}$ day$^{-1}$ range. Accounting for all pixels without sea-ice criterion (full lines) mathematically decrease the resulting mean flux. This mean flux (1950–2014) shows a factor of nearly 3 between the minimum (UKESM1-0-LL: 0.51 $\mu$mol m$^{-2}$ day$^{-1}$) and the maximum (NorESM2-LM: 1.47 $\mu$mol m$^{-2}$ day$^{-1}$).

All together, these observations made for the sea-ice extent, and the DMS concentration and flux, explain well the resulting DMS integrated emissions from May to August (top row in Fig. 14), which also differ by a factor of 2 to 3, with UKESM1-0-LL around 30 Gg S and NorESM2-LM around 80 Gg S at the end of the period. Rates of increase in DMS emission between 1980 and 2014 vary between 2.9 Gg S decade$^{-1}$ (lowest increase rate for UKESM1-0-LL) and 9.2 Gg S decade$^{-1}$ (highest rate for NorESM2-LM) while the estimate of Galí et al. (2019) over 1998–2016 is 13.3 ± 6.7 Gg S decade$^{-1}$.

Scatter plots of DMS emissions versus ice-free extent over the 1950-2014 period are shown in Figure 15, with related metrics in Table 9, to provide further insight into this relationship. As in G19 (see Figs. 4H and 4I), information is also given over a fourth domain, the central basin with heavier sea-ice cover, whose area is subtracted here from the Atlantic and non-Atlantic sectors. Over the pan-Arctic region, determination coefficients ($R^2$) are largely higher in all CMIP6 models than those of G19. This reflects the linear dependence of the flux to the free-water fraction in the models, though in observations additional factors

such as ocean productivity can be invoked to explain scatter in the DMS emission versus sea-ice extent relationship (Galí et al., 2018; Lewis et al., 2020). Smaller interannual variability in models compared to satellite observations can also contribute to higher $R^2$. Slopes of the relationship differ for the four models, with the smallest slope for UKESM1-0-LL (10.4 in Gg S per 10 % of free-water) and the largest slope for NorESM2-LM (31.8 Gg S per 10 % of free-water). To first order, these slopes result from the combination of the mean DMS flux and of the trend in sea-ice retreat. In total, the CMIP6 summertime DMS

emissions extrapolated at 100 % sea-ice free water vary between 72 and 310 Gg S, enlarging the corresponding estimation of $144 \pm 66$ in G19. Because the current ice-free extent is significantly larger (in the 60–85 % range) over the Atlantic sector, the error at 100 % extrapolation is reduced, with emission estimates ranging between 27–50 Gg S. Conversely there is a huge discrepancy in the central sector (11–151 Gg S).

    We further extended this analysis to the ssp585 scenario, using the 65-year long period from 2036 to 2100 so that both

analyses rely onto the same time windows. Scatter plots of DMS emissions versus ice-free extent (see right column of Figure 15) confirm the linear relationships determined from the years 1950–2014 between these two fields (see previous paragraph). Unlike in the 1950–2014 years, the Atlantic sector appears to be mostly free of ice during the summer months, thus the relationship between DMS flux (and thus emission) and ice-free extent is almost lost. Extrapolations of annual DMS emissions at 100 % ice-free extent for the 2036-2100 period (from 86 to 282 Gg S for the pan-Arctic region) are comparable to projections

inferred from the 1950–2014 period (72 to 310 Gg S, see Table 9). In the non-Atlantic sector and central basin, the disagreement in extrapolated values at 100 % ice-free water is reduced following the reduced extrapolation errors. All CMIP6 models thus depict a consistent evolution throughout the 21$^{st}$ century in the Arctic region, where DMS emission is determined to first order by the ice-free extent, while other factors are of secondary importance. This modelled behaviour agrees well with the conclusions of Hayashida et al. (2020), who found that the decline of Arctic sea ice is associated with a quasi-linear positive

trend of DMS flux.

## 5    Conclusions

In this study, we analyse surface ocean DMS concentration and flux into the atmosphere from four CMIP6 ESMs (CNRM-ESM2-1, MIROC-ES2L, NorESM2-LM and UKESM1-0-LL) over the historical and ssp585 simulations. The parameterisations of DMS in these ESMs have various degrees of complexity and while they may have already been evaluated, either in a

previous (e.g., Le Clainche et al., 2010) or in their current version, it is the first time that this is done in a common coupled atmosphere-ocean simulation framework. Our study also provides an evaluation of the performance of the current generation of ESMs against the most up-to-date observational products, both in terms of mean state and of current and future trends.

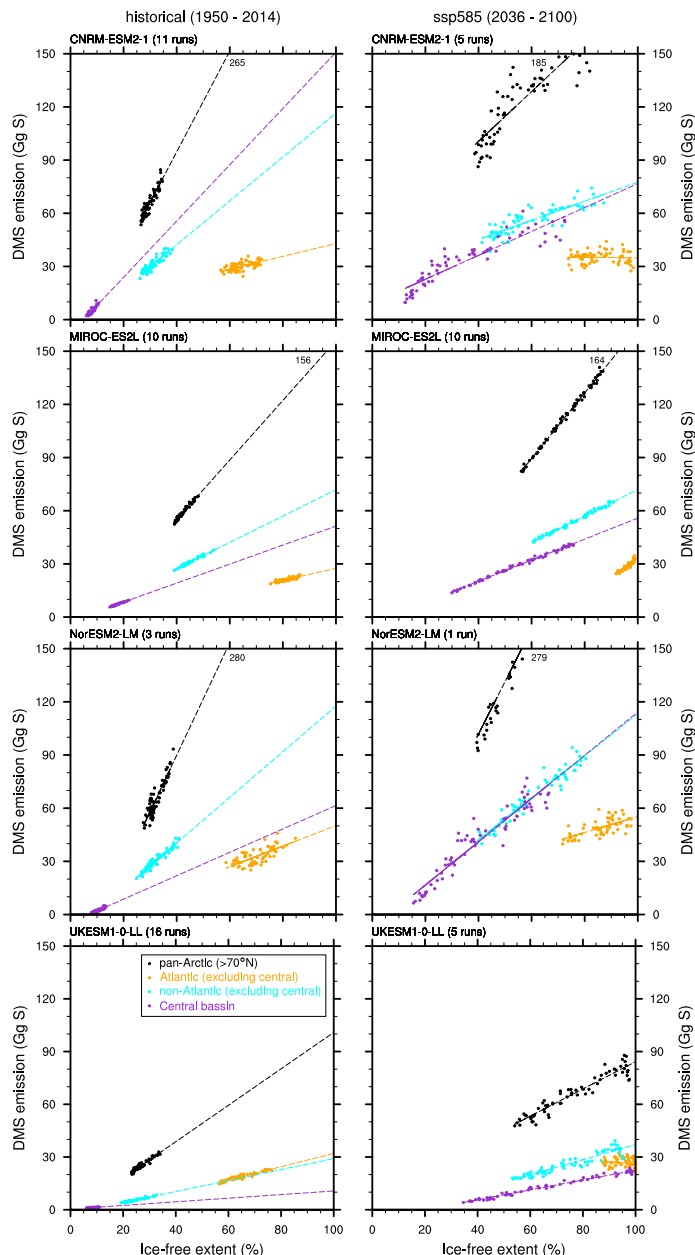

**Figure 15.** Scatter plots with mean annual May to August values of DMS emissions (Gg S) versus ice-free extent (%) over the same regions as those of Galí et al. (2019): pan Arctic (black dots), non-Atlantic sector without the central basin (cyan dots), Atlantic sector without the central basin (orange dots), and central basin (purple dots), for the 4 CMIP6 models (4 rows). Left column: 1950–2014 years, right column: 2036–2100 years. DMS emissions at 100 % ice-free extent are indicated for better clarity when the extrapolated values exceed the y-axis maximum. The related metrics of these scatter plots are displayed in Table 9)

**Table 9.** Metrics of the scatter plots of Fig. 15, DMS emissions versus ice free extent: slopes of the regression lines (Gg S per 10 percent of free-water extent), projected DMS emissions at 100 % free-water (Gg S year$^{-1}$), and determination coefficients. Metrics are shown for the pan-Arctic (with sums of the individual regions in parentheses), Atlantic and non-Atlantic sectors (excluding the central basin), and the central basin as in Galí et al. (2019), for the four models. Values from Galí et al. (2019) are also presented.

| Model | pan-Arctic | | | Atlantic | | | non-Atlantic | | | Central | | |
|---|---|---|---|---|---|---|---|---|---|---|---|---|
| | Slope | 100 % | $R^2$ | Slope | 100 % | $R^2$ | Slope | 100 % | $R^2$ | Slope | 100 % | $R^2$ |
| | | | | historical (1950-2014) | | | | | | | | |
| CNRM-ESM2-1 | 28.4 | 265 (310) | 0.87 | 3.6 | 43 | 0.37 | 12.4 | 116 | 0.84 | 15.9 | 151 | 0.80 |
| MIROC-ES2L | 16.8 | 156 (150) | 0.97 | 3.4 | 27 | 0.77 | 7.4 | 72 | 0.98 | 5.4 | 51 | 0.98 |
| NorESM2-LM | 31.8 | 280 (229) | 0.83 | 5.8 | 50 | 0.49 | 12.9 | 118 | 0.93 | 6.6 | 61 | 0.81 |
| UKESM1-0-LL | 10.4 | 101 (72) | 0.94 | 3.8 | 32 | 0.91 | 3.1 | 29 | 0.95 | 1.0 | 11 | 0.74 |
| | | | | ssp585 (2036-2100) | | | | | | | | |
| CNRM-ESM2-1 | 14.0 | 185 (190) | 0.81 | −0.2 | 35 | 0.00 | 5.4 | 78 | 0.72 | 6.7 | 77 | 0.85 |
| MIROC-ES2L | 18.5 | 164 (161) | 1.00 | 11.7 | 33 | 0.87 | 7.5 | 72 | 0.99 | 5.9 | 56 | 0.99 |
| NorESM2-LM | 29.6 | 279 (282) | 0.93 | 4.5 | 55 | 0.45 | 11.9 | 113 | 0.93 | 12.2 | 114 | 0.91 |
| UKESM1-0-LL | 7.7 | 84 (86) | 0.91 | −1.0 | 26 | 0.02 | 4.3 | 37 | 0.88 | 2.8 | 23 | 0.98 |
| Galí et al. (2019) | 14.4±0.66 | 200±54 (144±66) | 0.68 | 3.8±0.20 | 41 | 0.52 | 8.1±2.1 | 84 | 0.69 | 1.9±0.3 | 20 | 0.68 |

Our analysis of contemporary (1980–2009) climatologies of simulated surface DMS concentration shows that, overall, agreement is poor between models, and also between models and reference datasets, such as the L11 dataset. This is consistent with previous work (Tesdal et al., 2016). The use of multiple modern observational climatologies, L11, G18 and W20 sheds additional light on the multi-model performance analysis. As concluded by previous authors (see for instance Galí et al., 2018, Sect. 4.1), the widely used L11 climatology likely overestimates climatological surface DMS concentration at the spatial resolution of climate models due to the combination of scarce and biased sampling. This could in part explain the unanimous low bias of the CMIP6 models when compared to L11. Models show a better agreement with observation based estimates when compared to W20. The range of model global annual median DMS concentrations (1.39–1.90 nM) encompasses the W20 median (1.65 nM), whereas that of L11 is 2.25 nM. Our work also shows that models have better spatial correlation with W20 as a reference dataset (coefficients from 0.13 to 0.46 for annual fields) than with L11 as the reference (coefficients from 0.08 to 0.26).

Analysis of the annual cycles in each of the 54 biogeographical provinces defined by Longhurst (2007) reveals that models better reproduce the annual cycles in mid to high latitudes (polar and westerlies biomes) than in low latitudes (trades biomes), in agreement with past studies (e.g., Le Clainche et al., 2010). Coastal provinces seem to respond in a similar way, with the ones located in mid to high latitudes often displaying a better agreement between models and observations. We note, however, that annual cycles in low latitudes are less pronounced and this may partly explain weaker correlations. Coastal model deficiencies

may also be associated to the coarse model grid resolutions and poor process representation of coastal marine biota and sediments.

The multimodel ensemble mean (MMM) shows good skills in reproducing spatial patterns and seasonal variability, and compares generally better with observational climatologies than individual models.

The comparison of marine DMS emissions confirms the importance of the air-sea flux parameterisation on the resulting flux. The estimates of DMS emissions relying on observed surface ocean DMS concentrations and state-of-the-art air–sea flux parameterisations range between 16–28 $\mathrm{Tg\,S\,year^{-1}}$, while CMIP6 model estimates show a smaller consistent range (i.e., 16–24 $\mathrm{Tg\,S\,year^{-1}}$). As a consequence, the multi-model best estimate is 19±3 $\mathrm{Tg\,S\,year^{-1}}$, which is 10 % lower than the 18–24 $\mathrm{Tg\,S\,year^{-1}}$ best estimate proposed by Tesdal et al. (2016), with an identical uncertainty range.

The comparison of trends of DMS fluxes and of DMS concentrations over the whole simulation period (historical + scenario; 1850–2100) reveals that the current generation of CMIP6 ESMs disagree on the sign of these trends. Two models (CNRM-ESM2-1 and MIROC-ES2L) simulate an increase in ocean DMS concentrations and emissions, whereas two other models (NorESM2-LM and UKESM1-0-LL) predict a moderate decrease. As a consequence, our work shows that the lead-order uncertainty in the future evolution of marine DMS emissions has not been reduced in the current generation of models compared to that of previous modelling experiments (e.g. Bopp et al., 2003; Kloster et al., 2007; Halloran et al., 2010).

Our analyses using CMIP6 ESMs confirm the conclusions of Tesdal et al. (2016) that global DMS emission depends primarily on global mean surface ocean DMS concentration, while the spatial distribution of DMS concentration and the parameterisation of ocean-atmosphere exchange coefficient are of secondary importance. Our study further demonstrate that to first order, changes in marine global DMS concentration determine the evolution of the global DMS emission to the atmosphere. All models consistently predict that the relative change in DMS emission (−4 to 13.8 %) is shifted towards positive values compared to the relative change in DMS concentration (−12.3 to 10.0 %), which is likely caused by the positive temperature dependence in the air-sea flux parameterisations. Models also agree that global trends in DMS concentration and flux are dominated by the trends in the trade biome region, since this region accounts for half of the global ocean area. Furthermore, models agree on an increase in DMS emission in polar regions, following the dynamics of sea-ice retreat. This shows that the trend of ice-free extent overcomes that of DMS concentration as the main driver in these regions. In this work, we have assessed this feature and compared to the results of Galí et al. (2019) over the Arctic. In this region, models agree on an increase in DMS emission related to the increase in free-water extent. The extrapolation of this relationship to 100 % free-water leads to a summertime emission ranging from 84 to 280 Gg S, thus encompassing the value of 200 ± 54 Gg S reported by Galí et al. (2019).

On the contrary, there is no consensus on how the current generation of models simulate the long-term trends in DMS concentrations and emissions in low-latitude biomes. Further investigating the relationship between DMS concentration and biological productivity reveals that three models (CNRM-ESM2-1, NorESM2-LM and UKESM1-0-LL) predict a positive correlation between the trend in ocean surface DMS concentrations and the trend in marine primary production, while the fourth model (MIROC-ES2L) displays no strong correlation between these variables. This raises questions regarding the ability of empirical parameterisations of DMS concentration to predict future evolution, since they have been calibrated in present conditions. Despite the qualitative agreement of the three models regarding the NPP-DMS relationship, the predictive ability

is limited given the large uncertainties in the future evolution of marine primary production. The modelling challenge here is particularly vast as Kwiatkowski et al. (2020) shows that this uncertainty is larger in CMIP6 models than in CMIP5 models.

Although none of the marine biogeochemical models studied here are currently intended to represent specific taxa of marine phytoplankton, it is interesting to connect the most likely behavior of those taxa in response to climate change. For instance, Dani and Loreto (2017) suggest that climate change may reduce the range of latitudes where DMS-producer phytoplankton taxa thrive, and hence lead to a reduction in DMS emission to the atmosphere in warm low-latitudes oceans. Overall, our work shows that there is a major uncertainty in low-latitude ocean where the change in DMS concentration results from the interplay of marine biology factors with many other environmental drivers (e.g., temperature, salinity, stratification, nutrient availability, acidification, large-scale circulation), which all may affect in both directions the trends in DMS concentration (Wang et al., 2020). Further analysis to disentangle the role of these factors is required, for instance along the lines of the meta-analysis of Galí and Simó (2015) that specifically addresses the issue of the "summer paradox". This would require important coordination among modellers to work in a multi-model perspective as only a few CMIP6 models include DMS and their DMS-related output are limited and insufficient at present to conduct such analysis. In turn, this large uncertainty in DMS concentration results in uncertainty in marine DMS emission to the atmosphere. Progress in the representation of the biogeochemical, and particularly the lower trophic ecosystem, dynamics in the low-latitude oceans will improve our ability to tighten the range of uncertainty in marine DMS emissions and hence ultimately constrain the direction and the magnitude of the DMS-climate feedbacks.

*Code and data availability.* Datasets from CMIP6 simulations are available from every ESGF node, such as https://esgf-node.ipsl.upmc.fr/search/cmip6-ipsl/ (last accessed 14 April 2021).

The climatology of Lana et al. (2011) is available from the Surface Ocean - Lower Atmosphere Study (SOLAS) web site (https://www.bodc.ac.uk/solas_integration/implementation_products/group1/dms/, accessed 12 July 2019). Four datasets are provided (in .csv format): the monthly DMS climatology; its upper and lower bounds based on an assessment of the uncertainty in the climatology estimate; and the original in-situ measurement dataset binned on a $1°\times1°$ grid. These files were converted in netCDF format for comparison with model outputs.

The climatology of Galí et al. (2018) is available at https://doi.org/10.5281/zenodo.2558511. Several grid resolution are provided, we only used the $1°\times1°$ file (algorithm: globsat; chlorophyll product: CHL; euphotic layer product: KD490) provided in netCDF format. A few file structure adjustments were required to make it CF compliant and be able to compare with model outputs.

The climatology of Wang et al. (2020) is available at https://zenodo.org/record/3833233 on a regular $1°\times1°$ grid. Apart from a format conversion from the provided .mat file to netCDF, this dataset was used without pre-processing.

The scripts used to process and plot the data are available upon request to the author. All plots presented in this paper have been produced with NCL v6.6.2.

## Appendix A: Methods

### A1  Model data processing

 ### A1.1  Global analysis

The four studied models use tripolar grids in the oceans, none of the four being common to each other. Conversely, the climatologies are provided on regular $1° \times 1°$ grids. Thus, using a common grid is required for all model-climatology comparisons, and MMM computation. The remapping is achieved using CDO (version 1.9.8: Schulzweida, 2019). While a conservative interpolation (operator remapcon) would be the most suited method to handle concentration or flux variables, it fails when applied to tripolar grids. Thus, we turned towards a distance-weighted interpolation (operator remapdis). A minor issue arises after remapping ocean data, with pixels defined inland along the coastlines. Thus, a land-sea mask is systematically applied after any remapping step, in order to keep the surface integral correct. The relative error generated by these data processing steps has been evaluated to be in the $\pm 0$–2 % range.

The multimodel ensemble mean (MMM) is computed in pixels where at least three models have valid data. This criterion has been chosen so that no pixel in MMM is based on a single model (for instance, the Caspian Sea is only described in UKESM1-0-LL), but to retain areas which would be described in all but one model (the Red Sea and Persian Gulf are not described in the ocean model of MIROC-ES2L).

Readers may note differences in metrics provided in this article and metrics provided in other literature (e.g., mean and median of the L11 climatology displayed in Table 3 and that in Galí et al. (2018, Sect. 3.2.1)). We checked the numbers we present and we are confident in our processing done with the CDO and NCL tools. Note that differences in median values may arise as we compute area-weighted medians.

### A1.2  Arctic analysis

In the study of Galí et al. (2019), the mean DMS concentration and flux is computed on ice-free pixels only, since both are derived from satellite observations. In order to provide comparable results, we also computed the mean over ice-free pixels only (with the same criterion as Galí et al. (2019) of >90 % free-water) along with the regular area-weighted mean. We used monthly sea-ice concentration datasets from each model to compute the means over ice-free pixels. We paid special attention to the calculation of the mean, which is thus done on 3D arrays (2D in space and 4 months per year) with missing data in pixels where the sea-ice cover exceeds the chosen threshold. The mean is calculated in a single step using a weighting that combines pixel area and available months duration (if any).

A special case arose for NorESM2-LM, for which the ocean grid has one more row (j=385) than the sea-ice grid (j=384). In order to match both grids to apply a sea-ice mask over DMS data, we were advised (Yanchun He, Mats Bentsen, personnal communication, June 2020) to remove the last row using cdo selindexbox,1,360,1,384. When doing so, a warning is displayed since both (resized) grids do not share exactly the same coordinates.

In Fig. 15, the central basin in the Arctic is defined by Ardyna et al. (2013), which is a revised partitioning based on the work by Spalding et al. (2007) and the WWF (World Wildlife Fund) agency. The mask file (in netcdf format) of this partitioning was provided by M. Galí.

## A2 Longhurst oceanic provinces

As presented in Sect. 2.2.1, the climatology of Lana et al. (2011) is built upon a "first-guess analysis", which consists of partitioning all available in-situ data into ocean biogeographic provinces. The partitioning used by Lana et al. (2011) is that proposed by Longhurst (2007) which consists of 54 provinces. A shapefile defining these provinces is available at https://www.marineregions.org/downloads.php#longhurst. Note that another version of this global ocean partitioning, including 56 provinces (plus a 57[th] one located in the Chesapeake Bay) is often referred to, and corresponds to an older version (Longhurst et al., 1995). However, most provinces are identical in both versions, and only a handful of provinces differ (two provinces along the Californian coast, OCAL and CCAL, are merged in a single province in the latest version, and the partitioning of the northern North Pacific differs).

We are aware of more recent work about ocean partitioning into biomes that provides both updated static partitioning and dynamic partitioning (Reygondeau et al., 2013; Fay and McKinley, 2014). Such dynamic partitioning accounts for yearly variation of the environmental and biogeochemical characteristics of the oceans, and thus offers a more accurate representation of the biomes. However, since our study uses the climatology developed by Lana et al. (2011) as the main reference, we decided to keep the same static partitioning for the biomes-based analysis.

*Author contributions.* JB, MM, PN and RS designed the study; JB performed research and analyzed data; JB, MM and RS wrote the paper with significant inputs from all coauthors.

*Competing interests.* The authors declare no competing interests.

*Acknowledgements.* We gratefully acknowledge Martí Galí, Nadja Steiner, and an anonymous reviewer for their comments that helped to improve and clarify this paper.

This work was supported by the European Union's Horizon 2020 research and innovation program with the CRESCENDO project under the grant agreement No 641816, the TRIATLAS project under the grant agreement No 817578. JB thanks M. Galí for his help in using his climatology of DMS and for providing the map of Arctic regions. JB thanks W. Wang for his help in using his climatology of DMS. JB thanks Y. He and M. Bentsen for their advise in using NorESM2-LM datasets. JT and JS acknowledge Research Council of Norway funded projects KeyClim (295046) and Columbia (275268). Simulations of MIROC-ES2L are supported by the TOUGOU project "Integrated Research Program for Advancing Climate Models" (grant number: JPMXD0717935715) of the Ministry of Education, Culture, Sports, Science, and

Technology of Japan (MEXT). MIROC-team acknowledge JAMSTEC for use of the Earth Simulator supercomputer. JPM was supported by the Met Office Hadley Centre Climate Programme funded by BEIS and Defra (grant no. GA01101).

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
