# Peer review of "Evaluation of ocean dimethylsulfide concentration and emission in CMIP6 models"

_Biogeosciences, 2020_

## Referee Comment (RC1)

**Manuscript "Evaluation of ocean dimethylsulfide concentration and emission in CMIP6 models" by Bock and coauthors**

Review by Martí Galí

**GENERAL COMMENTS**

The article by Bock an coauthors gives a comprehensive overview of sea-surface DMS concentrations and sea-air fluxes in four CMIP6 models, which feature DMS parametrizations of different complexity (two diagnostic and the other two prognostic). The models also differ in their coupling (or lack thereof) between marine DMS emission and atmospheric chemistry and aerosol/cloud radiative forcing, although this aspect is not dealt with in the discussion. The main conclusions of the article are (1) the uncertainty in present-day model estimates, with spatial patterns and seasonal cycles that differ from observational products in many oceanic regions, and (2) the diverging trends in global DMS emissions in end-of-century projections. These divergences reflect the different factors that drive DMS concentrations in each model, rather than the complexity of the parametrizations. The findings reported by Bock and coworkers will be useful to advance the modelling of marine DMS cycling because they highlight the mixed success in representing DMS cycling appropriately in CMIP numerical models and, more generally, the gap between experimental/observational knowledge and model parametrizations.

Below I summarize my main criticisms, with the hope that they will help improve the manuscript:

**1. On the potential overestimation of global DMS concentration by the Lana et al. (2011) climatology, and the failure to capture extremes**. The comparison between ESM results, global DMS climatologies based on statistical relationships and EO datasets (G18, W20), and the L11 climatology based on objective interpolation of in situ data, indicates greater consistency between the former two. The authors conclude that L11 overestimates DMS globally, a conclusion that is supported by previous works (Tesdal et al. 2016; Galí et al, 2018). Although the conclusions of the latter studies still hold over many regions, here I would like to note that all model and statistical representations of sea-surface DMS fields fail to account for extreme DMS concentrations. As the authors note, extreme concentrations were removed to compute the L11 climatology, although they were not systematically identified as measurement artefacts. Webb et al. (2019, SciRep) showed the importance of extreme DMS events in Antarctica. Recently, Bell et al. (2020, Frontiers) showed that both the L11 climatology and the observations-based G18 and W20 diagnostics failed to capture high-DMS events in the North Atlantic. In the light if these findings, the message that L11 overestimates DMS globally should be nuanced. Obviously, there is a dichotomy between the ability to capture the mean state and the extremes, and the latter might play an important role in ocean-atmosphere interactions.

**2. Global NPP-DMS relationship**. Understanding the global NPP-DMS relationship would be useful to place emergent constraints on present-day and future DMS emission. The attempt made in this article to pinpoint this relationship is very welcome, but the discussion of the underlying factors is poor. First, I would not expect that studies covering small spatiotemporal scales (e.g. those cited in L601) could give relevant insights into the NPP-DMS relationship over multiyear periods at the biome scale. Perhaps the work that addressed this issue more explicitly was that of Kameyama et al. (2013, GRL. Strong relationship between dimethyl sulfide and net community production in the western subarctic Pacific); but they related DMS to NCP, not NPP. I also recommend the work of Osman et al. (2019, Nature. Industrial-era decline in subarctic Atlantic productivity). Second, the discussion of the NPP-DMS relationship disregards the complexity of food-web and abiotic processes that control DMS concentrations. The framework proposed by Galí and Simó (2015, GBC) could be useful to understand the contribution of different factors to DMS variability in prognostic models.

**3. Relevant literature**. I suggest that authors to dig deeper into the non-modelling literature, which is connected to the point above. The article currently gives the impression that the authors are not familiar enough with some aspects of the DMS and DMSP biogeochemistry because imprecise informations are scattered in the text (see specifics). Suggestions of relevant articles can be found through this review. In addition, I strongly suggest the authors to pay more attention to the body of modelling literature produced by the group of Elliott and colleagues:

Wang et al. (2015, JGR). Influence of explicit *Phaeocystis* parameterizations on the global distribution of marine dimethyl sulfide.

Wang et al. (2018, Biogeochemistry). Influence of dimethyl sulfide on the carbon cycle and biological production.

Xu et al. (2016, JGR). DMS role in ENSO cycle in the tropics.

Another important article that explains the different ability of models to decouple DMS from phytoplankton biomass is:

Le Clainche et al. (2010, GBC). A first appraisal of prognostic ocean DMS models and prospects for their use in climate models

**4. Extrapolation of DMS emission to an ice-free Arctic summer**. A correction is needed here. In the G19 paper we warned that extrapolations towards a 100% ice-free Arctic summer shouldn't be made using the pan-Arctic linear regression between DMS emission and ice-free extent. This would amount to assuming that the % contribution from each subregion will not change in the future, which we know is not true. The Atlantic sector can't contribute much more than it presently does because it already is mostly ice free; on the contrary, the Central Arctic is far from being ice-free in summer and has the lowest DMS concentrations (Uhlig et al. 2019, Frontiers), thus the lowest potential for an increase in emissions. Of course, this is illustrated by the fact that the sums of regional extrapolations do not equal the pan-Arctic extrapolation in Table 9. I argue that, to compute the range of future ice-free Arctic DMS emissions, only the sum of regional contributions in each model should be used. I also recommend the study of Hayashida et al. (2020, GBC. Spatiotemporal Variability in Modeled Bottom Ice and Sea Surface Dimethylsulfide Concentrations and Fluxes in the Arctic During 1979–2015) who used a regional model with higher resolution to estimate contemporary DMS emission from the Arctic.

**SPECIFIC COMMENTS**

L23: perhaps a "by-product of microbial food webs" is more accurate, given the important roles of heterotrophic processes (grazing, bacterial catabolism) in DMS production. In other words, DMS production can't be understood from phytoplankton processes alone (which is different from saying that it cannot be predicted from phytoplankton variables…).

L25: please don't forget MSA, a product of the addition pathway that can be produced in relevant proportions compared to sulfuric acid.

L30: the original CLAW hypothesis paper (Charlson et al., 1987, Nature) should be given credit here. Regarding the relationship between DMS and downwelling irradiance, please consider citing Vallina and Simó (2007, Science).

L87: This is inaccurate. I suggest: "Dissolved DMSP is then converted to DMS with yields that increase with bacterial nutrient stress". Note: PISCES assumes all DMS production arises from dissolved DMSP, unlike other models, and contrary to observations. To compensate for this, PISCES requires "bacterial" DMS yields that range between 40 and 60% (Belviso et al., 2012), which are clearly too high (Galí and Simó, 2015).

Table 1: In my opinion, this table should cite the original articles where the prognostic or diagnostic DMS models were described. In the case of the diagnostic ones it is very simple:
- MIROC uses Aranami and Tsunogai (2004)
- UKESM uses Anderson et al. (2001)
This may be more difficult in the case of prognostic models that have seen incremental development. In this case, I suggest citing in this table the most recent papers where each prognostic model was described.

Section 2.1.1: I did a little research on the prognostic sulfur modules.
- PISCES: As the authors describe in this section, the original sulfur module of Bopp et al. (2008) was updated by Belviso et al. (2012) based on the model PlankTOM5 of Vogt et al. (2010). This detail should be included.

L187: This is true but the explanation is inaccurate. What is right-skewed is the distribution of satellite-retrieved Chl concentration matched to the DMS database, compared to the global distribution (PDF) of satellite Chl. As the authors correctly point out in the following sentence, this is related to preferential sampling of productive waters that probably had higher-than-average DMS. Ultimately, this could partially explain the right skewness of the in situ measurements when compared to the global DMS fields estimated with the G18 algorithm. But the latter may also suffer from biases caused by the fitted equations and satellite observations.

L217: besides sea ice, what limits satellite ocean colour measurements (passive radiometry) at high latitudes in winter is the low solar elevation. In December, no reliable observations are available north of about 48 degrees, in November and January the boundary is slightly above 50 degrees, etc.

L306-308: I disagree here. For unknown reasons, models struggle to capture high DMS in regions like the NE Pacific (station PAPA) and around Antarctica. In the NE Pacific, occurrence of very high DMS (often >15 nM) in late summer has been extensively documented by the Line P program and by many studies from Philippe Tortell's group. The article of Steiner et al. (2012, Biogeochemistry) explored potential reasons. This was also discussed by Galí et al. 2018. In the Southern Ocean, *Phaeocystis antarctica* blooms can results in seawater DMS of several tens nM, e.g. del Valle et al. (2009, L&O), Webb et al. (2019, SciRep). Highest DMS concentrations, many of which measured around Antarctica, were removed by Lana et al. (2011) before computing their climatology. So there's a general failure at capturing extreme DMS concentrations under certain conditions. See general comments.

L324: Note that the high DMS in winter and spring at high northern latitudes in L11 could be an interpolation artifact, as explained by Hayashida et al. (2020, GBC). This is relevant to BPLR panel in Fig. 5.

L349: "six biomes", but only four are listed. Perhaps specify that N and S polar and westerlies biomes are treated separately (but not trades or coastal ones).

L362: Well, low correlations are expected in areas where DMS has low seasonal amplitude. This also applies to the paragraphs below. OK, this was answered in L383.

L372: Looking at Fig. 5 province by province, and at Table 5, my first impression is that NORESM usually does better at capturing seasonal cycles, except for the trades biome. Also, my gut feeling is that this model better captures summertime DMS maxima at subtropical to temperate latitudes (the DMS summer paradox; check Simó & Pedrós-Alió 1999, Nature; LeClainche et al. 2010, GBC; Galí and Simó 2015, GBC), at least in regions I know best (4-NADR, 6-NASW, 16-MEDI).

Figure 7: please specify somewhere (figure or caption) that CAMS uses the N00 parameterization.

L480-481: I tend to agree with Tesdal here. See general comments.

L565: "regions or processes"

L585: "modelling works". Otherwise a large body of non-modelling literature would be disregarded.

L601-606: The discussion of the relationship between NPP and DMS is poor, and misses some relevant literature. DMS production is a food web process, not just a phytoplankton process. See the general comments.

Figure 14: Axis labels are too small, and some horizontal reference lines (or a grid) would be very useful to guide comparisons among models.

L641-648: It is important to note that the non-Atlantic sector includes the Siberian shelves, which seem to be quite productive owing to nutrient inputs from large rivers, recycling on the shelves and coastal erosion (Terhaar et al., 2021, NatComm). However, the satellite data used in G19 may be biased high in the Siberian shelves due to optical interference of continental materials (which was also pointed out by Hayashida et al., 2020). So I would dare to say that uncertainty in satellite DMS is much higher in the non-Atlantic sector, and that ESMs possibly struggle to capture the biogeochemical functioning in shallow Arctic seas, due to both too-low resolution and non-represented processes.

L665: Since emissions arise mostly from ice-free areas in both satellite and ESM asessments, additional factors must be invoked to explain scatter in the ice extent vs. DMS emission relationships. I am pretty sure that lower R2 in models results from too-low interannual variability in models compared to satellite observations. In G19, we pointed out that after 2011 interannual variability was controlled mostly by ocean productivity, not ice extent. This was further analyzed, and confirmed, by Lewis et al. (2020, Science).

Figure 15 and the related analysis: see general comments.

L687: Please beware that Le Clainche et al. (2010) did a DMS model intercomparison.

L694: agree on sampling biases, disagree on global overestimation. See general comments.

L701: ...which has been known for a long time (see Le Clainche et al., 2010 and references thererin)

**Typos, technical corrections**
L23: space after (DMS)
L26: something is missing: "formed DMS"
L116: open parentheses before "Simó and Dachs"
L212-215: please consider breaking this sentence with a period somewhere.
L277: "maxima", not "maximums"
L376: hypotheses
L576: "at play"?

---

## Author Comment (AC1)

**Review 1: Martí Galí**

Dear Martí Galí, We want to thank you for your very thorough review of our article. Your suggestions for adding clarifications, precisions and references throughout the manuscript will help improve it. Please find below our responses to your comments, which appear below in italics. Our proposed amendments to the text of our paper appear in blue. Page and line numbers refer to the version of the paper you reviewed.

**GENERAL COMMENTS**

*The article by Bock an coauthors gives a comprehensive overview of sea-surface DMS concentrations and sea-air fluxes in four CMIP6 models, which feature DMS parametrizations of different complexity (two diagnostic and the other two prognostic). The models also differ in their coupling (or lack thereof) between marine DMS emission and atmospheric chemistry and aerosol/cloud radiative forcing, although this aspect is not dealt with in the discussion. The main conclusions of the article are (1) the uncertainty in present-day model estimates, with spatial patterns and seasonal cycles that differ from observational products in many oceanic regions, and (2) the diverging trends in global DMS emissions in end-of-century projections. These divergences reflect the different factors that drive DMS concentrations in each model, rather than the complexity of the parametrizations. The findings reported by Bock and coworkers will be useful to advance the modelling of marine DMS cycling because they highlight the mixed success in representing DMS cycling appropriately in CMIP numerical models and, more generally, the gap between experimental/observational knowledge and model parametrizations.*

*Below I summarize my main criticisms, with the hope that they will help improve the manuscript:*

1. *On the potential overestimation of global DMS concentration by the Lana et al. (2011) climatology, and the failure to capture extremes.*

   *The comparison between ESM results, global DMS climatologies based on statistical relationships and EO datasets (G18, W20), and the L11 climatology based on objective interpolation of in situ data, indicates greater consistency between the former two. The authors conclude that L11 overestimates DMS globally, a conclusion that is supported by previous works (Tesdal et al. 2016; Galí et al, 2018). Although the conclusions of the latter studies still hold over many regions, here I would like to note that all model and statistical representations of sea-surface DMS fields fail to account for extreme DMS concentrations. As the authors note, extreme concentrations were removed to compute the L11 climatology, although they were not systematically identified as measurement artefacts. Webb et al. (2019, SciRep) showed the importance of extreme DMS events in Antarctica. Recently, Bell et al. (2020, Frontiers) showed that both the L11*

*climatology and the observations-based G18 and W20 diagnostics failed to capture high-DMS events in the North Atlantic. In the light if these findings, the message that L11 overestimates DMS globally should be nuanced. Obviously, there is a dichotomy between the ability to capture the mean state and the extremes, and the latter might play an important role in ocean-atmosphere interactions.*

Thank you for pointing out that one of the messages of our paper should be clarified. The references that you suggest are indeed useful to illustrate that neither the models nor the observation based products are currently able to reproduce the observed high DMS concentrations. As you say, one difficulty is to distinguish between a climatological mean state and extremes that are localised in space and time.

For instance, Bell et al. (2021) analyze that both the L11 climatology and diagnostics based on monthly satellite observations made over a given month fail to capture high DMS in situ measurements made over that same month in the North Atlantic. However, in this study, measurements from a given month of the year have been made only once, and one could question to which extent such high DMS concentration appear rather repeatedly throughout the years and could become a climatological feature. More observations are clearly needed. Bell et al. (2021) indicate that "The aim of this paper is to present an overview of the seawater DMS observations during NAAMES and some of the environmental factors that influence DMS variability." and later "The DMS climatology (Lana et al., 2011) captures the seasonal progression well but, unsurprisingly, does not accurately represent the substantial variability in DMS over short spatial/temporal scales in the North Atlantic." So clearly, from this paper one cannot conclude whether the high DMS concentration values measured during the NAAMES campaign are a climatological feature (that L11 fails to represent), or if the small spatial and temporal scale of these features are insufficient to imprint the climatological concentration.

The Webb et al. (2019) paper is somehow more affirmative in its conclusions for the West Antarctic Peninsula (WAP) coastal zone where DMS measurements have been made over a 5 year period. In this "highly productive region" (as designated by the authors), very high DMS concentration ("exceeding 30 nM in four out of five summer seasons") were measured. In particular Webb et al. (2019) conclude that "the L11 climatology is not accurately predicting WAP summer DMS production, and in particular is missing peak-DMS production events." While these measurements carried out over several years are likely representative of climatological values in this area, the extrapolation to the entire Longhurst APLR province to estimate a resulting flux, evaluated twice as large as that in L11, is more questionable. Overall, one important conclusion of this study is that elevated DMS concentration are associated to the sea-ice break-up. Since models do not include this driving factor, they unsurprisingly fail to reproduce the early summer maximum DMS concentration.

We propose to nuance and clarify our wording about the L11 climatology in three places as follows:

L185-189: "Thus, as pointed out by Tesdal et al. (2016), small scale features are transformed into large scale ones by the interpolation procedure. Tesdal et al. (2016) also suggested that the extrapolation of a small number of data points could lead to unrealistic distributions. This was confirmed by Galí et al. (2018, Sect. 4.1) who analysed some sources or errors and biases in L11, and pointed out that the distribution of DMS concentration is right-skewed as compared to DMS concentration derived from satellite chlorophyll measurements. These authors suggested that a preferential sampling of DMS-productive conditions could explain this positive bias."

replaced by:

"Thus, as pointed out by Tesdal et al. (2016), small scale features are transformed into large scale ones by the interpolation procedure, and anomalous values observed at local scale could induce bias when extrapolated across data-sparse regions. This is illustrated by Hayashida et al. (2020), who show that the entire Arctic region in L11 is based on extremely limited data (0–4 % areal coverage north of 60° N). The resulting extrapolation of open water DMS concentration to sea-ice covered areas, where primary production is presumably lower, may lead to a positive bias in L11. Another potential positive bias in L11 stems from the overrepresentation of biologically productive conditions in the in-situ DMS database from which L11 is built upon. This is supported by the study of Galí et al. (2018, Fig. 7 and Sect. 4.1) who pointed out that the distribution of DMS concentration in L11 is right-skewed as compared to DMS concentration derived from satellite chlorophyll measurements. Conversely, recent studies report on high DMS concentrations measured in the North Atlantic (Bell et al., 2021) and in a coastal station of the West Antarctic Peninsula or in the Ross Sea (Webb et al., 2019; del Valle et al., 2009, respectively) which are not represented in L11."

l. 308: "Conversely, the very high concentration displayed in L11 around Antarctica, and to a lesser extent in the Indian Ocean and south of Alaska, are not predicted by any model or by G18 or W20, suggesting that this could be a bias in the L11 climatology."

replaced by:

""Conversely, the very high concentration displayed in L11 around Antarctica, and to a lesser extent in the south of Alaska and in the Indian Ocean, are not predicted by any model nor by G18 or W20. For the former two regions, high concentrations have been reported in long-time measurements, at a site of the West Antarctic Peninsula, 2012-2017 period (Webb et al., 2019), and at the Ocean Station P in the North East Pacific, 1996-2010 period (Steiner et al., 2012) and 2005-2017 period (Galí et al., 2018). Further investigations would be required to explain these discrepancies between

measurements and models or climatologies. Some specific processes, such as the DMS concentration enhancement following sea-ice break-up (Webb et al., 2019) are not accounted for in the models, but are not sufficient to explain all discrepancies. Overall, assessing the relevance of high DMS events at the global scale and the spatial resolution of climate models is mandatory to improve them."

l. 694 "As concluded by previous authors (see for instance Galí et al., 2018, Sect. 4.1), the widely used L11 climatology likely overestimates surface DMS concentration due to sampling biases."

replaced by

"As concluded by previous authors (see for instance Galí et al., 2018, Sect. 4.1), the widely used L11 climatology likely overestimates climatological surface DMS concentration at the spatial resolution of climate models due to the combination of scarce and biased sampling."

2. *Global NPP-DMS relationship Understanding the global NPP-DMS relationship would be useful to place emergent constraints on present-day and future DMS emission. The attempt made in this article to pinpoint this relationship is very welcome, but the discussion of the underlying factors is poor. First, I would not expect that studies covering small spatiotemporal scales (e.g. those cited in L601) could give relevant insights into the NPP-DMS relationship over multiyear periods at the biome scale. Perhaps the work that addressed this issue more explicitly was that of Kameyama et al.(2013, GRL. Strong relationship between dimethyl sulfide and net community production in the western subarctic Pacific); but they related DMS to NCP, not NPP. I also recommend the work of Osman et al. (2019, Nature. Industrial-era decline in subarctic Atlantic productivity). Second, the discussion of the NPP-DMS relationship disregards the complexity of food-web and abiotic processes that control DMS concentrations. The framework proposed by Galí and Simó (2015, GBC) could be useful to understand the contribution of different factors to DMS variability in prognostic models.*

**NB: our response below also answers the question raised by Reviewer #2 in his/her specific comment L601.**

Thank you for raising this issue of emergent constraints and for pointing out that our references may not be fully suitable to support our analysis.

We fully agree that placing emergent contraints on a DMS-related field would be a very useful tool to the overall climate community. However this is beyond our ambition for this paper.

The ambition of our paper, quite specific here and that could appear modest in compararison to existing much more specialised litterature, is to broadly assess how some CMIP6 climate models behave in terms of DMS ocean surface concentrations and DMS fluxes, both in the current climate and in the rest of the century.

We are indeed lacking a large-scale observational database that would enable us to draw robust conclusions on the relationship between NPP and DMS concentrations, or emissions, at the scale of global oceans.

We have reworded this part of our text, noting, as you said, that a number of observational studies have highlighted such relationship, at a local scale though, and complementing these local-scale studies with the recent studies of Uhlig et al. (2019) and of Osman et al. (2019) that have been conducted at a basin scale.

However, we think that there are other lines of evidence, other than observations, on the existence of such relationship.

Firstly, previous modelling work of Bopp et al. (2003) and of Kloster et al. (2007) show that the response of the marine biology (i.e., declining NPP) is one of the prominent drivers of changes in DMS emissions. Although the current generation of the PISCES and HAMOCC models derive from previous model versions, key processes have been revised and updated. These changes have implications on model performances and on future projections as reported and documented in Séférian et al. (2020) and Kwiatkowski et al. (2020). In consequence, our work shines light of an emergent property of marine biogeochemical models linking changes in NPP and changes in DMS that is robust across model generations.

Secondly, factorial experiments conducted by Wang et al. (2020) using an artificial neural network show that a 10 % decrease of Chl $a$, a proxy for NPP, leads to a reduction in DMS concentration in large open-ocean domains.

We aknowledge though that a number of studies observed no correlation between DMS and Chl $a$, reflecting the complex mechanisms that control DMS concentrations and fluxes (e.g., Wang et al., 2020, and references therein).

We thus replaced the text : "Local in situ observations (e.g., Simó et al., 2002; Becagli et al., 2016) have shown positive correlations between NPP and DMSP, and the link between DMSP and DMS concentration has been described in several studies (e.g., Stefels, 2000; Yoch, 2002; Asher et al., 2017; Lizotte et al., 2017). The first group of models (CNRM-ESM2-1, NorESM2-LM and UKESM1-0-LL) thus captures a relationship which is consistent with such ocean field experiments, while the response simulated in MIROC-ES2L is not consistent with the current understanding of the DMSP production pathways by marine phytoplankton (Stefels et al., 2007)."

with

"Although the limited current knowledge about the NPP-DMSP-DMS relationships hampers our ability to constrain this emergent property, several lines of evidence tend to suggest that there is a positive correlation between NPP and DMS concentration. Firstly, noting that some studies

observed no correlation between DMS and Chl $a$ (e.g., Wang et al., 2020, and references therein), a number of other studies showed positive correlations between NPP and DMS production: the link between NPP and DMSP is highlithed at the local scale (e.g., Simó et al., 2002) and at a basin wide scale (e.g., Uhlig et al., 2019), that between NPP and DMS concentration again at a basin wide scale in Osman et al. (2019), and the link between DMSP and DMS concentration has been described in several studies (e.g., Stefels, 2000; Yoch, 2002; Asher et al., 2017; Lizotte et al., 2017). Secondly, factorial experiments conducted by Wang et al. (2020) using an artificial neural network show that a 10 % decrease of Chl a leads to a reduction in DMS concentration in large open-ocean domains. Finally, previous modelling work of Bopp et al. (2003) and of Kloster et al. (2007) show that the response of the marine biology (i.e., declining NPP) is one of the prominent drivers of changes in DMS emissions. The first group of models..."

The framework you describe in Galí and Simó (2015) and that one could apply to these CMIP6 models is largely beyond the scope of our article. Not to mention all the distinct variables involved in your analysis that are not part of the official CMIP6 data request, and thus are not available for a comparable analysis. However, we cite Galí and Simó (2015) in our conclusions as a way forward to progress in DMS climate modelling. The text at L742 now reads:

"Overall, our work shows that there is a major uncertainty in low-latitude ocean where the change in DMS concentration results from the interplay of marine biology factors with many other environmental drivers (e.g., temperature, salinity, stratification, nutrient availability, acidification, large-scale circulation), which and all may affect in both directions the trends in DMS concentration (Wang et al., 2020). Further analysis to disentangle the role of these factors is required, for instance along the lines of the meta-analysis of Galí and Simó (2015) that specifically addresses the issue of the "summer paradox". This would require important coordination among modellers to work in a multi-model perspective as only a few CMIP6 models include DMS and their DMS-related outputs are limited and insufficient at present to conduct such analysis. In turn, .... "

3. *Relevant literature*

   *I suggest that authors to dig deeper into the non-modelling literature, which is connected to the point above. The article currently gives the impression that the authors are not familiar enough with some aspects of the DMS and DMSP biogeochemistry because imprecise informations are scattered in the text (see specifics). Suggestions of relevant articles can be found through this review. In addition, I strongly suggest the authors to pay more attention to the body of modelling literature produced by the group of Elliott and colleagues:*

   *Wang et al. (2015, JGR). Influence of explicit Phaeocystis parameteriza-*

*tions on the global distribution of marine dimethyl sulfide.*

*Wang et al. (2018, Biogeochemistry). Influence of dimethyl sulfide on the carbon cycle and biological production.*

*Xu et al. (2016, JGR). DMS role in ENSO cycle in the tropics.*

*Another important article that explains the different ability of models to decouple DMS from phytoplankton biomass is:*

*Le Clainche et al. (2010, GBC). A first appraisal of prognostic ocean DMS models and prospects for their use in climate models*

We added precisions and references throughout the text, following your multiple suggestions made below (see our responses below to your specific comments). Thank you for these suggestions. Based on the references you cite in this general comment we have added the following sentence in the text of the introduction: l. 47 "...allowing an assessment of the recent evolution of DMS in this region (Galí et al., 2019). These advances coincide also with those of global models, from ocean biogeochemistry ones (Le Clainche et al., 2010; Séférian et al., 2020) to full ESM ones enabling investigations on either (i) the physical factors that impact DMS behaviour, for instance Xu et al. (2016) demonstrate that there seems to be a two-way interaction between DMS and ENSO in the tropical region, or (ii) the ecological factors, for instance representing in the model more explicitly diverse phytoplankton groups (e.g., *Phaeocystis*: Wang et al., 2015)."

4. *Extrapolation of DMS emission to an ice-free Arctic summer*

*A correction is needed here. In the G19 paper we warned that extrapolations towards a 100% ice-free Arctic summer shouldn't be made using the pan-Arctic linear regression between DMS emission and ice-free extent. This would amount to assuming that the % contribution from each subregion will not change in the future, which we know is not true. The Atlantic sector can't contribute much more than it presently does because it already is mostly ice free; on the contrary, the Central Arctic is far from being ice-free in summer and has the lowest DMS concentrations (Uhlig et al. 2019, Frontiers), thus the lowest potential for an increase in emissions. Of course, this is illustrated by the fact that the sums of regional extrapolations do not equal the pan-Arctic extrapolation in Table 9. I argue that, to compute the range of future ice-free Arctic DMS emissions, only the sum of regional contributions in each model should be used. I also recommend the study of Hayashida et al. (2020, GBC. Spatiotemporal Variability in Modeled Bottom Ice and Sea Surface Dimethylsulfide Concentrations and Fluxes in the Arctic During 1979–2015) who used a regional model with higher resolution to estimate contemporary DMS emission from the Arctic.*

—— We do not completely agree with this comment, for several reasons. First, in the multimodel approach of this study, the main message is about the spread of projected emissions at 100 % ice-free ocean, which is much larger than the difference, for a single model, between the pan-Arctic projected change and the sum of projected changes in the three areas. We do agree that the projected change in the entire pan-Arctic region is a simplification, since the relative emission in the sub-basins might change. However, this shortcoming also applies to a sub-basin, which could itself be further split in smaller parts, whose relative contribution may change. Nevertheless, in historical and ssp585 runs, most estimations are rather close. There are only 2 cases (CNRM and UKESM, historical) where there is an important difference between both projected changes: for CNRM, the sum of extrapolated changes is 17 % larger than the extrapolated change on the pan-Arctic, but for UKESM, it is smaller by $\sim 30\%$, which does not allow to conclude about a general rule. Lastly, and this is our main point: in your 2019 paper, the regional breakdown is justified by the contrasted biogeochemical regimes. In the models, at least those two with prognostic DMS parameterisation, the differential biogeochemical regime is not accounted for, and the sea-ice retreat dynamics in each regions is likely the main driver. Nevertheless, we noted that the multimodel projected range of DMS emission is larger when using the sum of projected changes, and is thus better to provide a likely estimate. We took your comment into account and now report future emissions as the sum of the regional contributions. The text now reads: l. 669 "In total, the CMIP6 summertime DMS emissions extrapolated at 100% sea-ice free water vary between 72 and 310 Gg S, enlarging the corresponding estimation of 144±66 Gg S in G19."

l. 677 "Extrapolations of annual DMS emissions at 100 % ice-free extent for the 2036-2100 period (from 86 to 282 Gg S for the pan-Arctic region) are comparable to projections inferred from the 1950–2014 period (72 to 310 Gg S).

We also cited the study of Hayashida et al. (2020) as suggested:

l. 682 "This modelled behaviour agrees well with the conclusions of Hayashida et al. (2020), who found that the decline of Arctic sea ice is associated with a quasi-linear positive trend of DMS flux.

**SPECIFIC COMMENTS**

- *L23: perhaps a "by-product of microbial food webs" is more accurate, given the important roles of heterotrophic processes (grazing, bacterial catabolism) in DMS production. In other words, DMS production can't be understood from phytoplankton processes alone (which is different from saying that it cannot be predicted from phytoplankton variables...).*

  We changed "a by-product of marine primary production" into " a by-product of microbial food webs ".

- *L25: please don't forget MSA, a product of the addition pathway that can be produced in relevant proportions compared to sulfuric acid.*

We agree that other products, such as MSA or DMSO, are produced in addition to $SO_2$. But as the intention in this introduction was to synthesize informaton on the DMS cycle in the atmosphere, and since the contribution of MSA to new particle formation is expected to be negligible in certain environment compared to that of $H_2SO_4$ (Bardouki et al., 2003) we did not mention MSA here. We reformulated the sentence into: "Once in the atmosphere, DMS is mainly oxidised into $SO_2$ and then gas-phase sulfuric acid, which rapidly condenses onto pre-existing aerosol particles..."

- *L30: the original CLAW hypothesis paper (Charlson et al., 1987, Nature) should be given credit here. Regarding the relationship between DMS and downwelling irradiance, please consider citing Vallina and Simó (2007, Science).*

We had not mentioned on purpose the original paper Charlson et al. (1987) for the "CLAW hypothesis" as it has been largely discussed and revisited since 1987. We chose to cite papers that include references to both the original paper and to follow-on papers. As suggested, we have added the reference Vallina and Simo (2007) in the text that now reads "DMS-climate feedback Vallina and Simo (2007); Carslaw et al. (2010); Quinn and Bates (2011))."

- *L87: This is inaccurate. I suggest: "Dissolved DMSP is then converted to DMS with yields that increase with bacterial nutrient stress". Note: PISCES assumes all DMS production arises from dissolved DMSP, unlike other models, and contrary to observations. To compensate for this, PISCES requires "bacterial" DMS yields that range between 40 and 60% (Belviso et al., 2012), which are clearly too high (Galí and Simó, 2015).*

Thank you for the suggestion and additional explanation, we have modified the text accordingly.

- *Table 1: In my opinion, this table should cite the original articles where the prognostic or diagnostic DMS models were described. In the case of the diagnostic ones it is very simple: - MIROC uses Aranami and Tsunogai (2004) - UKESM uses Anderson et al. (2001) This may be more difficult in the case of prognostic models that have seen incremental development. In this case, I suggest citing in this table the most recent papers where each prognostic model was described.*

Thank you for your suggestion on references, but we chose to keep on with the latest references that refer to the exact versions of the models we used in our analysis (CMIP6 version). Indeed these references include both past references, possibly original references, as well as the descriptions of the evolutions implemented, or not, in the CMIP6 version (see for instance the Sellar et al. (2019) paper that provides details concerning their tuning of the Anderson et al. 2001 parametrization).

- *Section 2.1.1: I did a little research on the prognostic sulfur modules.- PISCES: As the authors describe in this section, the original sulfur module of Bopp et al. (2008) was updated by Belviso et al. (2012) based on the model PlankTOM5 of Vogt et al. (2010). This detail should be included.*

  Thank you for the time you spent to look for additional information of the modules. We have added what you propose. The text now reads (l. 83): "... and updated by Belviso et al. (2012) based on the PlankTOM5 model of Vogt et al. (2010).".

- *L187: This is true but the explanation is inaccurate. What is right-skewed is the distribution of satellite-retrieved Chl concentration matched to the DMS database, compared to the global distribution (PDF) of satellite Chl. As the authors correctly point out in the following sentence, this is related to preferential sampling of productive waters that probably had higher-than-average DMS. Ultimately, this could partially explain the right skewness of the in situ measurements when compared to the global DMS fields estimated with the G18 algorithm. But the latter may also suffer from biases caused by the fitted equations and satellite observations.*

  We have changed this, see the answer to the general comment #1.

- *L217: besides sea ice, what limits satellite ocean colour measurements (passive radiometry) at high latitudes in winter is the low solar elevation. In December, no reliable observations are available north of about 48 degrees, in November and January the boundary is slightly above 50 degrees, etc.*

  We rephrased the sentence that now reads: "Another limitation of this approach is the lack of satellite observations over sea-ice and at low solar elevations, resulting in observational gaps in high latitudes ($> 48°$) in winter."

- *L306-308: I disagree here. For unknown reasons, models struggle to capture high DMS in regions like the NE Pacific (station PAPA) and around Antarctica. In the NE Pacific, occurrence of very high DMS (often ¿15 nM) in late summer has been extensively documented by the Line P program and by many studies from Philippe Tortell's group. The article of Steiner et al. (2012, Biogeochemistry) explored potential reasons. This was also discussed by Galí et al. 2018. In the Southern Ocean, Phaeocystis antarctica blooms can results in seawater DMS of several tens nM, e.g. del Valle et al. (2009, L&O), Webb et al. (2019, SciRep). Highest DMS concentrations, many of which measured around Antarctica, were removed by Lana et al. (2011) before computing their climatology. So there's a general failure at capturing extreme DMS concentrations under certain conditions. See general comments.*

  We changed this according to your suggestions, see the answer to the general comment #1.

- *L324: Note that the high DMS in winter and spring at high northern latitudes in L11 could be an interpolation artifact, as explained by Hayashida et al. (2020, GBC). This is relevant to BPLR panel in Fig. 5.*

  We have taken note of your remark, and the text now reads: "UKESM1-0-LL shows a northern maximum starting earlier than the other models, in agreement with L11. However, as pointed out by Hayashida et al. (2020), this feature could be an extrapolation artefact in L11 (see Sect. 2.2.1)."

- *L349: "six biomes", but only four are listed. Perhaps specify that N and S polar and westerlies biomes are treated separately (but not trades or coastal ones).*

  We reformulated into "six biomes (polar N and S, westerlies N and S, trades and coastal)"

- *L362: Well, low correlations are expected in areas where DMS has low seasonal amplitude. This also applies to the paragraphs below. OK, this was answered in L383.*

  This is indeed analysed later on in the paper. We thought about reorganising this section so that the analysis follows more closely the description, but we keep the original version in the end.

- *L372: Looking at Fig. 5 province by province, and at Table 5, my first impression is that NORESM usually does better at capturing seasonal cycles, except for the trades biome. Also, my gut feeling is that this model better captures summertime DMS maxima at subtropical to temperate latitudes (the DMS summer paradox; check Simó & Pedrós-Alió 1999, Nature; LeClainche et al. 2010, GBC; Galí and Simó 2015, GBC), at least in regions I know best (4-NADR, 6-NASW, 16-MEDI).*

  It is indeed an impression that figures confirm or not. NorESM is indeed the best in provinces 4-NADR and 6-NASW, but the other models are rather good as well. In the 16-MEDI region MIROC has the best score, while still in these subtropical to temperate latitudes, in the 34-NPPF region MIROC again has the best score or in the 18-NASE region scores are the same for NorESM and CNRM-ESM. More generally, over the 54 provinces, NorESM2-LM has the best correlation over 18 regions, MIROC-ES2L in 15 regions, CNRM-ESM2-1 in 13 regions and UKESM1-0-LL in 10 regions. However, such piece of information is not really valuable, since provinces have very different weights (either considering their size or the corresponding emission). So, for the sake of balance between details and generalities we decided not to extend further our analysis of the results presented in Figure 5.

- *Figure 7: please specify somewhere (figure or caption) that CAMS uses the N00 parameterization.*

  We changed in the figure "CAMS" into "CAMS19". The text of the paper explicitly says that CAMS19 uses the N00 parameterisation (Sect. 2.2.2).

- *L480-481: I tend to agree with Tesdal here. See general comments.*

  We have modified our text that now reads: "Accounting only on the four models, the median flux (1980–2009) is $19 \pm 3$ Tg S year $^{-1}$, thus lowering the best estimate of Tesdal et al. (2016) by 10% but with an identical uncertainty range. Overall, we somewhat contradict Tesdal et al. (2016) who conclude in a low bias of model DMS fluxes. Indeed, apart from the CAMS19 and the L11/Nightingale et al. (2000) DMS flux estimates, the other current observational estimates coincide with our CMIP6 model estimate." However, the current grid size of ocean models and additional processes that are not accounted for, such as DMS enhancement during sea-ice break-up (see Sect. 3.1.1), prevent models from accounting for high DMS events localised in space and time. Thus, simulated DMS emission might represent the lower bound of actual fluxes.

- *L565: "regions or processes"L585: "modelling works". Otherwise a large body of non-modelling literature would be disregarded.*

  Thank you for the precision, we modified accordingly.

- *L601-606: The discussion of the relationship between NPP and DMS is poor, and misses some relevant literature. DMS production is a food web process, not just a phytoplankton process. See thegeneral comments.*

  We have enriched the discussion as described in response to your general comment above.

- *Figure 14: Axis labels are too small, and some horizontal reference lines (or a grid) would be very useful to guide comparisons among models.*

  We have added reference lines to help reading of the figure, and increased the labels font size.

- *L641-648: It is important to note that the non-Atlantic sector includes the Siberian shelves, which seem to be quite productive owing to nutrient inputs from large rivers, recycling on the shelves and coastal erosion (Terhaar et al., 2021, NatComm). However, the satellite data used in G19 may be biased high in the Siberian shelves due to optical interference of continental materials (which was also pointed out by Hayashida et al., 2020). So I would dare to say that uncertainty in satellite DMS is much higher in the non-Atlantic sector, and that ESMs possibly struggle to capture the biogeochemical functioning in shallow Arctic seas, due to both too-low resolution and non-represented processes.*

  Thank you for the explanations. We have modified the text following your wording. It now reads: "Galí et al. (2019) further discussed differences in biogeochemical and meteorological characteristics in the two Arctic sub-sectors to explain why DMS concentration is larger in the non-Atlantic sector. In particular, as the non-Atlantic sector includes the Siberian shelves, which seem to be quite productive owing to nutrient

inputs from large rivers (Terhaar et al., 2021), the G19 data may be biased high in the Siberian shelves due to optical interference of continental materials (Hayashida et al., 2020). While uncertainty in satellite DMS appears higher in the non-Atlantic sector, ESMs possibly struggle to capture the biogeochemical functioning in shallow Arctic seas, due to both too-low resolution and non-represented processes. Notwithstanding the biases in models as compared to G19, only MIROC-ES2L and NorESM2-LM correctly capture this difference between sectors with higher DMS concentration in the non-Atlantic sector."

- *L665: Since emissions arise mostly from ice-free areas in both satellite and ESM asessments, additional factors must be invoked to explain scatter in the ice extent vs. DMS emission relationships. I am pretty sure that lower R2 in models results from too-low interannual variability in models compared to satellite observations. In G19, we pointed out that after 2011 interannual variability was controlled mostly by ocean productivity, not ice extent. This was further analyzed, and confirmed, by Lewis et al. (2020, Science).*

  Thank you for pointing that out.We modified our text that now reads: "Over the pan-Arctic region, determination coefficients ($R^2$) are largely higher in all CMIP6 models than those of G19. This reflects the linear dependence of the flux to the free-water fraction in the models, though in observations additional factors such as ocean productivity can be invoked to explain scatter in the DMS emission versus sea-ice extent relationship (Galí et al., 2018; Lewis et al., 2020). Smaller interannual variability in models compared to satellite observations can also contribute to higher $R^2$."

- *Figure 15 and the related analysis: see general comments.*

  See our response to the general comments.

- *L687: Please beware that Le Clainche et al. (2010) did a DMS model intercomparison.*

  We rephrased our sentence so the context of our study is more clearly presented, and we added the Le Clainche et al. 2010 reference. The text now reads: "...and while they may have already been evaluated, either in a previous (e.g., Le Clainche et al., 2010) or in their current version, it is the first time that this is done in a common coupled atmosphere-ocean simulation framework."

- *L694: agree on sampling biases, disagree on global overestimation. See general comments.*

  See our response and amended sentence in general comment #1.

- *L701: ...which has been known for a long time (see Le Clainche et al., 2010 and references thererin)*

We added this information and the text now reads: "models better reproduce the annual cycles in mid to high latitudes (polar and westerlies biomes) than in low latitudes (trades biomes), in agreement with past studies (e.g., Le Clainche et al., 2010)."

**0.1 Typos, technical corrections**

- L23: space after (DMS)
  corrected

- L26: something is missing: "formed DMS"
  corrected: formed from

- L116: open parentheses before "Simó and Dachs"
  corrected

- L212-215: please consider breaking this sentence with a period somewhere.
  This sentence is left unchanged.

- L277: "maxima", not "maximums"
  Wee stick to "maximums", both are correct see: https://www.collinsdictionary.com/dictionary/english/m

- L376: hypotheses
  corrected

- L576: "at play"?
  corrected

[revised manuscript text omitted]

---

## Author Comment (AC2)

**Review 2**

We thank the second Reviewer for his/her useful comments, which help improving the manuscript. We reproduce below the comments in italics with our answers for each of them. The modified text is presented in blue. Line numbers refer to the first submitted version of the paper.

**General comments:**

- *This paper presents an evaluation of ocean DMS in CMIP6 models over the historical period, and discusses their projected changes by the late 21st century under SSP585. To my knowledge, no previous work on DMS has been done using CMIP6 models. Therefore, this paper provides useful insights into the current state of DMS represented in the latest generation of ESMs. I recommend publication after major revisions, addressing my general and specific comments below.*

  *The historical evaluation is very extensive, but maybe a bit too extensive to be included in the main text. I do not suggest to delete anything, but I do suggest to move some content into Supplementary Information (SI). One suggestion is to move Section 3.1.2 and associated figure/table (Figure 5 & Table 5) into SI. I particularly pick on this section because: (1) this section compares the models with L11 only, which is now considered to be outdated (i.e. G18 and W20 are better replacements); and (2) this section compares over small biogeographical regions, in which global models are not necessarily expected to perform well. I think Figure 3 is just sufficient for regional evaluation of these coarse-resolution models.*

  We carefully examined this suggestion to move Section 3.1.2 into SI. However, we believe that it must remain in main text for several reasons that we explain hereafter. First, this Section leads to an important conclusion that the model skills in reproducing observed seasonal cycles depend on the overall location, with better skills in mid and high latitudes regions (polar and westerly biomes) than in low latitudes regions (trade biome, and low latitude coastal provinces). Second, this Section introduces the methodology of analysis based on biogeographical provinces and biomes, which is further used in Section 4 (Figures 10 and 13). We believe that the presentation of the ocean partitioning (first paragraph of Section 3.1.2, coloured map in Figure 5) must be presented in the main text. Reviewer #2 points out that Figure 3 also provides a regional evaluation which is sufficient for our analysis. Figure 3 indeed presents a zonal average of DMS concentration. However, zonally averaged results do not allow a biome analysis such as that presented in Section 3.1.2, which brings useful insights about the model performance. We also highlight that as compared to Figure 3, Figure 5 and Table 5 provide an evaluation metric (correlation coefficients) which is helpful to draw the conclusions on factual elements. Last, Reviewer #2 also mention that "L11 (...) is now

considered to be outdated" to justify that Section 3.1.2 should be moved to SI. However, as stated in the last paragraph of this Section (l. 385-389), the same provinces- and biomes-based analysis carried out with W20 as the reference climatology, and provided in the SI, results in similar conclusions regarding the models skills. We thus believe that the comparison against L11 is meaningful and should be kept in main text.

- *In addition to the environmental variables considered in the paper, I suggest to consider three additional variables for analysis: pH, MLD, and SST. Projected changes in these variables might play a substantial or additive role, as they influence directly or indirectly the DMS concentration and flux, as parameterised in the models. The Arctic might have experienced greater changes in these variables, so it is worthwhile checking these variables.*

These three variables can indeed play a role on DMS concentration and flux, however a detailed sensitivity analysis of each of them is above the scope of this study. Furthermore, we believe that it would not necessarily be relevant to support the main conclusions of the paper, regarding the multimodel evaluation and assessment of future trends. To explain further our point, we first tackle the question of the relation of these three variables with the DMS concentration. The sensitivity study carried out by Wang et al. (2020, Fig. 9) shows a negative correlation between MLD and DMS concentration, which is quite straightforward (dilution effect, as explicitly accounted for in the parameterisation of Aranami and Tsunogai (2004) as used in MIROC-ES2L). In this sensitivity study by Wang et al. (2020), the effect of SST increase is more balanced and depends primarily on the latitude, with a decrease in DMS at low latitudes but an increase for higher latitudes, especially in the Southern Ocean. However, Wang et al. (2020) show that individual linear regressions for both variables explain very little ($< 10$ %) of the observed variance in DMS concentration (7 % for MLD, 2 % for SST). Thus, both variables are expected to have little predictive ability to infer the evolution of DMS concentration. To some extent, this is confirmed for SST, since all models unanimously predict an increase in ssp585 simulations, but disagree on the trend of DMS concentration. Conversely, the study of Wang et al. (2020) shows that SST and MLD are good predictors for DMS when they are associated to other biological and climatological variables in the neural networks. Regarding the pH, only CNRM-ESM2 accounted for a dependency of DMS concentration onto this variable, which prevents from any multimodel comparison.

Regarding the DMS emission to the atmosphere, we agree that there is a positive dependence of the flux on the SST, at least when a formulation of the Schmidt number for DMS as a function of temperature is used (such as that provided by Wanninkhof, 2014). We had already mentioned this effect in the paper at L504. For clarity, we added an additional Figure in the SI (Fig. SI-7) showing the timeseries of SST for both historical and ssp585 simulations as modelled by the four evaluated models. We updated

a sentence accordingly to refer to this new Figure (at L505): "This is likely explained by the positive dependence of the gas-exchange parameterisations on the SST, which is sharply increasing in ssp585 simulations (see the timeseries of modelled SST in Fig. SI-7)."

**0.1    Specific comments:**

- *L48: replace "last" with "latest".*

  The entire sentence has been changed.

- *L49: I'm not sure if "unprecedented" is an appropriate term here, considering that: (1) Tesdal et al. (2016) have incorporated more products (measurement/empirical/prognostic approaches) in their assessment; and (2) there are only 4 ESMs in CMIP6 that simulated ocean DMS. Has this number increased/decreased from CMIP5?*

  We agree with your comments. The entire sentence has been changed, and "unprecendented" is no longer used. To answer your question about the change between CMIP5 and CMIP6 models, interactive modelling of DMS is indeed a new feature in several CMIP6 models and no CMIP5 models include prognostic DMS (see Séférian et al., 2020, Table 2).

- *Sec.2.1.1: Given the dependence of DMSP (DMS) production rate on phytoplankton species, I suggest to list the types of phytoplankton and cellular quota (sulfur to carbon/CHL ratios) specified in all of the 4 models.*

  We chose not to provide additional details on the DMS production in this paper. Readers are refered to the specific references.

- *L98&L101: I'm confused about pH dependency. In L98, it says DMS release is computed as a function of pH. In L101, it says it has not been activated in CMIP6 runs. So which one is used for this paper? If it has not been activated, the word "pH" should be removed from L98.*

  Indeed, the pH dependency is implemented in NorESM2, but was not activated in CMIP6 runs. We removed "pH" at L98 as suggested. We also rephrased the sentence L100 from:
  "The tunable pH dependency was not present in the original parameterisation of Kloster et al. (2006), and has not been activated in CMIP6 runs (Tjiputra et al., 2020)"
  to:
  "Although a tunable pH dependency, that was not present in the original parameterisation of Kloster et al. (2006), has been implemented in NorESM2, it has not been activated in CMIP6 runs (Tjiputra et al., 2020)."

- *L142: Is there plan to publish the DMS data for MIROC-ES2L on ESGF nodes in the near future?*

MIROC-ES2L data have now been uploaded on ESGF nodes. We updated the sentence at L142 to remove the note about MIROC-ES2L data: "All datasets were downloaded from ESGF nodes, ." We also updated Table 2 and the relevant figures: all ancillary variables from MIROC-ES2L in ssp585 simulations now account for the same number of realisations (10).

- *L158: What about mixed layer depth (MLD)? Given its direct effect on DMS in diagnostic models, I think MLD should be assessed in addition to Chl.*

  Please see our answer to your general comment. Also, MLD is available for only 3 over 4 models, with limits the potential for a multimodel evaluation.

- *L159: In addition to these, I suggest to show pH of the models whose DMS depends on pH. The parameterisation of Six et al. (2016) has quite strong pH effect, so this might play an important role in some regions like the Arctic. Figures can go into SI.*

  See our answer to general comment. CNRM-ESM2-1 is the only model where pH dependency onto DMS concentration is implemented and used in CMIP6 simulations. However, the overall effect of this has been found to be limited.

- *L163: I am not sure if I get this correct. Is MMM calculated by averaging the ensemble means of the 4 models? Or is it calculated by averaging the ensembles of the 4 models (11 + 10 + 3 + 16 for historical)? I think it is the former, but it is not clear from this sentence.*

  The MMM is indeed calculated by averaging the ensemble means of the four models. We rephrased the sentence at L163 as follows:
  "In the following, the multi-model ensemble mean (hereafter MMM) is computed from the various model outputs using an equal weight for each model, irrespective of the number of realisations."
  now reads:
  "In the following, the multi-model ensemble mean (hereafter MMM) is calculated by averaging the ensemble means of the four models using an equal weight for each model, regardless of the number of realisations."

- *Figure 2: For readability, indicate in the caption whether these differences represent model-minus-obs or obs-minus-model.*

  We added "model minus climatology" in the caption.

- *L365: why is it "striking" that models do well in these regions?*

  We replaced "striking" by "noteworthy".

- *L392: Instead of text, it might be helpful to visualise the different wind-speed-based paramterisations used by these models. Consider creating a*

*simple plot like Figure 2 of Ho et al. (2006) (but do this for Schmidt number for DMS).*
*Ho et al. (2006): Measurements of air-sea gas exchange at high wind speeds in the Southern Ocean: Implications for global parameterizations, GRL, 10.1029/2006GL026817*

Thank you for pointing this specific figure in the Ho et al. (2006) paper. It is indeed very informative. However, although it appears fully appropriate in Ho et al. (2006) whose purpose is air-sea exchange parameterisation, it is not the case of our paper. We now refer to similar figures from the original papers. The modified version of our text now reads (L404):
"The reader is referred to Figures 12 of Nightingale et al. (2000) and Figure 2 of Wanninkhof (2014) to illustrate the relationships between wind speed and exchange coefficients for the three gas transfer parameterisations of interest here.

- *L410: In addition to wind, would temperature bias play a role in modifying flux via solubility/diffusivity? SST figures could be added to SI.*

  In the evaluated models, the solubility of DMS (reverse flux from the atmosphere to the ocean) is neglected. However, the diffusivity is indeed dependent onto SST, thus a temperature bias can affect the flux calculation. However, we evaluate that this bias is small as compared to the overall uncertainty of flux parameterisations, and to the large flux range obtained with various existing parameterisations. To give a rough estimation, using a 2 K SST bias, the difference in Schmidt number is around $\sim 5$ % at 20 °C, while the difference of emission depending on the flux parameterisation is in the 35–40 % range (see Tesdal et al., 2016, Fig. 7). We added a sentence at L404: A bias in modelled SST can thus contribute to the bias in flux calculation, but is estimated to be smaller than the uncertainty of flux parameterisation.

- *Figures 6&8: I suggest to add a subplot showing the results of Wang et al. (2020), which I assume are better obs-based products than L11/CAMS19? Without them, it just gives an impression that model are performing badly compared to the obs (L11/CAMS19). I think they compare better with Wang et al. (2020), and this point should be made clear in these figures.*

  We considered your suggestion but decided not to implement it for two reasons. First, we do not have in hand a gridded flux dataset from the G18 DMS concentration data, so including fluxes from W20 and not including fluxes from G18 would have been in our opinion bringing forward W20, which we have no reason for doing. Second, in agreement with previous studies such as that of Tesdal et al. (2016), our work shows that to first order, the global DMS emission depends on global mean DMS concentration, with a significant impact of the chosen flux parameterisation. Conversely, the patterns of DMS concentration are of second importance. Thus, while a detailed comparison of the fields of DMS concentration from

models and climatology is of major interest, it seems that a comparison of the flux fields is slightly redundant.

- *L465: The 4 CMIP6 models differ in the flux parameterisation, so the finding here does not confirm the conclusion of Tesdal et al. (2016) that global emission is roughly linearly dependent upon global mean concentration for "a given flux parameterisation".*

  Thank you for this remark. We modified the original text:
  "MIROC-ES2L has an intermediate emission value of 18.4 Tg S year$^{-1}$ which ranges in the low end of other studies also using the flux parameterisation of Nightingale et al. (2000). However, MIROC-ES2L has the lowest mean DMS concentration among the models, thus confirming the conclusion of Tesdal et al. (2016, Fig. 8) who show that global emission is roughly linearly dependant upon global mean concentration for a given flux parameterisation."
  to: "MIROC-ES2L has an intermediate emission value of 18.4 Tg S year$^{-1}$ which ranges in the low end of other studies also using the flux parameterisation of Nightingale et al. (2000). This is consistent with the rather low global mean DMS concentration in MIROC-ES2L (1.77 nM, Table 3), and the finding of Tesdal et al. (2016, Fig. 8) that to first order the global mean concentration of DMS determines the global mean flux."

- *L546: I recommend two papers from Wang et al. (2018), which incorporates perhaps more DMS producers than the 4 CMIP6 models, including Phaeocystis.*
  *Wang et al. (2018): Impacts of Shifts in Phytoplankton Community on Clouds and Climate via the Sulfur Cycle, Global Biogeochemical Cycles, 10.1029/2017GB005862*
  *Wang et al. (2018): Influence of dimethyl sulfide on the carbon cycle and biological production, Biogeochemistry, 10.1007/s10533-018-0430-5*

  Thank you for suggesting these references. The first one is mostly dedicated to an assessment of the radiative effect of the sulful cycle, and is not closely related to our study. Conversely, the second reference is well suited for the comparison. We added the following sentence at L556:
  Wang et al. (2018) performed simulations with the Community Earth System Model (CESM) with RCP8.5 scenario, which computes a global decrase in DMS flux of -8.1 % in 2100, with significant spatial variability. These findings are consistent with those of Kloster et al. (2007), and agree qualitatively with the results obtained with NorESM2-LM despite the projected DMS decrease is only half that found by Wang et al. (2018).

- *Figures 13,14,15: For understanding what each colour represents easily, could you plot a legend in one of the subplots? I know the colours are described in figure caption, but it is easier with a legend.*

  We have added a frame with a legend in Figs. 13, 14 and 15

- *L601: I think this paragraph deserves a bit more discussion. The strong relationship between DMS and Chl/NPP is probably true for a given phytoplankton species (and therefore, this replationship holds for in situ observations of a particular phytoplankton bloom or relatively simple-complexity phytoplankton models). However, should this really be the case at global scale where different phytoplankton species dominate in different regions and phytoplankton have a wide range of DMS production rates (i.e. cellular quota; Stefels et al. 2007)? I understand that this point leads to the conclusion in the subsequent paragraph, but I think the reality of the DMS-Chl/NPP relationship is highly variable regionally due to the diversity of phytoplankton species, which should be acknowledged.*

**NB: our response below also answers the question raised by Martí Galí in Review #1, General comment #2.**

The ambition of our paper, quite specific here and that could appear modest in compararison to existing much more specialised litterature, is to broadly assess how some CMIP6 climate models behave in terms of DMS ocean surface concentrations and DMS fluxes, both in the current climate and in the rest of the century.

We are indeed lacking a large-scale observational database that would enable us to draw robust conclusions on the relationship between NPP and DMS concentrations, or emissions, at the scale of global oceans.

We have reworded this part of our text, noting, as you said, that a number of observational studies have highlighted such relationship, at a local scale though, and complementing these local-scale studies with the recent studies of Uhlig et al. (2019) and of Osman et al. (2019) that have been conducted at a basin scale.

However, we think that there are other lines of evidence, other than observations, on the existence of such relationship.

Firstly, previous modelling work of Bopp et al. (2003) and of Kloster et al (2007) show that the response of the marine biology (i.e., declining NPP) is one of the prominent drivers of changes in DMS emissions. Although the current generation of the PISCES and HAMOCC models derive from previous model versions, key processes have been revised and updated. These changes have implications on model performances and on future projections as reported and documented in Séférian et al. (2020) and Kwiatkowski et al. (2020). In consequence, our work shines light of an emergent property of marine biogeochemical models linking changes in NPP and changes in DMS that is robust across model generations.

Secondly, factorial experiments conducted by Wang et al. (2020) using an artificial neural network show that a 10 % decrease of Chl $a$, a proxy for NPP, leads to a reduction in DMS concentration in large open-ocean domains.

We aknowledge though that a number of studies observed no correlation between DMS and Chl $a$, reflecting the complex mechanisms that control

DMS concentrations and fluxes (e.g., Wang et al., 2020, and references therein).

We thus replaced the text : "Local in situ observations (e.g., Simó et al., 2002; Becagli et al., 2016) have shown positive correlations between NPP and DMSP, and the link between DMSP and DMS concentration has been described in several studies (e.g., Stefels, 2000; Yoch, 2002; Asher et al., 2017; Lizotte et al., 2017). The first group of models (CNRM-ESM2-1, NorESM2-LM and UKESM1-0-LL) thus captures a relationship which is consistent with such ocean field experiments, while the response simulated in MIROC-ES2L is not consistent with the current understanding of the DMSP production pathways by marine phytoplankton (Stefels et al., 2007)."

with

"Although the limited current knowledge about the NPP-DMSP-DMS relationships hampers our ability to constrain this emergent property, several lines of evidence tend to suggest that there is a positive correlation between NPP and DMS concentration. Firstly, noting that some studies observed no correlation between DMS and Chl $a$ (e.g., Wang et al., 2020, and references therein), a number of other studies showed positive correlations between NPP and DMS production: the link between NPP and DMSP is highlithed at the local scale (e.g., Simó et al., 2002) and at a basin wide scale (e.g., Uhlig et al., 2019), that between NPP and DMS concentration again at a basin wide scale in Osman et al. (2019), and the link between DMSP and DMS concentration has been described in several studies (e.g., Stefels, 2000; Yoch, 2002; Asher et al., 2017; Lizotte et al., 2017). Secondly, factorial experiments conducted by Wang et al. (2020) using an artificial neural network show that a 10 % decrease of Chl a leads to a reduction in DMS concentration in large open-ocean domains. Finally, previous modelling work of Bopp et al. (2003) and of Kloster et al. (2007) show that the response of the marine biology (i.e., declining NPP) is one of the prominent drivers of changes in DMS emissions. The first group of models..."

The framework you describe in Galí and Simó (2015) and that one could apply to these CMIP6 models is largely beyond the scope of our article. Not to mention all the distinct variables involved in your analysis that are not part of the official CMIP6 data request, and thus are not available for a comparable analysis. However, we cite Galí and Simó (2015) in our conclusions as a way forward to progress in DMS climate modelling. The text at L742 now reads:

"Overall, our work shows that there is a major uncertainty in low-latitude ocean where the change in DMS concentration results from the interplay of marine biology factors with many other environmental drivers (e.g., temperature, salinity, stratification, nutrient availability, acidification, large-scale circulation), which and all may affect in both directions the trends

in DMS concentration (Wang et al., 2020). Further analysis to disentangle the role of these factors is required, for instance along the lines of the meta-analysis of Galí and Simó (2015) that specifically addresses the issue of the "summer paradox". This would require important coordination among modellers to work in a multi-model perspective as only a few CMIP6 models include DMS and their DMS-related outputs are limited and insufficient at present to conduct such analysis. In turn, .... "

- *L622: Briefly state what the conclusions are.*

  We left the sentence as is, as the rest of the section largely presents when CMIP6 models agree (or not) with the analysis of Galí et al. (2019).

- *L650: I don't really understand the latter part of this sentence: "the specific role . . .  are clearly visible." I think this latter part can be deleted, and combine the earlier part with the previous sentence, i.e. "Comparing the time series . . .  variables, especially when considering . . .  (dashed lines)."*

  We updated the text according to your suggestion.

- *Figure 15: DMS emissions at 100 % are indicated only for two models?*

  Thank you for this remark, the 100 % was indeed missing for one model, we updated the Figure.

- *Section 5: I think this section should be named as "Discussion and Conclusions", as it is quite extensive for just Conclusions.*

  We prefer to keep the current title for this section, since there is no further discussion of our results.

- *L694: I understand L11 has sampling biases. However, should W20 have similar sampling biases because it also relies on the same dataset (well, twice more) for both training and evaluation (L196-204)? So unless W20 accounts for a preferential sampling of DMS-productive conditions incorporated into the dataset (L189), how can we conclude that W20 does not suffer from similar sampling biases as in L11?*

  Two reasons can explain this. First, W20 handles the extreme DMS values differently than L11: while L11 removed data that were above the 99.9 percentile (with the indication that "The 0.1 % eliminated were seawater DMS concentrations greater than 148 nM"), W20 removed "ultralow ($< 0.1$ nM) and ultrahigh ($> 100$ nM) DMS measurements". The second reason is L11 and W20 rely on very different methodologies. In L11, the "first guess" field is obtained by extrapolating the averaged available measurements to the entire Longhurst province (see Fig. 1 in Lana et al., 2011). This can lead to important bias if the available measurements are not representative enough of the actual province. Conversely, the ANN developed by W20 uses the actual conditions (SST, SSS, nutients, etc.) at the measurement location. The values computed by the neural networks account for the variable conditions, and even if the measurements were

carried out in DMS productive area, the ANN is expected to account for this, and compute lower DMS concentrations in low-productivity places. To clarify, we added the following elements in the text:

L171: "... the largest values above the 99.9 percentile are removed (i.e., values above 148 nM)."

L197: "This study relies on the same database of in-situ DMS measurements, which now contains twice as many measurements (over 93k after removing concentrations below 0.1 nM and above 100 nM) as in the study of Lana et al. (2011)."

Regarding the second argument, in line with the remarks and suggestions of Martí Galí in Review #1 (see our answer to his general comment #1), we elaborated and clarified the text about potential biases in L11 as follows:

L185: "Thus, as pointed out by Tesdal et al. (2016), small scale features are transformed into large scale ones by the interpolation procedure, and anomalous values observed at local scale could induce bias when extrapolated across data-sparse regions. This is illustrated by Hayashida et al. (2020), who show that the entire Arctic region in L11 is based on extremely limited data (0–4 % areal coverage north of 60°N). The resulting extrapolation of open water DMS concentration to sea-ice covered areas, where primary production is presumably lower, may lead to a positive bias in L11."

We also rephrased the sentence at L694 to refer to both sampling and extrapolation biases, as follows: "As concluded by previous authors (see for instance Galí et al., 2018, Sect. 4.1), the widely used L11 climatology likely overestimates climatological surface DMS concentration at the spatial resolution of climate models due to the combination of scarce and biased sampling."

- *L719: Briefly state what the conclusions are.*

  We rephrased the sentence as follows:
  "Our analyses using CMIP6 ESMs confirm the conclusions of Tesdal et al. (2016) that global DMS emission depends primarily on global mean surface ocean DMS concentration, while the spatial distribution of DMS concentration and the parameterisation of ocean-atmosphere exchange coefficient are of secondary importance. Our study further demonstrate that to first order, changes in marine global DMS concentration determine the evolution of the global DMS emission to the atmosphere."

- *L725: I don't think the word "overcome" is appropriate here. Overcome suggests one effect counteracts and defeats another effect. The trend of DMS concentration is neutral (neither increasing/decreasing; Figure 14 bottom panel), so it's just that the positive trend of ice-free extent drives the trend of DMS emission.*

  Please mind that Fig. 14 shows only the 1950–2014 period, during which the trends are indeed weak or absent. Conversely, there is a marked projected trend over the 21$^{th}$ century (ssp585 simulations, see Fig. 10). In the Polar N biome, depending on the model, the trend in DMS concentration is weakly negative (UKESM), weakly positive (CNRM and MIROC) and markedly positive (NorESM) but for all four models, the trend in DMS emission is strongly positive due to the sea-ice retreat. We thus believe that the word "overcome" is appropriate.

- *L750: Data availability for the CMIP6 models should also be mentioned here.*

  We added the following sentence: "Datasets from CMIP6 simulations are available from every ESGF node, such as `https://esgf-node.ipsl.upmc.fr/search/cmip6-ipsl/` (last accessed 14 April 2021)."

**References**

[revised manuscript text omitted]

---

## Author Comment (AC3)

**Review 3: Nadja Steiner**

Dear Nadja Steiner, Thank you very much for your positive review of our paper, and for the interesting points that you raised. Your suggestions are very useful and help to improve the manuscript. Please find below our responses to your comments, which appear below in italics. Our proposed amendments to the text of our paper appear in blue. Page and line numbers refer to the version of the paper you reviewed.

*The paper by Bock et al evaluates the ocean dimethylsulfide concentrations and emissions in CMIP6 models.*

*Only two of the DMS models are actually prognostic DMS models, while the other two use different diagnostic algorithms. Hence the comparison is difficult to evaluate. Quite a bit of work has been done comparing the various algorithms estimating DMS based on Chl, light MLD etc. The way the simulated DMS is used (or not used) in the models also varies significantly - One model calculates DMS prognostically but does not use it, one calculates DMS prognostically and uses it in the atmosphere chemistry module for conversion and presumably aerosol formation, one model calculates DMS diagnostically and uses it directly in the aerosol module and the last one calculates DMS diagnostically and uses it in the atmosphere chemistry and aerosol formation module.*

*Essentially, the authors ran into the not unfamiliar problem to try compare multiple models which are not only vastly different with respect to the parameterizations of the variable in question, but also with respect to several other components such as gas exchange velocity, atmospheric feedbacks etc. In addition the climatologies used for evaluation have their own issues and potential errors. This makes it extremely difficult to understand respective differences among the output.*

*However, I feel that despite these difficulties the authors did an excellent job in comparing the models, and identifying and describing the cause of differences. The evaluations are well linked and brought into context with earlier estimates and evaluations which helps to assess advancement from those and the uncertainties and concerns are well stated. This does provide scientific value despite the variety of parameterisations in the models.*

*The manuscript is well written and the evaluation procedures are sound. In fact the manuscript provides an excellent template for future analysis in (hopefully) more coordinated DMS model intercomparisons (Maybe consider adding a note with such a recommendation in the paper).*

*Hence, I recommend publication of the manuscript with minor changes.*

*What I am missing is a brief note on the potential impact of DMS emissions in the atmosphere, i.e. a note indicating that areas with highest emissions are not necessarily those where the emissions have the highest impact, particularly with respect to the Arctic (see notes below).*

Please see our response to your comment about L527 below that covers also this comment.

*Since the authors do include a focus section on the Arctic, I would also recommend to include a sentence or two on the missing ice algae component*

*(see note below).*

See our answer to your specific comment at L634/635.

*There are a few spelling mistakes which I noted ( if I caught them)*

*Detailed comments:*

- *l23 insert space after DMS) , rm "as" after considered*

  corrected

- *l26 "sulfate aerosols formed DMS" - from DMS?*

  corrected

- *l27 could mention the Arctic here, too (Abbatt citation?)*

  Thank you for this suggestion, we completed this sentence as follows:
  "Among natural aerosols, sulfate aerosols formed from DMS represent a major part of the aerosol-climate interactions in large pristine regions such as the Southern Ocean (Mulcahy et al., 2018, and references therein) or the Arctic (Abbatt et al., 2019, and references therein).

- *l60 measurements*

  corrected

- *l107 "the" marine DMS cycle*

  corrected

- *l135 adjusted to compensate... for what?*

  We rephrased this sentence:
  "During calibration of UKESM1-0-LL, the minimum DMS concentration ($a$) was changed to 1 nM and the threshold ($s = 1.56$) adjusted to compensate (Sellar et al., 2019)"
  which now reads:
  "During calibration of UKESM1-0-LL, the minimum DMS concentration ($a$) was changed to 1 nM and the threshold ($s$) was adjusted to 1.56 (Sellar et al., 2019).

- *l205 unclear what "these " refers to in "to compare the skills of these methods" which makes the following sentence confusing. Please clarify what is compared to what etc.*

  The text now reads:"Along with this ANN method, the authors also performed conventional linear and multi-linear regression analyses, to compare the skills of the three methods."

- *l206 The yearly mean of this climatology - what does "this climatology" refer to? ANN?*

  The text now reads:"The yearly mean of the climatology derived from the ANN method is shown in Fig. 1."

- *L220 also issues with CDOM in coastal areas (see Hayashida et al. 2020)*

  Thank you for pointing out this specific issue. We mentioned it in an additional sentence:: "In coastal regions, the remotely sensed Chl signal may be biased by the presence of riverine coloured dissolved organic matter (CDOM), and ultimately lead to an overestimation of the computed DMS concentration (Galí et al., 2019; Hayashida et al., 2020)."

- *L344 higher than*
  corrected

- *L363 The is paragraph seams a bit too generalized. E.g. it might be relevant to note that some of the regions indicates as poorly represented show hardly any variation and generally a much smaller range than the regions which have a clearer signal in both model and obs. I notice that the lower seasonality is discussed at the end of the section, but would help to briefly mention at time of the figure discussion.*

  We agree that the low seasonality and small concentrations contribute to the poor match between observations and models. However, regions 37 (Southern Equatorial Pacific) or 38 (Northern Equatorial Pacific) illustrate that the agreement between models and climatology is sometimes much better despite the weak seasonality and low DMS concentration.
  We rephrased the last sentence of this paragraph:
  "Conversely, the seasonal cycle in equatorial regions of both Atlantic (regions 7, 8, 14, 17) and Pacific Oceans (regions 39, 40, 41) are poorly reproduced by all the models."
  to:
  "Equatorial regions are characterised by a weak seasonality and low DMS concentrations, and models poorly reproduce the climatology in these provinces (regions 9, 17 and 41 for instance)."

- *L417 mirror - mirrors*
  corrected

- *L506 a weakly*
  corrected

- *L517 To help understanding => To help understand*
  corrected (and L410 as well)

- *L526 The coastal biome*
  corrected

- *L527 I am a bit concerned with the statement: "improving the models in the low latitudes regions is needed to gain confidence in the predicted global trends of DMS" => While this may be true, the question is if the global emission is the relevant one or if the emission is more important to improve in regions where DMS emissions have significant impact (as in the clean polar atmosphere) eventhough it might be a smaller contribution*

*to the global mean (e.g, Abbatt et al. 2019,https://doi.org/10.5194/acp-19-2527-2019, and references therein)- maybe something to pick up in the discussion???*

What we meant in this sentence was that globally mean values we analyse, i.e., annual global DMS sea surface concentrations and DMS fluxes, are determined to a large extent by trade/low-latitude values. It is defacto necessary that models perform reasonably well in these low-latitude regions if one wants to be reasonably confident in conclusions concerning global mean quantities. This being said, we agree fully with your statement, regions with the highest emissions do not necessarily have the highest climate impact. We looked further in the literature to better apprehend this point but we could not identify fully relevant statements. The literature is quite abundant with regards to DMS in the high latitude regions, i.e. Abbatt et al. (2019); Galí et al. (2018) for instance with all references thererin, but it is much less so with regards to DMS in the tropical regions. In these tropical regions an emerging topic is that of the role of DMS in local climate of coral reefs is recognised as amongst the strongest individual sources of natural atmospheric sulful (Jackson et al., 2020). Furthermore, measurements of DMS air-sea fluxes have been reported in the western Indian Ocean (Zavarsky et al., 2018) and have illustrated the regional coupling with aerosol products.

We included additional details in our introduction and the text now reads: "A recent estimate deduced from three CMIP6 models (GISS-E2-1-G-CC, NorESM2-LM, UKESM1-0-LL) suggests a slight amplification of global warming due to a positive feedback of $0.005 \pm 0.006$ W $m^{-2}$ $K^{-1}$ (Thornhill et al., 2021). These global estimates hide large regional differences both in terms of radiative forcing and in terms of changes in DMS emissions under global warming (Thornhill et al., 2021). So far, studies have focused more closely on high latitudes regions eventhough a few recent ones demonstrate the dominant role of DMS on marine low cloud albedo over most oceans (e.g., Quinn et al., 2017) or illustrate regional impacts on low latitudes (Zavarsky et al., 2018). In polar regions, studies ..."

- *L615 suggest including reference to Hayashida et al 2020 (10.1029/2019GB006456) DMS model for the Arctic (also provides detailed comparison with G19), including a note on the ice algae contribution which is not represented in the described ESMs (see note below)*

We have added Hayashida et al. (2020) reference in several parts of the text, including in this section 4.3 that focuses on the Arctic, as you and Martí Galí suggested.

- *Also suggest a note here on the impact of DMS in an otherwise clean atmosphere (Arctic spring summer, see note above)*

We have added Abbatt et al. (2019) reference on this issue in the text of our introduction (see above).

- *L634/635 "This means that the models consistently predict lower DMS concentration below the sea-ice, in line with reduced photosynthetically available radiation." suggest adding a note on ice algae DMS production here*

  We added the following sentence in the text : "This means that the models consistently predict lower DMS concentration below the sea-ice, in line with reduced photosynthetically active radiation. We want to note here however that a number of recent studies highlighted the large DMS production of ice algae and acknowledged that models likely underestimate the contribution of bottom ice DMS (Hayashida et al., 2020, and references therein).

**References**

[revised manuscript text omitted]